# Beyond neural scaling laws:
# beating power law scaling via data pruning

**Ben Sorscher**[*1]  **Robert Geirhos**[*2]  **Shashank Shekhar**[3]

**Surya Ganguli**[1,3§]  **Ari S. Morcos**[3§]

[*]equal contribution
[1]Department of Applied Physics, Stanford University
[2]University of Tübingen
[3]Meta AI (FAIR)
[§]Joint senior authors

## Abstract

Widely observed neural scaling laws, in which error falls off as a power of the training set size, model size, or both, have driven substantial performance improvements in deep learning. However, these improvements through scaling alone require considerable costs in compute and energy. Here we focus on the scaling of error with dataset size and show how in theory we can break beyond power law scaling and potentially even reduce it to exponential scaling instead if we have access to a high-quality data pruning metric that ranks the order in which training examples should be discarded to achieve any pruned dataset size. We then test this improved scaling prediction with pruned dataset size empirically, and indeed observe better than power law scaling in practice on ResNets trained on CIFAR-10, SVHN, and ImageNet. Next, given the importance of finding high-quality pruning metrics, we perform the first large-scale benchmarking study of ten different data pruning metrics on ImageNet. We find most existing high performing metrics scale poorly to ImageNet, while the best are computationally intensive and require labels for every image. We therefore developed a new simple, cheap and scalable self-supervised pruning metric that demonstrates comparable performance to the best supervised metrics. Overall, our work suggests that the discovery of good data-pruning metrics may provide a viable path forward to substantially improved neural scaling laws, thereby reducing the resource costs of modern deep learning.

## 1 Introduction

Empirically observed neural scaling laws [1, 2, 3, 4, 5, 6, 7, 8] in many domains of machine learning, including vision, language, and speech, demonstrate that test error often falls off as a power law with either the amount of training data, model size, or compute. Such power law scaling has motivated significant societal investments in data collection, compute, and associated energy consumption. However, power law scaling is extremely weak and unsustainable. For example, a drop in error

---

[*]work done during an internship at Meta AI (FAIR)

36th Conference on Neural Information Processing Systems (NeurIPS 2022).

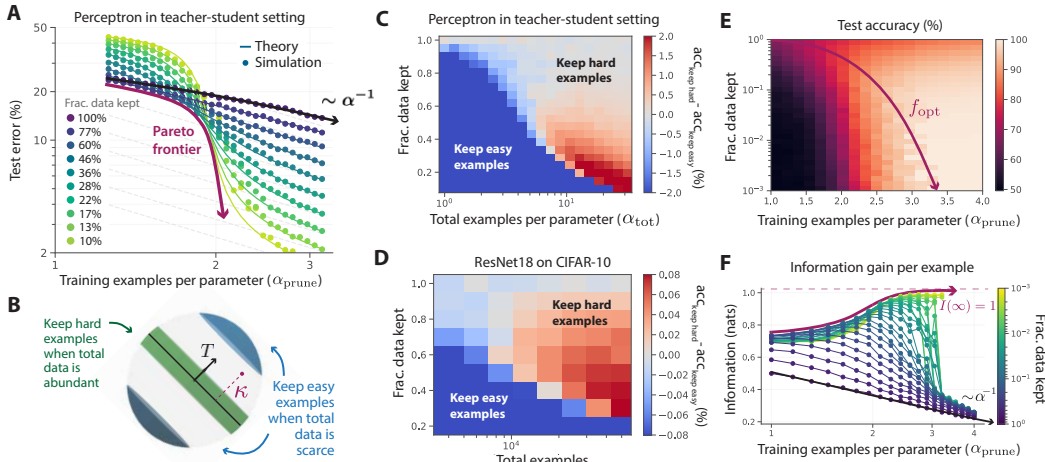

Figure 1: Our analytic theory of data pruning predicts that power law scaling of test error with respect to dataset size can be beaten. **A:** Test error as a function of $\alpha_{\text{prune}} = f\alpha_{\text{tot}}$ with $\theta = 0$. We observe an excellent match between our analytic theory (solid curves) and numerical simulations (dots) of perceptron learning at parameters N=200 (here: N=200 constant throughout figure). The red curve indicates the Pareto optimal test error $\varepsilon$ achievable from a tradeoff between $\alpha_{\text{tot}}$ and $f$ at fixed $\alpha_{\text{prune}}$. **B:** We find that when data is abundant (scarce) corresponding to large (small) $\alpha_{\text{tot}}$, the better pruning strategy is to keep the hard (easy) examples. **C:** Color indicates difference in test error in keeping hard versus easy examples, revealing the change in strategy in (B). **D:** We tested this prediction on a ResNet18 trained on CIFAR-10, finding remarkably the same shift in optimal pruning strategy under the EL2N metric. **E:** Test accuracy as a function of $f$ and $\alpha_{\text{prune}}$. For every fixed $\alpha_{\text{prune}}$, there is an optimal $f_{\text{opt}}$ (purple curve). **F:** $I(\alpha_{\text{prune}})$ for different $f$.

from 3% to 2% might require an *order of magnitude* more data, compute, or energy. In language modeling with large transformers, a drop in cross entropy loss from about 3.4 to 2.8 nats[2] requires *10 times* more training data (Fig. 1 in [2]). Also, for large vision transformers, an additional 2 *billion* pre-training data points (starting from 1 billion) leads to an accuracy gain on ImageNet of a few percentage points (Fig. 1 in [7]). Here we ask whether we might be able to do better. For example, can we achieve exponential scaling instead, with a good strategy for selecting training examples? Such vastly superior scaling would mean that we could go from 3% to 2% error by only adding a few carefully chosen training examples, rather than collecting 10× more random ones.

Focusing on scaling of performance with training dataset size, we demonstrate that exponential scaling is possible, both in theory and practice. The key idea is that power law scaling of error with respect to data suggests that many training examples are highly redundant. Thus one should in principle be able to prune training datasets to much smaller sizes and train on the smaller pruned datasets without sacrificing performance. Indeed some recent works [9, 10, 11] have demonstrated this possibility by suggesting various metrics to sort training examples in order of their difficulty or importance, ranging from easy or redundant examples to hard or important ones, and pruning datasets by retaining some fraction of the hardest examples. However, these works leave open fundamental theoretical and empirical questions: When and why is successful data pruning possible? What are good metrics and strategies for data pruning? Can such strategies beat power law scaling? Can they scale to ImageNet? Can we leverage large *unlabeled* datasets to successfully prune labeled datasets? We address these questions through both theory and experiment. Our main contributions are:

1. Employing statistical mechanics, we develop a new analytic theory of data pruning in the student-teacher setting for perceptron learning, where examples are pruned based on their teacher margin, with large (small) margins corresponding to easy (hard) examples. Our theory quantitatively matches numerical experiments and reveals two striking predictions:

---

[2]However, note that nats is on a logarithmic scale and and small improvements in nats can lead to large improvements in downstream tasks.

(a) The optimal pruning strategy changes depending on the amount of initial data; with abundant (scarce) initial data, one should retain only hard (easy) examples.

(b) Exponential scaling is possible with respect to pruned dataset size provided one chooses an increasing Pareto optimal pruning fraction as a function of initial dataset size.

2. We show that the two striking predictions derived from theory hold also in practice in much more general settings. Indeed we empirically demonstrate signatures of exponential scaling of error with respect to pruned dataset size for ResNets trained from scratch on SVHN, CIFAR-10 and ImageNet, and Vision Transformers fine-tuned on CIFAR-10.

3. Motivated by the importance of finding good quality metrics for data pruning, we perform a large scale benchmarking study of 10 different data pruning metrics at scale on ImageNet, finding that most perform poorly, with the exception of the most compute intensive metrics.

4. We leveraged self-supervised learning (SSL) to developed a new, cheap *unsupervised* data pruning metric that does *not* require labels, unlike prior metrics. We show this unsupervised metric performs comparably to the best supervised pruning metrics that require labels and much more compute. This result opens the door to the exciting possibility of leveraging pre-trained foundation models to prune new datasets even before they are labeled.

Overall these results shed theoretical and empirical insights into the nature of data in deep learning and our ability to prune it, and suggest our current practice of collecting extremely large datasets may be highly inefficient. Our initial results in beating power law scaling motivate further studies and investments in not just inefficently collecting large amounts of random data, but rather, intelligently collecting much smaller amounts of carefully selected data, potentially leading to the creation and dissemination of *foundation datasets*, in addition to foundation models [12].

## 2 Background and related work

Our work brings together 3 largely disparate strands of intellectual inquiry in machine learning: (1) explorations of different metrics for quantifying differences between individual training examples; (2) the empirical observation of neural scaling laws; and (3) the statistical mechanics of learning.

### 2.1 Pruning metrics: not all training examples are created equal

Several recent works have explored various metrics for quantifying individual differences between data points. To describe these metrics in a uniform manner, we will think of all of them as ordering data points by their difficulty, ranging from "easiest" to "hardest." When these metrics have been used for data pruning, the hardest examples are retained, while the easiest ones are pruned away.

**EL2N scores.** For example [10] trained small ensembles (of about 10) networks for a very short time (about 10 epochs) and computed for every training example the average $L_2$ norm of the error vector (EL2N score). Data pruning by retaining only the hardest examples with largest error enabled training from scratch on only $50\%$ and $75\%$ of CIFAR-10 and CIFAR-100 respectively without any loss in final test accuracy. However the performance of EL2N on ImageNet has not yet been explored.

**Forgetting scores and classification margins.** [9] noticed that over the entire course of training, some examples are learned early and never forgotten, while others can be learned and unlearned (i.e. forgotten) repeatedly. They developed a forgetting score which measures the degree of forgetting of each example. Intuitively examples with low (high) forgetting scores can be thought of as easy (hard) examples. [9] explored data pruning using these metrics, but not at ImageNet scale.

**Memorization and influence.** [13] defined a memorization score for each example, corresponding to how much the probability of predicting the correct label for the example increases when it is present in the training set relative to when it is absent; a large increase means the example must be memorized (i.e. the remaining training data do not suffice to correctly learn this example). Additionally [13] also considered an influence score that quantifies how much adding a particular example to the training set increases the probability of the correct class label of a test example. Intuitively, low memorization and influence scores correspond to easy examples that are redundant with the rest of the data, while high scores correspond to hard examples that must be individually learned. [13] did not use these

scores for data pruning as their computation is expensive. We note since memorization explicitly approximates the increase in test loss due to removing each individual example, it is likely to be a good pruning metric (though it does not consider interactions).

**Ensemble active learning.** Active learning iterates between training a model and selecting new inputs to be labeled [14, 15, 16, 17, 18]. In contrast, we focus on data pruning: one-shot selection of a data subset sufficient to train to high accuracy from scratch. A variety of coreset algorithms (e.g. [19]) have been proposed for this, but their computation is expensive, and so data-pruning has been less explored at scale on ImageNet. An early clustering approach [20] allowed training on 90% of ImageNet without sacrificing accuracy. Notably [11] reduced this to 80% by training a large ensemble of networks on ImageNet and using ensemble uncertainty to define the difficulty of each example, with low (high) uncertainty corresponding to easy (hard) examples. We will show how to achieve similar pruning performance without labels or the need to train a large ensemble.

**Diverse ensembles (DDD).** [21] assigned a score to every ImageNet image, given by the number of models in a diverse ensemble (10 models) that misclassified the image. Intuitively, low (high) scores correspond to easy (hard) examples. The pruning performance of this metric remains unexplored.

**Summary.** We note: (1) only one of these metrics has tested well for its efficacy in data pruning at scale on ImageNet; (2) *all* of these metrics require label information; (3) there is no theory of when and why data pruning is possible for any of these metrics; and (4) none of these works suggest the possibility of exponential scaling. We thus go beyond this prior work by benchmarking the data pruning efficacy of not only these metrics but also a new unsupervised metric we introduce that does not require label information, all at scale on ImageNet. We also develop an analytic theory for data-pruning for the margin metric that predicts not only the possibility of exponential scaling but also the novel finding that retaining easy instead of hard examples is better when data is scarce.

## 2.2 Neural scaling laws and their potential inefficiency

Recent work [1, 2, 3, 4, 5, 6, 7, 8] has demonstrated that test loss $\mathcal{L}$ often falls off as a power law with different resources like model parameters ($N$), number of training examples ($P$), and amount of compute ($C$). However, the exponents $\nu$ of these power laws are often close to 0, suggesting potentially inefficient use of resources. For example, for large models with lots of compute, so that the amount of training data constitutes a performance bottleneck, the loss scales as $\mathcal{L} \approx P^{-\nu}$. Specifically for a large transformer based language model, $\nu = 0.095$, which implies *an order of magnitude* increase in training data drops cross-entropy loss by only about 0.6 nats (Fig. 1 in [2]). In neural machine translation experiments $\nu$ varies across language pairs from 0.35 to 0.48 (Table 1 in [5]). Interestingly, [8] explored a fixed computation budget $C$ and optimized jointly over model size $N$ and training set size $P$, revealing that scaling both $N$ and $P$ commensurately as $C$ increases is compute optimal, and can yield smaller high performing models (trained on more data) than previous work. Nevertheless, for a transformer based language model, a $100\times$ increase in compute, corresponding to $10\times$ increases in *both* model size and training set size, leads to a drop in cross-entropy loss of only about 0.5 nats (Fig. 2 in [8]). Similar slow scaling holds for large vision transformers where adding 2 billion pre-training images reduces ImageNet performance by a few percentage points (Fig. 1 in [7]). While all of these results constitute significant improvements in performance, they do come at a substantial resource cost whose fundamental origin arises from power law scaling with small exponents. Recent theoretical works [22, 23, 24] have argued that the power law exponent is governed by the dimension of a data manifold from which training examples are uniformly drawn. Here we explore whether we can beat power law scaling through careful data selection.

## 2.3 Statistical mechanics of perceptron learning

Statistical mechanics has long played a role in analyzing machine learning problems (see e.g. [25, 26, 27, 28] for reviews). One of the most fundamental applications is perceptron learning in the student-teacher setting [29, 30], in which random i.i.d. Gaussian inputs are labeled by a teacher perceptron to construct a training set. The test error for another student perceptron learning from this training set then scales as a power law with exponent $-1$ for such data. Such perceptrons have also been analyzed in an active learning setting where the learner is free to design *any* new input to be

labeled [31, 32], rather than choose from a fixed set of inputs, as in data-pruning. Recent work [33] has analyzed this scenario but focused on message passing algorithms that are tailored to the case of Gaussian inputs and perceptrons, and are hard to generalize to real world settings. In contrast we analyze margin based pruning algorithms that are used in practice in diverse settings, as in [9, 10].

## 3 An analytic theory of data pruning

To better understand data pruning, we employed the replica method from statistical mechanics [34] to develop an analytic theory of pruning for the perceptron in the student-teacher setting [25] (see App. A for detailed derivations of all results). Consider a training dataset of $P$ examples $\{\mathbf{x}^\mu, y^\mu\}_{\mu=1,\dots,P}$ where $\mathbf{x}^\mu \in \mathbb{R}^N$ are i.i.d. zero mean unit variance random Gaussian inputs and $y^\mu = \text{sign}(\mathbf{T} \cdot \mathbf{x}^\mu)$ are labels generated by a teacher perceptron with weight vector $\mathbf{T} \in \mathbb{R}^N$. We work in the high dimensional statistics limit where $N, P \to \infty$ but the ratio $\alpha_{\text{tot}} = \frac{P}{N}$ of the number of total training examples to parameters remains $O(1)$. We then consider a pruning algorithm used in [9, 10], namely: (1) train a probe student perceptron for very few epochs on the training data, obtaining weights $\mathbf{J}_{\text{probe}}$; (2) compute the margin $m^\mu = \mathbf{J}_{\text{probe}} \cdot (y^\mu \mathbf{x}^\mu)$ of each training example, where large (small) margins correspond to easy (hard) examples; (3) construct a pruned dataset of size $P_{\text{prune}} = fP$, where $f$ is the fraction of examples kept, by retaining the $P_{\text{prune}}$ hardest examples, (4) train a new perceptron to completion on the smaller dataset with a smaller ratio $\alpha_{\text{prune}} = \frac{P_{\text{prune}}}{N}$ of examples to parameters.

We are interested in the test error $\varepsilon$ of this final perceptron as a function of $\alpha_{\text{tot}}$, $f$, and the angle $\theta$ between the probe student $\mathbf{J}_{\text{probe}}$ and the teacher $\mathbf{T}$. Our theory approximates $\mathbf{J}_{\text{probe}}$ as simply a random Gaussian vector conditioned to have angle $\theta$ with the teacher $\mathbf{T}$. Under this approximation we obtain an analytic theory for $\varepsilon(\alpha_{\text{tot}}, f, \theta)$ that is asymptotically exact in the high dimensional limit (App. A). We first examine results when $\theta = 0$, so we are pruning training examples according to their veridical margins with respect to the teacher (Fig. 1A). We find two striking phenomena, each of which constitute predictions in real-world settings that we will successfully confirm empirically.

**The best pruning strategy depends on the amount of initial data.** First, we note the test error curve for $f = 1$ in Fig. 1A corresponding to no pruning, or equivalently to *randomly* pruning a larger dataset of size $\alpha_{\text{tot}}$ down to a size $\alpha_{\text{prune}}$, exhibits the well known classical perceptron learning power law scaling $\varepsilon \propto \alpha_{\text{prune}}^{-1}$. Interestingly though, for small $\alpha_{\text{tot}}$, keeping the hardest examples performs *worse* than random pruning (lighter curves above darkest curve for small $\alpha_{\text{prune}}$ in Fig. 1A). However, for large $\alpha_{\text{tot}}$, keeping the hardest examples performs *substantially better* than random pruning (lighter curves below darkest curve for large $\alpha_{\text{prune}}$ in Fig. 1A). It turns out keeping the *easiest* rather than hardest examples is a better pruning strategy when $\alpha_{\text{tot}}$ is small (Fig. 1C). If one does not have much data to start with, it is better to keep the easiest examples with largest margins (i.e. the blue regions of Fig. 1B) to avoid overfitting. The easiest examples provide coarse-grained information about the target function, while the hard examples provide fine-grained information about the target function which can prevent the model from learning if one starts with lots of data. In cases where overfitting is less of an issue, it is best to keep the hardest examples with smallest margin that provide more information about the teacher's decision boundary (i.e. the green region of Fig. 1B). Intuitively, in the limited data regime, it is challenging to model outliers since the basics are not adequately captured; hence, it is more important to keep easy examples so that the model can get to moderate error. However, with a larger dataset, the easy examples can be learned without difficulty, making modeling outliers the fundamental challenge.

Fig. 1C reveals which pruning strategy is best as a joint function of $\alpha_{\text{tot}}$ and $f$. Note the transition between optimal strategies becomes sharper at small fractions $f$ of data kept. This transition between optimal pruning strategies can be viewed as a prediction in more general settings. To test this prediction we trained a ResNet18 on pruned subsets of the CIFAR-10 dataset (Fig. 1D), and observed strikingly similar behavior, indicating the prediction can hold far more generally, beyond perceptron learning. Interestingly, [9, 10] missed this transition, likely because they started pruning from large datasets.

**Pareto optimal data pruning can beat power law scaling.** A second prediction of our theory is that when keeping a *fixed* fraction $f$ of the hardest examples as $\alpha_{\text{tot}}$ increases (i.e. constant color curves in Fig. 1A), the error initially drops exponentially in $\alpha_{\text{prune}} = f\alpha_{\text{tot}}$, but then settles into the universal power law $\varepsilon \propto \alpha_{\text{prune}}^{-1}$ for all fixed $f$. Thus there is no asymptotic advantage to data

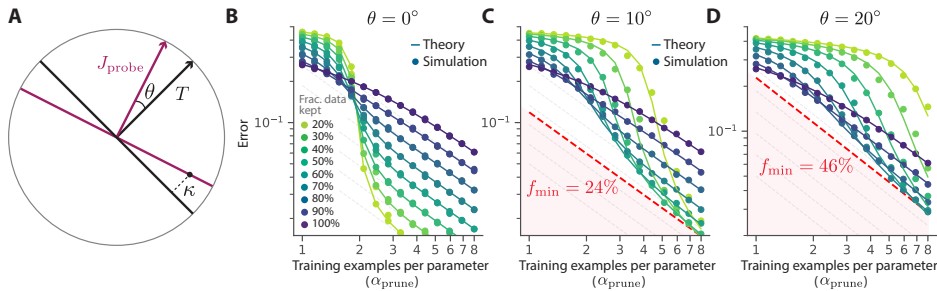

Figure 2: Data pruning with an imperfect metric. **A:** Weight vectors and decision boundaries for a teacher (black) and probe student (red) separated by angle $\theta$. The black point has margin $0$ ($\kappa$) w.r.t. the probe (teacher). **B–D:** Test error as a function of $\alpha_{\text{prune}}$ for different $f$ and different $\theta$.

pruning at a fixed $f$. However, by pruning more aggressively (smaller $f$) when given more initial data (larger $\alpha_{\text{tot}}$), one can achieve a Pareto optimal test error as a function of pruned dataset size $\alpha_{\text{prune}}$ that remarkably traces out at least an exponential scaling law (Fig. 1A, purple curve). Indeed our theory predicts for each $\alpha_{\text{prune}}$ a Pareto optimal point in $\alpha_{\text{tot}}$ and $f$ (subject to $\alpha_{\text{prune}} = f\alpha_{\text{tot}}$), yielding for every fixed $\alpha_{\text{prune}}$ an optimal $f_{\text{opt}}$, plotted in Fig. 1E. Note $f_{\text{opt}}$ decreases with $\alpha_{\text{prune}}$ indicating more aggressive pruning (smaller $f_{\text{opt}}$) of original datasets of larger size $\alpha_{\text{tot}}$ is required to obtain larger Pareto optimal pruned datasets of size $\alpha_{\text{prune}}$. We will test this striking scaling prediction in Fig. 3.

**Beating power law scaling: an information-theoretic perspective.** Classical randomly selected data generates slow power law error scaling because each extra training example provides less new information about the correct decision boundary than the previous example. More formally, let $S(\alpha_{\text{tot}})$ denote the typical entropy of the posterior distribution over student perceptron weights consistent with a training set of size $\alpha_{\text{tot}}$. The information gain $I(\alpha_{\text{tot}})$ due to additional examples beyond $\alpha_{\text{tot}}$ can be defined as the rate at which the posterior entropy is reduced: $I(\alpha_{\text{tot}}) = -\frac{d}{d\alpha_{\text{tot}}}S(\alpha_{\text{tot}})$. In classical perceptron learning $I(\alpha_{\text{tot}})$ decays to zero as a power law in $\alpha_{\text{tot}}$, reflecting a vanishing amount of information per each new example, leading to the slow power law decay of test error $\varepsilon \propto \alpha_{\text{tot}}^{-1}$. However, data pruning can increase the information gained per example by pruning away the uninformative examples. To show this, we generalized the replica calculation of the posterior entropy $S$ and information gain $I$ from random datasets of size $\alpha_{\text{tot}}$ to pruned datasets of size $\alpha_{\text{prune}}$ (App. A). We plot the resulting information gain $I(\alpha_{\text{prune}})$ for different $f$ in Fig. 1F. For any fixed $f$, $I(\alpha_{\text{prune}})$ will eventually decay as a power law as $\alpha_{\text{prune}}^{-1}$. However, by more aggressively pruning (smaller $f$) datasets of larger size $\alpha_{\text{tot}}$, $I(\alpha_{\text{prune}})$ can converge to a finite value $I(\infty) = 1$ nat/example, resulting in larger pruned datasets only adding useful non-redundant information. Since each new example under Pareto optimal data pruning conveys finite information about the target decision boundary, as seen in Fig. 1F, the test error can decay at least exponentially in pruned dataset size as in Fig. 1A. Classical results [30] have shown that training examples chosen by maximizing the disagreement of a committee of student perceptrons can provide an asymptotically finite information rate, leading to exponential decay in test error. Intriguingly, the Pareto-optimal data pruning strategy we study in this work leads to *faster* than exponential decay, because it includes (partial) information about the target function provided by the probe student (Fig. 11).

**An imperfect pruning metric yields a cross over from exponential to power law scaling.** We next examine the case of nonzero angle $\theta$ between the probe student $\mathbf{J}_{\text{probe}}$ and the teacher $\mathbf{T}$, such that the ranking of training examples by margin is no longer completely accurate (Fig. 2A). Retaining the hard examples with smallest margin with respect to the probe student will always result in pruned datasets lying near the probe's decision boundary. But if $\theta$ is large, such examples might be far from the teacher's decision boundary, and therefore could be less informative about the teacher (Fig. 2A). As a result our theory, confirmed by simulations, predicts that under nonzero angles $\theta$, the Pareto optimal lower envelope of test error over both $\alpha_{\text{tot}}$ and $f$ initially scales exponentially as a function of $\alpha_{\text{prune}} = f\alpha_{\text{tot}}$ but then crosses over to a power law (Fig. 2BCD). Indeed, at any given nonzero $\theta$, our theory reveals that as $\alpha_{\text{tot}}$ (and therefore $\alpha_{\text{prune}}$) becomes large, one cannot decrease test error any further by retaining less than a minimum fraction $f_{\text{min}}(\theta)$ of all available data. For example when $\theta = 10°$ ($\theta = 20°$) one can do no better asymptotically than pruning down to 24% (46%) of the total

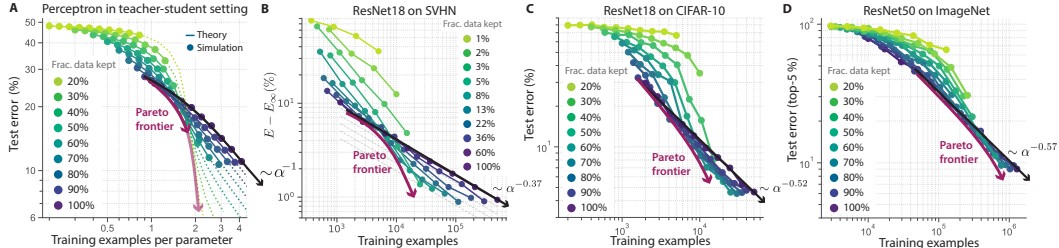

Figure 3: Beating power law scaling in practice. **A–D:** Curves of test error against pruned dataset size in 4 settings. Pruning scores were EL2N [10] for CIFAR-10 and SVHN and memorization [13] for ImageNet. See App. B for all pruning/training details and App. D for similar ImageNet plots with EL2N. Note solid curves reflect performance with a fixed total dataset size; if we prune more aggressively with even larger datasets, scaling could improve further (e.g., dashed lines in **A**). Error curves with no data pruning ($f = 1$) are labeled with their best-fit power law scaling $\sim \alpha^{-\nu}$. (Note that for SVHN in **B** an asymptotic constant error $E(P \to \infty) = 1.1\%$ is subtracted from each of the curves to visualize the power law scaling more clearly.)

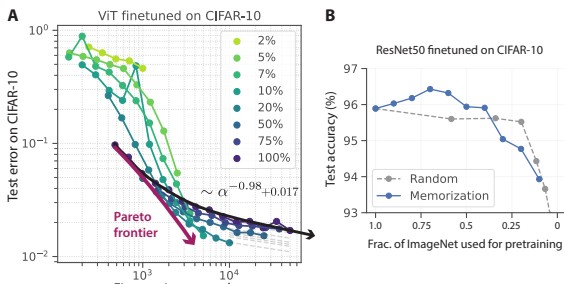

Figure 4: Data pruning improves transfer learning. **A**: CIFAR-10 performance of a ViT pre-trained on all of ImageNet21K and fine-tuned on different pruned subsets of CIFAR-10 under the EL2N metric. **B**: CIFAR-10 performance of ResNet50s pre-trained on different pruned subsets of ImageNet1K and fine-tuned on all of CIFAR-10.

data (Fig. 2CD). As $\theta$ approaches 0, $f_{\min}(\theta)$ approaches 0, indicating that one can prune extremely aggressively to arbitrarily small $f$ while still improving performance, leading to at least exponential scaling for arbitrarily large $\alpha_{\text{prune}}$ in Fig. 2B. However, for nonzero $\theta$, the lack of improvement for $f < f_{\min}(\theta)$ at large $\alpha_{\text{prune}}$ renders aggressive pruning ineffective. This result highlights the importance of finding high quality pruning metrics with $\theta \approx 0$. Such metrics can delay the cross over from exponential to power law scaling as pruned dataset size $\alpha_{\text{prune}}$ increases, by making aggressive pruning with very small $f$ highly effective. Strikingly, in App. Fig. 10 we demonstrate this cross-over in a real-world setting by showing that the test error on SVHN is bounded below by a power law when the dataset is pruned by a probe ResNet18 under the EL2N metric, trained for 4 epochs (weak pruning metric) but not a probe ResNet18 trained for 40 epochs (strong pruning metric).

## 4 Data pruning can beat power law scaling in practice

Our theory of data pruning for the perceptron makes three striking predictions which can be tested in more general settings, such as deep neural networks trained on benchmark datasets: (1) relative to random data pruning, keeping only the hardest examples should *help* when the initial dataset size is large, but *hurt* when it is small; (2) data pruning by retaining a fixed fraction $f$ of the hardest examples should yield power law scaling, with exponent equal to that of random pruning, as the initial dataset size increases; (3) the test error optimized over both initial data set size and fraction of data kept can trace out a Pareto optimal lower envelope that beats power law scaling of test error as a function of pruned dataset size, through more aggressive pruning at larger initial dataset size. We verified all three of these predictions on ResNets trained on SVHN, CIFAR-10, and ImageNet using varying amounts of initial dataset size and fractions of data kept under data pruning (compare theory in Fig. 3A with deep learning experiments in Fig. 3BCD). In each experimental setting we see better than power law scaling at larger initial data set sizes and more aggressive pruning. Moreover we would likely see even better scaling with even larger initial datasets (as in Fig.3A dashed lines).

**Data pruning improves transfer learning.** Modern foundation models are pre-trained on a large initial dataset, and then transferred to other downstream tasks by fine-tuning on them. We therefore examined whether data-pruning can be effective for both reducing the amount of fine-tuning data and the amount of pre-training data. To this end, we first analyzed a vision transformer (ViT) pre-trained on ImageNet21K and then fine-tuned on different pruned subsets of CIFAR-10. Interestingly, pre-trained models allow for far more aggressive data pruning; fine-tuning on only 10% of CIFAR-10 can match or exceed performance obtained by fine tuning on *all* of CIFAR-10 (Fig. 4A). Furthermore Fig. 4A provides a new example of beating power law scaling in the setting of fine-tuning. Additionally, we examined the efficacy of pruning pre-training data by pre-training ResNet50s on different pruned subsets of ImageNet1K (exactly as in Fig. 3D) and then fine-tuning them on all of CIFAR-10. Fig. 4B demonstrates pre-training on as little as $50\%$ of ImageNet can match or exceed CIFAR-10 performance obtained by pre-training on all of ImageNet. Thus intriguingly pruning pre-training data on an upstream task can still maintain high performance on a different downstream task. Overall these results demonstrate the promise of data pruning in transfer learning for both the pre-training and fine-tuning phases.

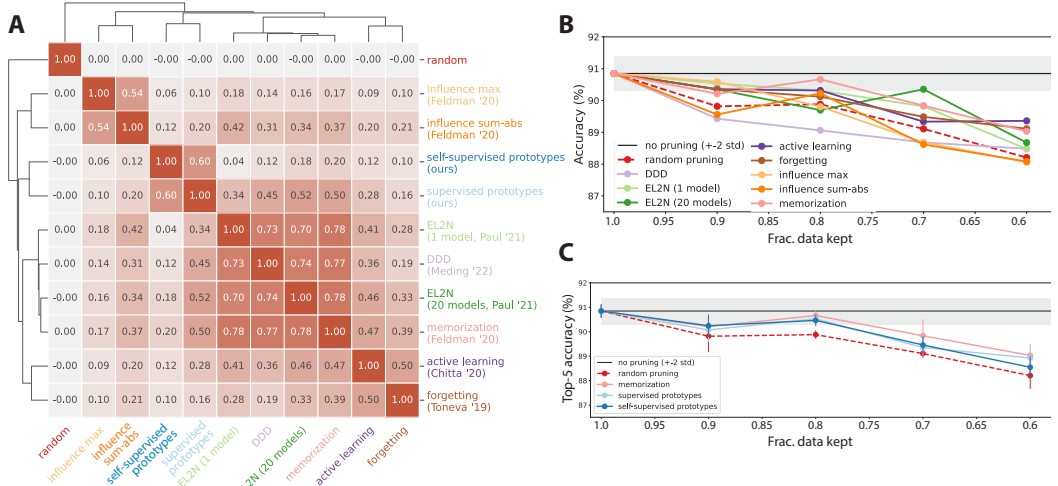

Figure 5: Dataset pruning at ImageNet scale. **A:** Spearman's rank correlation between all pairs of ImageNet metric scores, along with hierarchical clustering (as provided by `seaborn.clustermap`). **B:** Benchmarking existing supervised metrics on ImageNet (top-5 validation accuracy). **C:** Comparing top-5 performance on ImageNet when pruning according to the best existing supervised metric (memorization) and our supervised and self-supervised prototype metrics. In all 3 cases, training on 80% of ImageNet approximates training on 100%. See App. B for pruning and training details.

## 5 Benchmarking supervised pruning metrics on ImageNet

We note that the majority of data pruning experiments have been performed on small-scale datasets (i.e. variants of MNIST and CIFAR), while the few pruning metrics proposed for ImageNet have rarely been compared against baselines designed on smaller datasets. Therefore, it is currently unclear how most pruning methods scale to ImageNet and which method is best. Motivated by how strongly the quality of a pruning metric can impact performance in theory (Fig. 2), we decided to fill this knowledge gap by performing a systematic evaluation of 8 different supervised pruning metrics on ImageNet: two variants of influence scores [13], two variants of EL2N [10], DDD [21], memorization [13], ensemble active learning [11], and forgetting [9]. See Section 2 for a review of these metrics. Additionally, we include two new prototypicality metrics that we introduce in the next section.

We first asked how consistent the rankings induced by different metrics are by computing the Spearman rank correlation between each pair of metrics (Fig. 5A). Interestingly, we found substantial diversity across metrics, though some (EL2N, DDD, and memorization) were fairly similar with rank correlations above $0.7$. However, we observed marked performance differences between metrics: Fig 5BC shows test performance when a fraction $f$ of the hardest examples under each metric are kept in the training set. Despite the success of many of these metrics on smaller datasets, only a

few still match performance obtained by training on the full dataset, when selecting a significantly smaller training subset (i.e. about 80% of ImageNet). Nonetheless, most metrics continue to beat random pruning, with memorization in particular demonstrating strong performance (Fig. 5C). We note that data pruning on ImageNet may be more difficult than data pruning on other datasets, because ImageNet is already carefully curated to filter out uninformative examples.

We found that all pruning metrics amplify class imbalance, which results in degraded performance. To solve this we used a simple 50% class balancing ratio for all ImageNet experiments. Further details and baselines without class balancing are shown in App. H.

# 6  Self-supervised data pruning through a prototypicality metric

Fig. 5 shows many data pruning metrics do not scale well to ImageNet, while the few that do require substantial amounts of compute. Furthermore, all these metrics require labels, thereby limiting their ability to prune data for large-scale foundation models trained on massive unlabeled datasets [12]. Thus there is a clear need for simple, scalable, self-supervised pruning metrics.

To compute a self-supervised pruning metric for ImageNet, we perform $k$-means clustering in the embedding space of an ImageNet pre-trained self-supervised model (here: SWaV [35]), and define the difficulty of each data point by the Euclidean distance to its nearest cluster centroid, or prototype. Thus easy (hard) examples are the most (least) prototypical. Encouragingly, in Fig. 5C, we find our self-supervised prototype metric matches or exceeds the performance of the best supervised metric, memorization, until only 70–80% of the data is kept, despite the fact that our metric does not use labels and is much simpler and cheaper to compute than many previously proposed supervised metrics. See App. Fig. 9 for further scaling experiments using the self-supervised metric.

To assess whether the clusters found by our metric align with ImageNet classes, we compared their overlaps in Fig. 6A. Interestingly, we found alignment for some but not all classes. For example, class categories such as snakes were largely aligned to a small number of unsupervised clusters, while other classes were dispersed across many such clusters. If class information is available, we can enforce alignment between clusters and classes by simply computing a single prototype for each class (by averaging the embeddings of all examples of this class). While originally intended to be an additional baseline metric (called supervised prototypes, light blue in Fig 5C), this metric remarkably outperforms other supervised metrics and largely matches the performance of memorization, which is prohibitively expensive to compute. Moreover, the performance of the best self-supervised and supervised metrics are similar, demonstrating the promise of self-supervised pruning.

One important choice for the self-supervised prototype metric is the number of clusters $k$. We found, reassuringly, our results were robust to this choice: $k$ can deviate one order of magnitude more or less than the true number of classes (i.e. 1000 for ImageNet) without affecting performance (App. F).

To better understand example difficulty under various metrics, we visualize extremal images for our self-supervised prototype metric and the memorization metric for one class (Fig 6B,C). Qualitatively, easy examples correspond to highly similar, redundant images, while hard examples look like idiosyncratic outliers. See App. E, Figs. 12,13,14,15,16,17,18,19 for more classes and metrics.

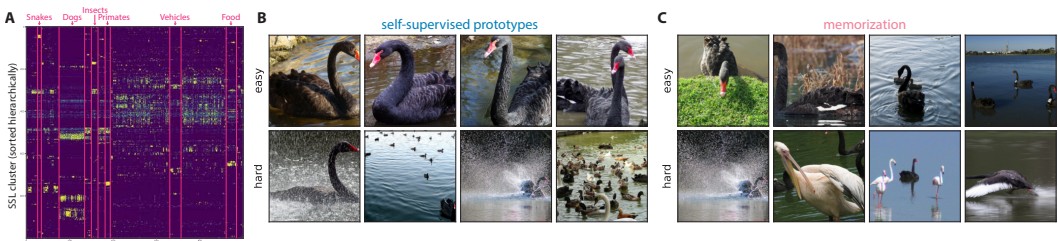

Figure 6: **A:** Heat map where each row denotes the probability that images in a given cluster come from each ImageNet class. **B:** The four easiest and hardest images under our self-supervised pruning metric and the best previously published supervised metric (memorization, shown in **C**) for ImageNet class 100 (`black swan`).

# 7 Discussion

**Summary.** We have shown, both in theory and practice, how to break beyond slow power law scaling of error versus dataset size to faster exponential scaling, through data pruning. Additionally we have developed a simple self-supervised pruning metric that enables us to discard 20% of ImageNet without sacrificing performance, on par with the best and most compute intensive supervised metric.

**Limitations.** The most notable limitation is that achieving exponential scaling requires a high quality data pruning metric. Since most metrics developed for smaller datasets scale poorly to ImageNet, our results emphasize the importance of future work in identifying high quality, scalable metrics. Our self-supervised metric provides a strong initial baseline. Moreover, a key advantage of data pruning is reduced computational cost due to training on a smaller dataset for the same number of epochs as the full dataset (see App. C). However, we found that performance often increased when training on the pruned dataset for the same number of *iterations* as on the full dataset, resulting in the same training time, but additional training epochs. However, this performance gain saturated *before* training time on the pruned dataset approached that on the whole dataset (App. J) thereby still yielding a computational efficiency gain. Overall this tradeoff between accuracy and training time on pruned data is important to consider in evaluating potential gains due to data pruning. Finally, we found that class-balancing was essential to maintain performance on data subsets (App. H). Future work will be required to identify ways to effectively select the appropriate amount of class-balancing.

**Ethical considerations.** A potential negative societal impact could be that data-pruning leads to unfair outcomes for certain groups. We have done a preliminary analysis of how data-pruning affects performance on individual ImageNet classes (App. I), finding no substantial differential effects across classes. However proper fairness tests specific to deployment settings should always be conducted on every model, whether trained on pruned data or not. Additionally, we analyzed the impact of pruning on OOD performance (App. K).

**Outlook: Towards foundation datasets.** We believe the most promising future direction is the further development of scalable, unsupervised data pruning metrics. Indeed our theory predicts that the application of pruning metrics on larger scale datasets should yield larger gains by allowing more aggressive pruning. This makes data pruning especially exciting for use on the massive unlabeled datasets used to train large foundation models (e.g. 400M image-text pairs for CLIP [36], 3.5B Instagram images [37], 650M images for the DALLE-2 encoder [38], 780B tokens for PALM [39]). If highly pruned versions of these datasets can be used to train a large number of different models, one can conceive of such carefully chosen data subsets as *foundation datasets* in which the initial computational cost of data pruning can be amortized across efficiency gains in training many downstream models, just at the initial computational cost of training foundation models is amortized across the efficiency gains of fine-tuning across many downstream tasks. Together, our results demonstrate the promise and potential of data pruning for large-scale training and pretraining.

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
