# Appendix

## Table of Contents

## A   A theory of data-pruning for the perceptron: detailed derivations

All code required to reproduce the theory figures and numerical simulations throughout this paper can be run in the Colab notebook at `https://colab.research.google.com/drive/1in35C6jh7y_ynwuWLBmGOWAgmUgpl8dF?usp=sharing`.

### A.1   Problem setup

In this section we introduce a theory for data pruning in the teacher-student perceptron setting, using the tools of statistical mechanics. We study the problem of classifying a dataset of $P$ examples $\{\mathbf{x}^\mu, y^\mu\}_{\mu=1,...,P}$, where $\mathbf{x}^\mu \sim \mathcal{N}(0, I_N)$ are i.i.d. zero mean unit variance random Gaussian inputs, and $y^\mu = \text{sign}(\mathbf{T} \cdot x)$ are labels generated by a teacher perceptron $\mathbf{T} \in \mathbb{R}^N$, which we will assume is randomly drawn from a uniform distribution on the sphere $\mathbf{T} \sim \text{Unif}(\mathbb{S}^{N-1}(\sqrt{N}))$. We work in the high-dimensional statistics limit where $N, P \to \infty$ but the ratio $\alpha_{\text{tot}} = P/N$ remains $O(1)$. The generalization error of a perceptron trained on such an isotropic dataset is a classical problem (see e.g. [25]). However, we are interested in the setting where the training dataset is not isotropic, but instead has inherited some structure due to data pruning.

In particular, consider pruning the training dataset by keeping only the examples with the smallest margin $|z^\mu| = |\mathbf{J}_{\text{probe}} \cdot \mathbf{x}^\mu|$ along a probe student $\mathbf{J}_{\text{probe}}$. The pruned dataset will follow some distribution $p(z)$ along the direction of $\mathbf{J}_{\text{probe}}$, and remain isotropic in the nullspace of $\mathbf{J}_{\text{probe}}$. In what follows we will derive a general theory for an arbitrary data distribution $p(z)$, and specialize to the case of small-margin pruning only at the very end (in which case $p(z)$ will take the form of a truncated Gaussian). We will also make no assumptions on the form of the probe student $\mathbf{J}_{\text{probe}}$ or the learning rule used to train it; only that $\mathbf{J}_{\text{probe}}$ has developed some overlap with the teacher, quantified by the angle $\theta = \cos^{-1}\left(\frac{\mathbf{J}_{\text{probe}} \cdot \mathbf{T}}{\|\mathbf{J}_{\text{probe}}\|_2 \|\mathbf{T}\|_2}\right)$ (Fig. 2**A**).

After the dataset has been pruned, we consider training a new student $J$ from scratch on the pruned dataset. A typical training algorithm (used in support vector machines and the solution to which SGD converges on separable data) is to find the solution $J$ which classifies the training data with the maximal margin $\kappa = \min_\mu \mathbf{J} \cdot (y^\mu \mathbf{x}^\mu)$. Our goal is to compute the generalization error $\varepsilon_g$ of this student, which is simply governed by the overlap between the student and the teacher, $\varepsilon_g = \cos^{-1}(R)/\pi$, where $R = \mathbf{J} \cdot \mathbf{T}/\|\mathbf{J}\|_2\|\mathbf{T}\|_2$.

## A.2  Main result and overview

Our main result is a set of self-consistent equations which can be solved to obtain the generalization error $\varepsilon(\alpha, p, \theta)$ for any $\alpha$ and any data distribution $p(z)$ along a probe student at any angle $\theta$ relative to the teacher. These equations take the form,

$$\frac{R - \rho\cos\theta}{\sin^2\theta} = \frac{\alpha}{\pi\Lambda}\left\langle \int_{-\infty}^{\kappa} dt \; \exp\left(-\frac{\Delta(t,z)}{2\Lambda^2}\right)(\kappa - t)\right\rangle_z \tag{1}$$

$$1 - \frac{\rho^2 + R^2 - 2\rho R\cos\theta}{\sin^2\theta} = 2\alpha\left\langle \int_{-\infty}^{\kappa} dt \frac{e^{-\frac{(t-\rho z)^2}{2(1-\rho^2)}}}{\sqrt{2\pi}\sqrt{1-\rho^2}} H\left(\frac{\Gamma(t,z)}{\sqrt{1-\rho^2}\Lambda}\right)(\kappa - t)^2\right\rangle_z \tag{2}$$

$$\frac{\rho - R\cos\theta}{\sin^2\theta} = 2\alpha\left\langle \int_{-\infty}^{\kappa} dt \frac{e^{-\frac{(t-\rho z)^2}{2(1-\rho^2)}}}{\sqrt{2\pi}\sqrt{1-\rho^2}} H\left(\frac{\Gamma(t,z)}{\sqrt{1-\rho^2}\Lambda}\right)\left(\frac{z - \rho t}{1-\rho^2}\right)(\kappa - t)\right.$$
$$\left. + \frac{1}{2\pi\Lambda}\exp\left(-\frac{\Delta(t,z)}{2\Lambda^2}\right)\left(\frac{\rho R - \cos\theta}{1-\rho^2}\right)(\kappa - t)\right\rangle_z \tag{3}$$

Where,

$$\Lambda = \sqrt{\sin^2\theta - R^2 - \rho^2 + 2\rho R\cos\theta}, \tag{4}$$

$$\Gamma(t,z) = z(\rho R - \cos\theta) - t(R - \rho\cos\theta), \tag{5}$$

$$\Delta(t,z) = z^2\left(\rho^2 + \cos^2\theta - 2\rho R\cos\theta\right) + 2tz(R\cos\theta - \rho) + t^2\sin^2\theta. \tag{6}$$

Where $\langle\cdot\rangle_z$ represents an average over the pruned data distribution $p(z)$ along the probe student. For any $\alpha, p(z), \theta$, these equations can be solved for the order parameters $R, \rho, \kappa$, from which the generalization error can be easily read off as $\varepsilon_g = \cos^{-1}(R)/\pi$. This calculation results in the solid theory curves in Figs 1,2,3, which show an excellent match to numerical simulations. In the following section we will walk through the derivation of these equations using replica theory. In Section A.6 we will derive an expression for the information gained per training example, and show that with Pareto optimal data pruning this information gain can be made to converge to a finite rate, resulting in at least exponential decay in test error. In Section A.7, we will show that super-exponential scaling eventually breaks down when the probe student does not match the teacher perfectly, resulting in power law scaling at at a minimum pruning fraction $f_{\min}(\theta)$.

## A.3  Replica calculation of the generalization error

To obtain Eqs. 1,2,3, we follow the approach of Elizabeth Gardner and compute the volume $\Omega(\mathbf{x}^\mu, \mathbf{T}, \kappa)$ of solutions $J$ which perfectly classify the training data up to a margin $\kappa$ (known as the Gardner volume) [29, 25]. As $\kappa$ grows, the volume of solutions shrinks until it reaches a unique solution at a critical $\kappa$, the max-margin solution. The Gardner volume $\Omega$ takes the form,

$$\Omega(\mathbf{x}^\mu, \mathbf{T}, \kappa) = \int d\mu(\mathbf{J}) \prod_\mu \Theta\left(\frac{\mathbf{T} \cdot \mathbf{x}^\mu}{\sqrt{N}}\left(\frac{\mathbf{J} \cdot \mathbf{x}^\mu}{\sqrt{N}} - \kappa\right)\right) \tag{7}$$

Because the student's decision boundary is invariant to an overall scaling of $\mathbf{J}$, we enforce normalization of $\mathbf{J}$ via the measure $d\mu(\mathbf{J})$,

$$d\mu(\mathbf{J}) = \frac{1}{(2\pi e)^{N/2}}\delta(\|\mathbf{J}\|^2 - N) \tag{8}$$

In the thermodynamic limit $N, P \to \infty$ the typical value of the entropy $S(\kappa) = \langle\langle \log \Omega(\mathbf{x}^\mu, \mathbf{T}, \kappa)\rangle\rangle$ is dominated by particular values of $R, \kappa$, where the double angle brackets $\langle\langle \cdot \rangle\rangle$ denote a quenched average over disorder introduced by random realizations of the training examples $\mathbf{x}^\mu$ and the teacher $\mathbf{T}$. However, computing this quenched average is intractable since the integral over $\mathbf{J}$ cannot be performed analytically for every individual realization of the examples. We rely on the replica trick from statistical physics,

$$\ln(x) = \lim_{n \to 0} \frac{x^n - 1}{n} \tag{9}$$

Which allows us to evaluate $S(\kappa)$ in terms of easier-to-compute powers of $\Omega$,

$$S(\kappa) = \langle\langle \ln \Omega(\mathbf{x}^\mu, \mathbf{T}, \kappa)\rangle\rangle = \frac{\langle\langle \Omega^n(\mathbf{x}^\mu, \mathbf{T}, \kappa)\rangle\rangle - 1}{n} \tag{10}$$

This reduces our problem to computing powers of $\Omega$, which for integer $n$ can be written in terms of $\alpha = 1, \ldots, n$ replicated copies of the original system,

$$\Omega^{(n)} \equiv \langle\langle \Omega^n(\mathbf{x}^\mu, \mathbf{T}, \kappa)\rangle\rangle = \left\langle\!\left\langle \int \prod_{\alpha=1}^n d\mu(\mathbf{J}^\alpha) \prod_{\alpha,\mu} \Theta\left(\frac{\mathbf{T} \cdot \mathbf{x}^\mu}{\sqrt{N}}\left(\frac{\mathbf{J} \cdot \mathbf{x}^\mu}{\sqrt{N}} - \kappa\right)\right) \right\rangle\!\right\rangle \tag{11}$$

We begin by introducing auxiliary variables,

$$\lambda_\mu^\alpha = \frac{\mathbf{J}^\alpha \cdot \mathbf{x}^\mu}{\sqrt{N}}, \quad u_\mu = \frac{\mathbf{T} \cdot \mathbf{x}^\mu}{\sqrt{N}} \tag{12}$$

by $\delta-$functions, to pull the dependence on $\mathbf{J}$ and $\mathbf{T}$ outside of the Heaviside function,

$$\Omega^{(n)} = \int \prod_{\alpha=1}^n d\mu(\mathbf{J}^\alpha) \int \prod_{\alpha,\mu} d\lambda_\mu^\alpha \int \prod_\mu du_\mu \prod_{\alpha,\mu} \Theta\big(u_\mu(\lambda_\mu^\alpha - \kappa)\big)$$
$$\times \left\langle\!\left\langle \delta\left(\lambda_\mu^\alpha - \frac{1}{\sqrt{N}}\mathbf{J}^\alpha \cdot \mathbf{x}^\mu\right)\delta\left(u_\mu - \frac{1}{\sqrt{N}}\mathbf{T} \cdot \mathbf{x}^\mu\right) \right\rangle\!\right\rangle \tag{13}$$

Using the integral representation of the $\delta$-functions,

$$\Omega^{(n)} = \int \prod_{\alpha=1}^n d\mu(\mathbf{J}^\alpha) \int \prod_{\alpha,\mu} \frac{d\lambda_\mu^\alpha d\hat{\lambda}_\mu^\alpha}{2\pi} \int \prod_\mu \frac{du_\mu d\hat{u}_\mu}{2\pi} \tag{14}$$
$$\times \prod_{\alpha,\mu} \Theta\big(u_\mu(\lambda_\mu^\alpha - \kappa)\big) \exp\left(i \sum_{\mu,\alpha} \lambda_\mu^\alpha \hat{\lambda}_\mu^\alpha + i \sum_\mu u_\mu \hat{u}_\mu\right) \tag{15}$$
$$\times \left\langle\!\left\langle \exp\left(-\frac{i}{\sqrt{N}}\sum_{\mu,\alpha}\hat{\lambda}_\mu^\alpha \mathbf{J}^\alpha \cdot \mathbf{x}^\mu - \frac{i}{\sqrt{N}}\sum_\mu \hat{u}_\mu \mathbf{T} \cdot \mathbf{x}^\mu\right)\right\rangle\!\right\rangle \tag{16}$$

The data obeys some distribution $p(z)$ along the direction of $\mathbf{J}_{\text{probe}}$ and is isotropic in the nullspace of $\mathbf{J}_{\text{probe}}$. Hence we can decompose a training example $\mathbf{x}^\mu$ as follows, $\mathbf{x}^\mu = \mathbf{J}_{\text{probe}}z^\mu + (I - \mathbf{J}_{\text{probe}}\mathbf{J}_{\text{probe}}^T)\mathbf{s}^\mu$, where $z^\mu \sim p(z)$ and $\mathbf{s}^\mu \sim \mathcal{N}(0, I_N)$,

$$\Omega^{(n)} = \int \prod_{\alpha=1}^n d\mu(\mathbf{J}^\alpha) \int \prod_{\alpha,\mu} \frac{d\lambda_\mu^\alpha d\hat{\lambda}_\mu^\alpha}{2\pi} \int \prod_\mu \frac{du_\mu d\hat{u}_\mu}{2\pi} \tag{17}$$

$$\times \prod_{\alpha,\mu} \Theta\big(u_\mu(\lambda_\mu^\alpha - \kappa)\big) \exp\left(i\sum_{\mu,\alpha} \lambda_\mu^\alpha \hat{\lambda}_\mu^\alpha + i\sum_\mu u_\mu \hat{u}_\mu\right) \tag{18}$$

$$\times \left\langle\!\!\left\langle \exp\left(-\frac{i}{\sqrt{N}}\sum_{\mu,\alpha} \hat{\lambda}_\mu^\alpha(\mathbf{J}^\alpha \cdot \mathbf{J}_{\text{probe}}z^\mu + \mathbf{J}_\perp^\alpha \cdot \mathbf{s}^\mu) - \frac{i}{\sqrt{N}}\sum_\mu \hat{u}_\mu(\mathbf{T}\cdot\mathbf{J}_{\text{probe}}z^\mu + \mathbf{T}_\perp \cdot \mathbf{s}^\mu)\right)\right\rangle\!\!\right\rangle \tag{19}$$

Where $\mathbf{J}_\perp = (1 - \mathbf{J}_{\text{probe}}\mathbf{J}_{\text{probe}}^T)\mathbf{J}$ and $\mathbf{T}_\perp = (1 - \mathbf{J}_{\text{probe}}\mathbf{J}_{\text{probe}}^T)\mathbf{T}$. Now we can average over the patterns $\mathbf{s}^\mu \sim \mathcal{N}(0, I_N)$,

$$\left\langle \exp\left(-\frac{i}{\sqrt{N}}\sum_{\mu,\alpha}\hat{\lambda}_\mu^\alpha \mathbf{J}_\perp^\alpha \cdot \mathbf{s}^\mu - \frac{i}{\sqrt{N}}\sum_\mu \hat{u}_\mu \mathbf{T}_\perp \cdot \mathbf{s}^\mu\right)\right\rangle_{s^\mu} = \exp\left(-\frac{1}{2N}\|\sum_{\mu,\alpha}\hat{\lambda}_\mu^\alpha\mathbf{J}_\perp^\alpha + \hat{u}_\mu\mathbf{T}_\perp\|^2\right) \tag{20}$$

$$= \exp\left(-\frac{1}{2N}\sum_\mu\left(\sum_{\alpha\beta}\hat{\lambda}_\mu^\alpha\hat{\lambda}_\mu^\beta\mathbf{J}_\perp^\alpha\cdot\mathbf{J}_\perp^\beta + 2\sum_\alpha\hat{\lambda}_\mu^\alpha\hat{u}_\mu\mathbf{J}_\perp^\alpha\cdot\mathbf{T}_\perp + \hat{u}_\mu^2\|\mathbf{T}_\perp\|^2\right)\right). \tag{21}$$

Inserting this back into our expression for the Gardner volume,

$$\begin{aligned}\Omega^{(n)} = &\int \prod_{\alpha=1}^n d\mu(\mathbf{J}^\alpha) \int \prod_{\alpha,\mu} \frac{d\lambda_\mu^\alpha d\hat{\lambda}_\mu^\alpha}{2\pi} \int \prod_\mu \frac{du_\mu d\hat{u}_\mu}{2\pi} \\ &\times \prod_{\alpha,\mu} \Theta\big(u_\mu(\lambda_\mu^\alpha - \kappa)\big) \exp\left(i\sum_{\mu,\alpha} \lambda_\mu^\alpha \hat{\lambda}_\mu^\alpha + i\sum_\mu u_\mu \hat{u}_\mu\right) \\ &\times \left\langle\!\!\left\langle \exp\left[-\frac{1}{2N}\sum_\mu\left(\sum_{\alpha\beta}\hat{\lambda}_\mu^\alpha\hat{\lambda}_\mu^\beta\mathbf{J}_\perp^\alpha\cdot\mathbf{J}_\perp^\beta + 2\sum_\alpha\hat{\lambda}_\mu^\alpha\hat{u}_\mu\mathbf{J}_\perp^\alpha\cdot\mathbf{T}_\perp + \hat{u}_\mu^2\|\mathbf{T}_\perp\|^2\right)\right.\right. \\ &\left.\left.- i\sum_\mu\left(\sum_\alpha\hat{\lambda}_\mu^\alpha\mathbf{J}^\alpha\cdot\mathbf{J}_{\text{probe}} + \hat{u}_\mu\mathbf{T}\cdot\mathbf{J}_{\text{probe}}\right)z^\mu\right]\right\rangle\!\!\right\rangle_{T,z^\mu}\end{aligned} \tag{22}$$

As is typical in replica calculations of this type, we now introduce order parameters,

$$q^{\alpha\beta} = \frac{\mathbf{J}^\alpha \cdot \mathbf{J}^\beta}{N}, \quad R^\alpha = \frac{\mathbf{T}\cdot\mathbf{J}^\alpha}{N} \tag{23}$$

which will allow us to decouple the $\mathbf{J}$- from the $\lambda$-$\mu$-$z$- integrals. $q^{\alpha\beta}$ represents the overlaps between replicated students, and $R^\alpha$ the overlap between each replicated student and the teacher. However, because our problem involves the additional role of the probe student, we must introduce an additional order parameter,

$$\rho^\alpha = \frac{\mathbf{J}^\alpha \cdot \mathbf{J}_{\text{probe}}}{N} \tag{24}$$

which represents the overlap between each replicated student and the probe student. Notice that,

$$\mathbf{J}^\alpha_\perp \cdot \mathbf{J}^\beta_\perp = \mathbf{J}^\alpha \cdot \mathbf{J}^\beta - \mathbf{J}^\alpha_\parallel \cdot \mathbf{J}^\beta_\parallel = N(q^{\alpha\beta} - \rho^\alpha \rho^\beta) \tag{25}$$

$$\mathbf{J}^\alpha_\perp \cdot \mathbf{T}_\perp = \mathbf{J}^\alpha \cdot \mathbf{T} - \mathbf{J}^\alpha_\parallel \cdot \mathbf{T}_\parallel = N(R^\alpha - \rho^\alpha \cos\theta) \tag{26}$$

With this new set of order parameters in hand, we can decouple the $\mathbf{J}$ from the $\lambda - u - z-$integrals.

$$
\begin{aligned}
\Omega^{(n)} =& \int \prod_{\alpha<\beta} dq^{\alpha\beta} \int \prod_\alpha dR^\alpha \int \prod_\alpha d\rho^\alpha \\
&\times \int \prod_{\alpha=1}^n d\mu(\mathbf{J}^\alpha) \Big\langle \prod_\alpha \delta(\mathbf{T}\cdot\mathbf{J}^\alpha - NR^\alpha) \Big\rangle_{\mathbf{T}} \prod_{\alpha<\beta} \delta(\mathbf{J}^\alpha\cdot\mathbf{J}^\beta - Nq^{\alpha\beta}) \prod_\alpha \delta(\mathbf{J}^\alpha\cdot\mathbf{J}_{\text{probe}} - N\rho^\alpha) \\
&\times \int \prod_{\alpha,\mu} \frac{d\lambda^\alpha_\mu d\hat{\lambda}^\alpha_\mu}{2\pi} \int \prod_\mu \frac{du_\mu d\hat{u}_\mu}{2\pi} \prod_{\alpha,\mu} \Theta\big(u_\mu(\lambda^\alpha_\mu - \kappa)\big) \prod_\mu \exp\Big(i\sum_\alpha \lambda^\alpha_\mu \hat{\lambda}^\alpha_\mu + iu_\mu \hat{u}_\mu\Big) \\
&\times \Big\langle \exp\Big[ -\frac{1}{2}\sum_{\alpha\beta} \hat{\lambda}^\alpha_\mu \hat{\lambda}^\beta_\mu (q^{\alpha\beta} - \rho^\alpha\rho^\beta) - \sum_\alpha \hat{\lambda}^\alpha_\mu \hat{u}_\mu (R^\alpha - \rho^\alpha\cos\theta) - \frac{1}{2}\hat{u}^2_\mu \sin^2\theta \\
&- i\Big(\sum_\alpha \hat{\lambda}^\alpha_\mu \rho^\alpha + \hat{u}_\mu \cos\theta\Big) z^\mu \Big] \Big\rangle_{z^\mu}
\end{aligned}
\tag{27}
$$

We can now perform the gaussian integral over $\hat{u}_\mu$,

$$
\begin{aligned}
\Omega^{(n)} =& \int \prod_{\alpha<\beta} dq^{\alpha\beta} \int \prod_\alpha dR^\alpha \int \prod_\alpha d\rho^\alpha \\
&\times \int \prod_{\alpha=1}^n d\mu(\mathbf{J}^\alpha) \Big\langle \prod_\alpha \delta(\mathbf{T}\cdot\mathbf{J}^\alpha - NR^\alpha) \Big\rangle_{\mathbf{T}} \prod_{\alpha<\beta} \delta(\mathbf{J}^\alpha\cdot\mathbf{J}^\beta - Nq^{\alpha\beta}) \prod_\alpha \delta(\mathbf{J}^\alpha\cdot\mathbf{J}_{\text{probe}} - N\rho^\alpha) \\
&\times \int \prod_{\alpha,\mu} \frac{d\lambda^\alpha_\mu d\hat{\lambda}^\alpha_\mu}{2\pi} \int \prod_\mu \frac{du_\mu d\hat{u}_\mu}{2\pi} \prod_{\alpha,\mu} \Theta\big(u_\mu(\lambda^\alpha_\mu - \kappa)\big) \prod_\mu \exp\Big(i\sum_\alpha \lambda^\alpha_\mu \hat{\lambda}^\alpha_\mu\Big) \\
&\times \Big\langle \exp\Big[ -\frac{1}{2}\sum_{\alpha\beta} \hat{\lambda}^\alpha_\mu \hat{\lambda}^\beta_\mu (q^{\alpha\beta} - \rho^\alpha\rho^\beta) - i\sum_\alpha \hat{\lambda}^\alpha_\mu \rho^\alpha z^\mu \\
&+ \frac{1}{2\sin^2\theta}\Big(i(u_\mu - z^\mu\cos\theta) - \sum_\alpha \hat{\lambda}^\alpha_\mu (R^\alpha - \rho^\alpha\cos\theta)\Big)^2 \Big] \Big\rangle_{z^\mu}
\end{aligned}
\tag{28}
$$

Expanding,

$$\Omega^{(n)} = \int \prod_{\alpha<\beta} dq^{\alpha\beta} \int \prod_\alpha dR^\alpha \int \prod_\alpha d\rho^\alpha$$

$$\times \int \prod_{\alpha=1}^n d\mu(\mathbf{J}^\alpha) \Big\langle \prod_\alpha \delta(\mathbf{T} \cdot \mathbf{J}^\alpha - NR^\alpha) \Big\rangle_{\mathbf{T}} \prod_{\alpha<\beta} \delta(\mathbf{J}^\alpha \cdot \mathbf{J}^\beta - Nq^{\alpha\beta}) \prod_\alpha \delta(\mathbf{J}^\alpha \cdot \mathbf{J}_{\text{probe}} - N\rho^\alpha)$$

$$\times \int \prod_{\alpha,\mu} \frac{d\lambda_\mu^\alpha d\hat{\lambda}_\mu^\alpha}{2\pi} \int \prod_\mu \frac{du_\mu d\hat{u}_\mu}{2\pi} \prod_{\alpha,\mu} \Theta(u_\mu(\lambda_\mu^\alpha - \kappa)) \prod_\mu \exp\Big(i \sum_\alpha \lambda_\mu^\alpha \hat{\lambda}_\mu^\alpha - \frac{1}{2}\frac{(u_\mu - z^\mu \cos\theta)^2}{\sin^2\theta}\Big)$$

$$\times \Big\langle \exp\Big[ -\frac{1}{2}\sum_{\alpha\beta} \hat{\lambda}_\mu^\alpha \hat{\lambda}_\mu^\beta (q^{\alpha\beta} - \rho^\alpha \rho^\beta) - i\sum_\alpha \hat{\lambda}_\mu^\alpha \rho^\alpha z^\mu$$

$$- \frac{i}{\sin^2\theta}(u_\mu - z^\mu \cos\theta)\sum_\alpha \hat{\lambda}_\mu^\alpha(R^\alpha - \rho^\alpha \cos\theta) + \frac{1}{2\sin^2\theta}\sum_{\alpha\beta} \hat{\lambda}_\mu^\alpha \hat{\lambda}_\mu^\beta(R^\alpha - \rho^\alpha \cos\theta)(R^\beta - \rho^\beta \cos\theta)\Big]\Big\rangle_{z^\mu}$$

$$(29)$$

Simplifying,

$$\Omega^{(n)} = \int \prod_{\alpha<\beta} dq^{\alpha\beta} \int \prod_\alpha dR^\alpha \int \prod_\alpha d\rho^\alpha$$

$$\times \int \prod_{\alpha=1}^n d\mu(\mathbf{J}^\alpha) \Big\langle \prod_\alpha \delta(\mathbf{T} \cdot \mathbf{J}^\alpha - NR^\alpha) \Big\rangle_{\mathbf{T}} \prod_{\alpha<\beta} \delta(\mathbf{J}^\alpha \cdot \mathbf{J}^\beta - Nq^{\alpha\beta}) \prod_\alpha \delta(\mathbf{J}^\alpha \cdot \mathbf{J}_{\text{probe}} - N\rho^\alpha)$$

$$\times \int \prod_{\alpha,\mu} \frac{d\lambda_\mu^\alpha d\hat{\lambda}_\mu^\alpha}{2\pi} \int \prod_\mu \frac{du_\mu d\hat{u}_\mu}{2\pi} \prod_{\alpha,\mu} \Theta(u_\mu(\lambda_\mu^\alpha - \kappa)) \exp\Big(i \sum_{\mu,\alpha} \lambda_\mu^\alpha \hat{\lambda}_\mu^\alpha - \frac{1}{2}\frac{(u_\mu - z^\mu \cos\theta)^2}{\sin^2\theta}\Big)$$

$$\times \Big\langle \exp\Big[ -\frac{1}{2}\sum_\mu \sum_{\alpha\beta} \hat{\lambda}_\mu^\alpha \hat{\lambda}_\mu^\beta \Big(q^{\alpha\beta} - \rho^\alpha \rho^\beta - \frac{(R^\alpha - \rho^\alpha \cos\theta)(R^\beta - \rho^\beta \cos\theta)}{\sin^2\theta}\Big) - i\sum_\mu \sum_\alpha \hat{\lambda}_\mu^\alpha \rho^\alpha z^\mu$$

$$- \frac{i}{\sin^2\theta}(u_\mu - z^\mu \cos\theta)\sum_\alpha \hat{\lambda}_\mu^\alpha(R^\alpha - \rho^\alpha \cos\theta)\Big]\Big\rangle_{z^\mu}$$

$$(30)$$

Now we introduce integral representations for the remaining delta functions, including the measure $d\mu(\mathbf{J}^\alpha)$, for which we introduce the parameter $\hat{k}^\alpha$,

$$\Omega^{(n)} = \int \prod_\alpha \frac{d\hat{k}^\alpha}{4\pi} \int \prod_{\alpha<\beta} \frac{dq^{\alpha\beta} d\hat{q}^{\alpha\beta}}{2\pi/N} \int \prod_\alpha \frac{dR^\alpha d\hat{R}^\alpha}{2\pi/N} \int \prod_\alpha \frac{d\rho^\alpha d\hat{\rho}^\alpha}{2\pi/N}$$

$$\times \exp\left( i\frac{N}{2} \sum_\alpha \hat{k}^\alpha + iN \sum_{\alpha<\beta} q^{\alpha\beta} \hat{q}^{\alpha\beta} + iN \sum_\alpha R^\alpha \hat{R}^\alpha + iN \sum_\alpha \rho^\alpha \hat{\rho}^\alpha \right)$$

$$\times \int \prod_{i,\alpha} \frac{dJ_i^\alpha}{\sqrt{2\pi e}} \exp\left( -\frac{i}{2} \sum_\alpha \hat{k}^\alpha \|\mathbf{J}^\alpha\|^2 - i \sum_{\alpha<\beta} \hat{q}^{\alpha\beta} \mathbf{J}^\alpha \cdot \mathbf{J}^\beta - i \sum_\alpha \hat{R}_\alpha \mathbf{J}^\alpha \cdot \mathbf{T} - i \sum_\alpha \hat{\rho}_\alpha \mathbf{J}^\alpha \cdot \mathbf{J}_{\text{probe}} \right)$$

$$\times \int \prod_{\alpha,\mu} \frac{d\lambda_\mu^\alpha d\hat{\lambda}_\mu^\alpha}{2\pi} \int \prod_\mu \frac{du_\mu d\hat{u}_\mu}{2\pi} \prod_{\alpha,\mu} \Theta\big(u_\mu(\lambda_\mu^\alpha - \kappa)\big) \exp\left( i\sum_{\mu,\alpha} \lambda_\mu^\alpha \hat{\lambda}_\mu^\alpha - \frac{1}{2} \frac{(u_\mu - z^\mu \cos\theta)^2}{\sin^2\theta} \right)$$

$$\times \left\langle \exp\left[ -\frac{1}{2} \sum_\mu \sum_{\alpha\beta} \hat{\lambda}_\mu^\alpha \hat{\lambda}_\mu^\beta \left( q^{\alpha\beta} - \rho^\alpha \rho^\beta - \frac{(R^\alpha - \rho^\alpha \cos\theta)(R^\beta - \rho^\beta \cos\theta)}{\sin^2\theta} \right) - i \sum_\mu \sum_\alpha \hat{\lambda}_\mu^\alpha \rho^\alpha z^\mu \right. \right.$$

$$\left. \left. -\frac{i}{\sin^2\theta}\big(u_\mu - z^\mu \cos\theta\big) \sum_\alpha \hat{\lambda}_\mu^\alpha (R^\alpha - \rho^\alpha \cos\theta) \right] \right\rangle_{z^\mu}$$

(31)

Notice that the $u_\mu - \lambda_\mu^\alpha - \hat{\lambda}_\mu^\alpha - z_\mu$-integrals factorize in $\mu$, and can be written as a single integral to the power of $P = \alpha N$.

$$\Omega^{(n)} = k \int \prod_\alpha \frac{d\hat{k}^\alpha}{4\pi} \int \prod_{\alpha<\beta} \frac{dq^{\alpha\beta} d\hat{q}^{\alpha\beta}}{2\pi/N} \int \prod_\alpha \frac{dR^\alpha d\hat{R}^\alpha}{2\pi/N} \int \prod_\alpha \frac{d\rho^\alpha d\hat{\rho}^\alpha}{2\pi/N}$$

$$\times \exp\left( N\left[ \frac{i}{2} \sum_\alpha \hat{k}^\alpha + i \sum_{\alpha<\beta} q^{\alpha\beta} \hat{q}^{\alpha\beta} + i \sum_\alpha R^\alpha \hat{R}^\alpha + i \sum_\alpha \rho^\alpha \hat{\rho}^\alpha \right. \right.$$

(32)

$$\left. \left. + G_S(\hat{k}^\alpha, \hat{q}^{\alpha\beta}, \hat{R}^\alpha, \hat{\rho}^\alpha) + \alpha G_E(q^{\alpha\beta}, R^\alpha, \rho^\alpha) \right] \right)$$

Where we have written the Gardner volume in terms of an *entropic* part $G_S$, which measures how many spherical couplings satisfy the constraints,

$$G_S = \frac{1}{N} \log \int \prod_\alpha \frac{d\mathbf{J}^\alpha}{\sqrt{2\pi e}} \exp\left( -\frac{i}{2} \sum_\alpha \hat{k}^\alpha \|\mathbf{J}^\alpha\|^2 - i \sum_{\alpha<\beta} \hat{q}^{\alpha\beta} \mathbf{J}^\alpha \cdot \mathbf{J}^\beta - i \sum_\alpha \hat{R}^\alpha \mathbf{J}^\alpha \cdot \mathbf{T} - i \sum_\alpha \hat{\rho}^\alpha \mathbf{J}^\alpha \cdot \mathbf{J}_{\text{probe}} \right)$$

(33)

And an *energetic* part $G_E$,

$$G_E = \log \int \frac{du}{\sqrt{2\pi}} \int \prod_\alpha \frac{d\lambda^\alpha d\hat{\lambda}^\alpha}{2\pi} \prod_\alpha \Theta\big(u(\lambda^\alpha - \kappa)\big) \exp\left( i\sum_\alpha \lambda^\alpha \hat{\lambda}^\alpha - \frac{1}{2} \frac{(u_\mu - z^\mu \cos\theta)^2}{\sin^2\theta} \right)$$

$$\times \left\langle \exp\left[ -\frac{1}{2} \sum_{\alpha\beta} \hat{\lambda}^\alpha \hat{\lambda}^\beta \left( q^{\alpha\beta} - \rho^\alpha \rho^\beta - \frac{(R^\alpha - \rho^\alpha \cos\theta)(R^\beta - \rho^\beta \cos\theta)}{\sin^2\theta} \right) - i \sum_\alpha \hat{\lambda}^\alpha \rho^\alpha z \right. \right.$$

$$\left. \left. -\frac{i}{\sin^2\theta}\big(u - z\cos\theta\big) \sum_\alpha \hat{\lambda}^\alpha (R^\alpha - \rho^\alpha \cos\theta) \right] \right\rangle_z$$

(34)

We first evaluate the entropic part, $G_S$, by introducing the $n \times n$ matrices $A, B$,

$$A_{\alpha\beta} = i\hat{k}^\alpha \delta_{\alpha\beta} + i\hat{q}^{\alpha\beta}(1 - \delta_{\alpha\beta})$$ (35)

$$B_{\alpha\beta} = \delta_{\alpha\beta} + q^{\alpha\beta}(1 - \delta_{\alpha\beta})$$ (36)

Inserting this our expression for $G_S$ becomes

$$G_S = \frac{1}{N} \log \int \prod_\alpha \frac{d\mathbf{J}^\alpha}{\sqrt{2\pi e}} \exp\left(-\frac{1}{2}\sum_{\alpha,\beta} \mathbf{J}^{\alpha T} A_{\alpha\beta} \mathbf{J}^\beta - i\sum_\alpha \mathbf{J}^\alpha \cdot (\mathbf{T}\hat{R}^\alpha + \mathbf{J}_{\text{probe}}\hat{\rho}^\alpha)\right) \tag{37}$$

Integrating over $\mathbf{J}^\alpha$,

$$G_S = -\frac{n}{2} - \frac{1}{2}\log(\det A) - \frac{1}{2N}\sum_{\alpha,\beta}(\mathbf{T}\hat{R}^\alpha + \mathbf{J}_{\text{probe}}\hat{\rho}^\alpha)^T A_{\alpha\beta}^{-1}(\mathbf{T}\hat{R}^\beta + \mathbf{J}_{\text{probe}}\hat{\rho}^\beta) \tag{38}$$

Now we can include the remaining terms in the expression for $\Omega^{(n)}$ outside of $G_E$ and $G_S$ by noting that

$$tr(AB) = \sum_{\alpha\beta} A_{\alpha\beta} B_{\beta\alpha} \tag{39}$$

$$= \sum_{\alpha\beta}(i\hat{k}^\alpha \delta_{\alpha\beta} + i\hat{q}^{\alpha\beta}(1 - \delta_{\alpha\beta}))(\delta_{\alpha\beta} + q^{\alpha\beta}(1 - \delta_{\alpha\beta})) \tag{40}$$

$$= \sum_\alpha i\hat{k}^\alpha + 2\sum_{\alpha<\beta} iq^{\alpha\beta}\hat{q}^{\alpha\beta} \tag{41}$$

Additionally, we can use $\log \det A = tr(\log A)$. Thus all terms in the exponent except $G_E$ can be written as

$$-\frac{n}{2} - \frac{1}{2}tr(\log A) - \frac{1}{2N}\sum_{\alpha,\beta}(\mathbf{T}\hat{R}^\alpha + \mathbf{J}_{\text{probe}}\hat{\rho}^\alpha)^T A_{\alpha\beta}^{-1}(\mathbf{T}\hat{R}^\beta + \mathbf{J}_{\text{probe}}\hat{\rho}^\beta) + \frac{1}{2}tr(AB) + i\sum_\alpha R^\alpha \hat{R}^\alpha + i\sum_\alpha \rho^\alpha \hat{\rho}^\alpha \mathbf{J}_{\text{probe}} \tag{42}$$

Now we extremize wrt $\hat{R}^\alpha$ and the elements of $A$ by setting the derivatives wrt $\hat{R}^\gamma$, $\hat{\rho}^\gamma$ and $A^{\gamma\delta}$ equal to zero:

$$0 = -\sum_\alpha A_{\alpha\gamma}^{-1}\mathbf{T} \cdot (\mathbf{T}\hat{R}^\alpha + \mathbf{J}_{\text{probe}}\hat{\rho}^\alpha) + iR^\gamma = -\sum_\alpha A_{\alpha\gamma}^{-1}(\hat{R}^\alpha + \hat{\rho}^\alpha \cos\theta) + iR^\gamma \tag{43}$$

$$0 = -\sum_\alpha A_{\alpha\gamma}^{-1}\mathbf{J}_{\text{probe}} \cdot (\mathbf{T}\hat{R}^\alpha + \mathbf{J}_{\text{probe}}\hat{\rho}^\alpha) + i\rho^\gamma = -\sum_\alpha A_{\alpha\gamma}^{-1}(\hat{R}^\alpha \cos\theta + \hat{\rho}^\alpha) + i\rho^\gamma \tag{44}$$

$$0 = -\frac{1}{2}A_{\gamma\delta}^{-1} + \frac{1}{2}\sum_{\alpha,\beta}(\mathbf{T}\hat{R}^\alpha + \mathbf{J}_{\text{probe}}\hat{\rho}^\alpha)^T A_{\alpha\gamma}^{-1} A_{\beta\delta}^{-1}(\mathbf{T}\hat{R}^\beta + \mathbf{J}_{\text{probe}}\hat{\rho}^\beta) + \frac{1}{2}B_{\gamma\delta} \tag{45}$$

Solving these gives

$$\hat{R}^\alpha = i\sum_\beta A_{\alpha\beta}\frac{R^\beta - \rho^\beta \cos\theta}{\sin^2\theta} \tag{46}$$

$$\hat{\rho}^\alpha = i\sum_\beta A_{\alpha\beta}\frac{\rho^\beta - R^\beta \cos\theta}{\sin^2\theta} \tag{47}$$

and

$$A_{\gamma\delta}^{-1} = B_{\gamma\delta} - \frac{R^\gamma R^\delta - R^\gamma \rho^\delta \cos\theta - R^\delta \rho^\gamma \cos\theta + \rho^\gamma \rho^\delta}{\sin^2\theta} \equiv C_{\gamma\delta} \tag{48}$$

and now we are left with

$$\Omega^{(n)} \sim \exp\left(N\text{extr}_{q^{\alpha\beta},R^\alpha,\rho^\alpha}\left[\frac{1}{2}tr(\log C) + \alpha G_E(q^{\alpha\beta}, R^\alpha)\right]\right) \tag{49}$$

### A.3.1 Replica symmetry ansatz

In order to extremize wrt $q^{\alpha\beta}, R^\alpha, \rho^\alpha$, we take the replica symmetry ansatz [25],

$$q^{\alpha\beta} = q, \quad R^\alpha = R, \quad \rho^\alpha = \rho \tag{50}$$

Then $C$ takes the form

$$C_{\alpha\beta} = \delta_{\alpha\beta} - \frac{R^2 - 2R\rho\cos\theta + \rho^2}{\sin^2\theta} + q(1 - \delta_{\alpha\beta}) \tag{51}$$

A matrix with $E$ on the diagonal and $F$ elsewhere, $C_{\alpha\beta} = E\delta_{\alpha\beta} + F(1 - \delta_{\alpha\beta})$, has $n-1$ degenerate eigenvalues $E - F$ and one eigenvalue $E + (n-1)F$. Hence in our case $C$ has $n-1$ degenerate eigenvalues

$$\left(1 - \frac{R^2 - 2R\rho\cos\theta + \rho^2}{\sin^2\theta}\right) - \left(q - \frac{R^2 - 2R\rho\cos\theta + \rho^2}{\sin^2\theta}\right) = 1 - q \tag{52}$$

and one other eigenvalue,

$$\left(1 - \frac{R^2 - 2R\rho\cos\theta + \rho^2}{\sin^2\theta}\right) + (n-1)\left(q - \frac{R^2 - 2R\rho\cos\theta + \rho^2}{\sin^2\theta}\right) = 1 - q + n\left(q - \frac{R^2 - 2R\rho\cos\theta + \rho^2}{\sin^2\theta}\right) \tag{53}$$

Therefore,

$$\begin{aligned}
tr(\log C) &= (n-1)\log\left(1 - q\right) + \log\left[1 - q + n\left(q - \frac{R^2 - 2R\rho\cos\theta + \rho^2}{\sin^2\theta}\right)\right] \\
&= n\log\left(1 - q\right) + \log\left[1 + n\left(\frac{q\sin^2\theta - (R^2 - 2R\rho\cos\theta + \rho^2)}{(1-q)\sin^2\theta}\right)\right]
\end{aligned} \tag{54}$$

We next evaluate the energetic part, $G_E$,

$$\begin{aligned}
G_E = \log \int \frac{du}{\sqrt{2\pi}} \int \prod_\alpha \frac{d\lambda^\alpha d\hat\lambda^\alpha}{2\pi} \prod_\alpha \Theta\left(u(\lambda^\alpha - \kappa)\right) \exp\left(i\sum_\alpha \lambda^\alpha \hat\lambda^\alpha - \frac{1}{2}\frac{(u - z\cos\theta)^2}{\sin^2\theta}\right) \\
\times \left\langle \exp\left[-\frac{1}{2}\sum_\alpha (\hat\lambda^\alpha)^2\left(1 - \rho^2 - \frac{(R - \rho\cos\theta)^2}{\sin^2\theta}\right) - \frac{1}{2}\sum_{\alpha\neq\beta} \hat\lambda^\alpha \hat\lambda^\beta\left(q - \rho^2 - \frac{(R - \rho\cos\theta)^2}{\sin^2\theta}\right)\right. \right. \\
\left.\left. - i\sum_\alpha \hat\lambda^\alpha \rho^\alpha z - \frac{i}{\sin^2\theta}(u - z\cos\theta)\sum_\alpha \hat\lambda^\alpha(R^\alpha - \rho^\alpha\cos\theta)\right]\right\rangle_z
\end{aligned} \tag{55}$$

First note that we can rewrite the terms

$$\begin{aligned}
&-\frac{1}{2}\sum_\alpha\left(1 - \rho^2 - \frac{(R - \rho\cos\theta)}{\sin^2\theta}\right)(\hat\lambda^\alpha)^2 - \frac{1}{2}\sum_{\alpha\neq\beta}\hat\lambda^\alpha\hat\lambda^\beta\left(q - \rho^2 - \frac{(R - \rho\cos\theta)^2}{\sin^2\theta}\right) \\
&= -\frac{1}{2}\sum_\alpha\left(1 - \rho^2 - \frac{(R - \rho\cos\theta)^2}{\sin^2\theta}\right)(\hat\lambda^\alpha)^2 - \frac{1}{2}\left(q - \rho^2 - \frac{(R - \rho\cos\theta)^2}{\sin^2\theta}\right)\left[(\sum_\alpha \hat\lambda^\alpha)^2 - \sum_\alpha(\hat\lambda^\alpha)^2\right] \\
&= -\frac{1}{2}\sum_\alpha(1 - q)(\hat\lambda^\alpha)^2 - \frac{1}{2}\left(q - \rho^2 - \frac{(R - \rho\cos\theta)^2}{\sin^2\theta}\right)(\sum_\alpha \hat\lambda^\alpha)^2
\end{aligned} \tag{56}$$

To simplify the last term we apply the Hubbard-Stratonovich transformation, $e^{b^2/2} = \int Dt e^{bt}$, introducing auxiliary field $t$,

$$
\begin{aligned}
= \log \int \frac{du}{\sqrt{2\pi}} \int \prod_\alpha \frac{d\lambda^\alpha d\hat{\lambda}^\alpha}{2\pi} \prod_\alpha \Theta(u(\lambda^\alpha - \kappa)) \int Dt \Big\langle \exp\Big[ -\frac{1-q}{2} \sum_\alpha (\hat{\lambda}^\alpha)^2 \\
+ i \sum_\alpha \hat{\lambda}^\alpha \Big( \lambda^\alpha - \rho^\alpha z - \frac{(u - z\cos\theta)(R^\alpha - \rho^\alpha \cos\theta)}{\sin^2\theta} - \sqrt{q - \rho^2 - \frac{(R - \rho\cos\theta)^2}{\sin^2\theta}} t \Big) - \frac{1}{2} \frac{(u - z\cos\theta)^2}{\sin^2\theta} \Big] \Big\rangle_z
\end{aligned}
$$
(57)

Using the $\Theta$-function to restrict the bounds of integration,

$$
\begin{aligned}
= \log 2 \int_0^\infty \frac{du}{\sqrt{2\pi}} \int_\kappa^\infty \prod_\alpha \frac{d\lambda^\alpha}{\sqrt{2\pi}} \int \prod_\alpha \frac{d\hat{\lambda}^\alpha}{\sqrt{2\pi}} \int Dt \Big\langle \exp\Big[ -\frac{1-q}{2} \sum_\alpha (\hat{\lambda}^\alpha)^2 \\
+ i \sum_\alpha \hat{\lambda}^\alpha \Big( \lambda^\alpha - \rho^\alpha z - \frac{(u - z\cos\theta)(R^\alpha - \rho^\alpha \cos\theta)}{\sin^2\theta} - \sqrt{q - \rho^2 - \frac{(R - \rho\cos\theta)^2}{\sin^2\theta}} t \Big) - \frac{1}{2} \frac{(u - z\cos\theta)^2}{\sin^2\theta} \Big] \Big\rangle_z
\end{aligned}
$$
(58)

Now we can perform the gaussian integrals over $\hat{\lambda}^\alpha$,

$$
\begin{aligned}
= \log 2 \int_0^\infty \frac{du}{\sqrt{2\pi}} \int_\kappa^\infty \prod_\alpha \frac{d\lambda^\alpha}{\sqrt{2\pi}} \int Dt \Big\langle \exp\Big[ -\frac{1}{2(1-q)} \Big( \lambda^\alpha - \rho^\alpha z \\
- \frac{(u - z\cos\theta)(R^\alpha - \rho^\alpha \cos\theta)}{\sin^2\theta} - \sqrt{q - \rho^2 - \frac{(R - \rho\cos\theta)^2}{\sin^2\theta}} t \Big)^2 - \frac{1}{2} \frac{(u - z\cos\theta)^2}{\sin^2\theta} \Big] \Big\rangle_z
\end{aligned}
$$
(59)

And $\lambda^\alpha$,

$$
\begin{aligned}
= \log 2 \int_0^\infty \frac{du}{\sqrt{2\pi}} \int Dt \Big\langle H^n\Big[ -\frac{1}{\sqrt{1-q}} \Big( \kappa - \rho^\alpha z + \frac{(u - z\cos\theta)(R^\alpha - \rho^\alpha \cos\theta)}{\sin^2\theta} + \sqrt{q - \rho^2 - \frac{(R - \rho\cos\theta)^2}{\sin^2\theta}} t \Big)^2 \Big] \\
\times \exp\Big[ -\frac{1}{2} \frac{(u - z\cos\theta)^2}{\sin^2\theta} \Big] \Big\rangle_z
\end{aligned}
$$
(60)

Shifting the integration variable $t \to (\rho^\alpha z + \frac{(u - z\cos\theta)(R^\alpha - \rho^\alpha \cos\theta)}{\sin^2\theta} + \sqrt{q - \rho^2 - \frac{(R - \rho\cos\theta)^2}{\sin^2\theta}} t)/\sqrt{q}$, we can finally perform the gaussian integral over $u$,

$$
\begin{aligned}
= \log 2 \int \frac{dt}{\sqrt{2\pi}} \Big\langle \exp\Big( -\frac{(\sqrt{q}t - z\rho)^2}{2(q - \rho)^2} \Big) \sqrt{\frac{q}{q - \rho^2}} H^n\Big( -\sqrt{\frac{q}{1-q}} t \Big) \\
\times H\Big( \frac{1}{\sqrt{q - \rho^2}} \frac{\kappa - (qR_0 z + z\rho(R - 2\rho\cos\theta) - \sqrt{q}t(R - \rho\cos\theta))}{\sqrt{q\sin^2\theta + 2R\rho\cos\theta - R^2 - \rho}} \Big) \Big\rangle_z
\end{aligned}
$$
(61)

We can simplify this further by taking $t \to (\sqrt{q}t - z\rho)/\sqrt{q - \rho^2}$,

$$
= \log 2 \int Dt \Big\langle H^n\Big( \frac{\kappa - (z\rho + \sqrt{q - \rho^2})t}{\sqrt{1-q}} \Big) H\Big( \frac{t\big(\sqrt{q - \rho^2} + \rho z\big)(R - \rho\cos\theta) + qz\cos\theta - \rho Rz}{\sqrt{-(q - \rho^2)(\rho^2 - q\sin^2\theta + R^2 - 2\rho R\cos\theta)}} \Big) \Big\rangle_z
$$
(62)

## A.4 Quenched entropy

Putting everything together, and using the replica identity, $\langle \log \Omega \rangle = \lim_{n \to 0} (\langle \Omega^n \rangle - 1)/n$, we obtain an expression for the quenched entropy of the teacher-student perceptron under data pruning:

$$
\frac{1}{N}\langle \log \Omega \rangle = \mathrm{extr}_{q,R,\rho} \left[ \frac{1}{2} \log \left( 1 - q \right) + \frac{1}{2} \left( \frac{q - (R^2 - 2R\rho \cos\theta + \rho^2)/\sin^2\theta}{1 - q} \right) \right.
$$
$$
+ 2\alpha \left\langle \int Dt \log H \left( \frac{\kappa - (z\rho + \sqrt{q - \rho^2})t}{\sqrt{1 - q}} \right) \right.
$$
$$
\left. \left. \times H \left( \frac{t \left( \sqrt{q - \rho^2} + \rho z \right)(R - \rho \cos\theta) + z(q \cos\theta - \rho R)}{\sqrt{(q - \rho^2)\left( R^2 + \rho^2 - q \sin^2\theta - 2\rho R \cos\theta \right)}} \right) \right\rangle_z \right] \quad (63)
$$

We will now unpack this equation and use it to make predictions in several specific settings.

## A.5 Perfect teacher-probe overlap

We will begin by considering the case where the probe student has learned to perfectly match the teacher, $J_{\mathrm{probe}} = T$, which we can obtain by the limit $\theta \to 0$, $\rho \to R$. In this limit the second $H$-function in Eq. 63 becomes increasingly sharp, approaching a step function:

$$
H \left( \frac{t \left( \sqrt{q - \rho^2} + \rho z \right)(R - \rho \cos\theta) + z(q \cos\theta - \rho R)}{\sqrt{(q - \rho^2)\left( R^2 + \rho^2 - q \sin^2\theta - 2\rho R \cos\theta \right)}} \right) \to \Theta(z) \quad (64)
$$

Hence we are left with,

$$
\frac{1}{N}\langle \langle \ln \Omega(\mathbf{x}^\mu, T, \kappa) \rangle \rangle = \mathrm{extr}_{q,R} \left[ \frac{1}{2} \log \left( 1 - q \right) + \frac{1}{2} \left( \frac{q - R^2}{1 - q} \right) \right.
$$
$$
\left. + 2\alpha \int Dt \int dz p(z) \Theta(z) \log H \left( -\frac{\sqrt{q - R^2}t + Rz - \kappa}{\sqrt{1 - q}} \right) \right] \quad (65)
$$

### A.5.1 Saddle point equations

We can now obtain a set of self-consistent saddle point equations by setting set to zero the derivatives with respect to $R$ and $q$ of the right side of Eq. 65. As $\kappa$ approaches its critical value, the space of solutions shrinks to a unique solution, and hence the overlap between students $q$ approaches one. In the limit $q \to 1$, after some partial integration, we find,

$$
R = 2\alpha \int_{-\infty}^{\kappa} \frac{dt}{\sqrt{2\pi}\sqrt{1 - R^2}} \int_0^\infty dz p(z) \exp \left( -\frac{(t - Rz)^2}{2(1 - R^2)} \right) \left( \frac{z - Rt}{1 - R^2} \right)(\kappa - t) \quad (66)
$$
$$
1 - R^2 = 2\alpha \int_{-\infty}^{\kappa} \frac{dt}{\sqrt{2\pi}\sqrt{1 - R^2}} \int_0^\infty dz p(z) \exp \left( -\frac{(t - Rz)^2}{2(1 - R^2)} \right) (\kappa - t)^2 \quad (67)
$$

These saddle point equations can be solved numerically to find $R$ and $\kappa$ as a function of $\alpha$ for a student perceptron trained on a dataset with an arbitrary distribution along the teacher direction $p(z)$. We can specialize to the case of data pruning by setting $p(z)$ to the distribution found after pruning an initially Gaussian-distributed dataset so that only a fraction $f$ of those examples with the smallest margin along the teacher are kept, $p(z) = \frac{e^{-z^2/2}}{\sqrt{2\pi}f}\Theta(\gamma - |z|)$, where the threshold $\gamma = H^{-1}\left(\frac{1-f}{2}\right)$.

$$R = \frac{2\alpha}{f\sqrt{2\pi}\sqrt{1-R^2}} \int_{-\infty}^{\kappa} Dt \, \exp\left(-\frac{R^2t^2}{2(1-R^2)}\right)\left[1 - \exp\left(-\frac{\gamma(\gamma - 2Rt)}{2(1-R^2)}\right)\right](\kappa - t) \quad (68)$$

$$1 - R^2 = \frac{2\alpha}{f} \int_{-\infty}^{\kappa} Dt \left[H\left(-\frac{Rt}{\sqrt{1-R^2}}\right) - H\left(-\frac{Rt - \gamma}{\sqrt{1-R^2}}\right)\right](\kappa - t)^2 \quad (69)$$

Solving these saddle point equations numerically for $R$ and $\kappa$ yields an excellent fit to numerical simulations, as can be seen in Fig. 1**A**. It is also easy to verify that in the limit of no data pruning ($f \to 1, \gamma \to \infty$) we recover the saddle point equations for the classical teacher-student perceptron (Eqs. 4.4 and 4.5 in [25]),

$$R = \frac{2\alpha}{\sqrt{2\pi}\sqrt{1-R^2}} \int Dt \exp\left(-\frac{R^2t^2}{2(1-R^2)}\right)(\kappa - t) \quad (70)$$

$$1 - R^2 = 2\alpha \int Dt \, H\left(-\frac{Rt}{\sqrt{1-R^2}}\right)(\kappa - t)^2 \quad (71)$$

### A.6   Information gain per example

Why does data pruning allow for super-exponential performance with dataset size $\alpha$? We can define the amount of information gained from each new example, $I(\alpha)$, as the fraction by which the space of solutions which perfectly classify the data is reduced when a new training example is added, $I(\alpha) = \Omega(\frac{P+1}{N})/\Omega(\frac{P}{N})$. Or, equivalently, the rate at which the entropy is reduced, $I(\alpha) = -\frac{d}{d\alpha}S(\alpha)$. Of coure, the volume of solutions shrinks to zero at the max-margin solution; so to study the volume of solutions which perfectly classify the data we simply set the margin to zero $\kappa = 0$. In [31] the information gain for a perceptron in the classical teacher-student setting is shown to take the form,

$$I(\alpha) = -2 \int Dt \, H\left(\sqrt{\frac{R}{1-R}}t\right) \log H\left(\sqrt{\frac{R}{1-R}}t\right) \quad (72)$$

Which goes to zero in the limit of large $\alpha$ as $I(\alpha) \sim 1/\alpha$. Data pruning can increase the information gained per example by pruning away the uninformative examples. To show this, we generalize the calculation of the information gain to pruned datasets, using the expression for the entropy we obtained in the previous section (Eq. 65).

$$S(\alpha) = \frac{1}{N}\langle \log \Omega \rangle = \mathrm{extr}_{q,R}\left[\frac{1}{2}\log\left(1-R\right) + \frac{1}{2}R + 2\alpha \int Dt \int_0^\infty dz \, p(z) \log H\left(-\sqrt{R}t - \frac{R}{\sqrt{1-R}}z\right)\right] \quad (73)$$

Hence the information gain $I(\alpha) = -\frac{d}{d\alpha}S(\alpha)$ is given by

$$I(\alpha) = 2\alpha \int Dt \int dz \, p(z)\Theta(z) \log H\left(-\sqrt{R}t - \frac{R}{\sqrt{1-R}}z\right) \quad (74)$$

Changing variables to $t \to -(\sqrt{R}t + \frac{R}{\sqrt{1-R}}z)/\sqrt{q}$,

$$I(\alpha) = 2\alpha \int Dt \int_0^\infty dz \, p(z)\frac{1}{\sqrt{1-R}}\frac{1}{\sqrt{2\pi}}\exp\left(-\frac{z^2 + 2\sqrt{R}tz}{2(1-R)}\right)\log H\left(\sqrt{\frac{R}{1-R}}t\right) \quad (75)$$

Now, assuming that we prune to a fraction $f$, so that $p(z) = \Theta(|z| - \gamma)\frac{\exp(-z^2/2)}{\sqrt{2\pi}f}$, where $\gamma = H^{-1}\left(\frac{1-f}{2}\right)$

$$I(\alpha) = \frac{2\alpha}{f} \int Dt \left[ H\left( \sqrt{\frac{R}{1-R}} t \right) - H\left( \frac{\gamma + \sqrt{R}t}{\sqrt{1-R}} \right) \right] \log H\left( \sqrt{\frac{R}{1-R}} t \right) \qquad (76)$$

$I(\alpha)$ is plotted for varying values of $f$ in Fig. 1**F**. Notice that for $f \to 1$, $\gamma \to \infty$ and we recover Eq. 72. To obtain the optimal pruning fraction $f_{\text{opt}}$ for any $\alpha$, we first need an equation for $R$, which can be obtained by taking the saddle point of Eq. 73. Next we optimize $I(\alpha)$ by setting the derivative of Eq. 76 with respect to $f$ equal to zero. This gives us a pair of equations which can be solved numerically to obtain $f_{\text{opt}}$ for any $\alpha$.

Finally, Eq. 76 reveals that as we prune more aggressively the information gain per example approaches a finite rate. As $f \to 0$, $\gamma \to 0$, and we obtain,

$$I(\alpha) = - \int Dt \ \log H(\sqrt{R}t) \qquad (77)$$

Which allows us to produce to trace the Pareto frontier in Fig. 1**F**. For $R \to 1$, Eq. 77 gives the asymptotic information gain $I(\infty) = 1$ nat/example.

### A.7 Imperfect teacher-probe overlap

In realistic settings we expect the probe student to have only partial information about the target function. What happens if the probe student does not perfectly match the teacher? To understand this carefully, we need to compute the full set of saddle point equations over $R$, $q$, and $\rho$, which we will do in the following section. But to first get an idea for what goes wrong, we include in this section a simple sketch which reveals the limiting behavior.

Consider the case where the angle between the probe student and teacher is $\theta$. Rotate coordinates so that the first canonical basis vector aligns with the student $J = (1, 0, \ldots, 0)$, and the teacher lies in the span of the first two canonical basis vectors, $T = (\cos\theta, \sin\theta, 0, \ldots, 0)$. Consider the margin along the teacher of a new training example $x$ drawn from the pruned distribution.

$$\mathbb{E}|T \cdot x|^2 = \mathbb{E}[x_0^2 \cos^2\theta + x_1^2 \sin^2\theta] \qquad (78)$$

As the fraction of examples kept goes to zero, $\mathbb{E}x_0^2 \to 0$, and the average margin of a new example converges to a fixed value,

$$\mathbb{E}|T \cdot x|^2 = \sin^2\theta \qquad (79)$$

Hence the data ultimately stops concentrating around the teacher's decision boundary, and the information gained from each new example goes to zero. Therefore we expect the generalization error to converge to a power law, where the constant prefactor is roughly that of pruning with a prune fraction $f_{\text{min}}$ which yields an average margin of $1 - R^2$. This "minimum" pruning fraction lower bounds the generalization error envelope (see Fig. 2), and satisfies the following equation,

$$\int_{-\gamma_{\text{min}}}^{\gamma_{\text{min}}} dx p(x) x^2 = \frac{1}{2} - \frac{e^{-\gamma_{\text{min}}^2/2} \gamma_{\text{min}}}{\sqrt{2\pi}(1 - 2H(\gamma_{\text{min}}))} = 1 - R^2 \qquad (80)$$

where $\gamma_{\text{min}} = H^{-1}\left( \frac{1 - f_{\text{min}}}{2} \right)$. Eq. 80 can be solved numerically, and we use it to produce the lower-bounding power laws shown in red in Fig. 2**C,D**. The minimum achievable pruning fraction $f_{\text{min}}(\theta)$ approaches zero as the angle between the probe student and the teacher shrinks, and we can obtain its scaling by taking $R \to 1$, in which case we find,

$$f_{\text{min}}(\theta) \sim \theta \qquad (81)$$

## A.8 Optimal pruning policy

The saddle point equations Eq. 66,67 reveal that the optimal pruning policy varies as a function of $\alpha_{\text{prune}}$. For $\alpha_{\text{prune}}$ large the best policy is to retain only the "hardest" (smallest-margin) examples. But when $\alpha_{\text{prune}}$ is small, keeping the "hardest" examples performs worse than chance, suggesting that the best policy in the $\alpha_{\text{prune}}$ small regime is to keep the easiest examples. Indeed by switching between the "keep easy" and "keep hard" strategies as $\alpha_{\text{prune}}$ grows, one can achieve a lower Pareto frontier than the one shown in Fig. 1A in the small $\alpha_{\text{prune}}$ regime (Fig. 7C).

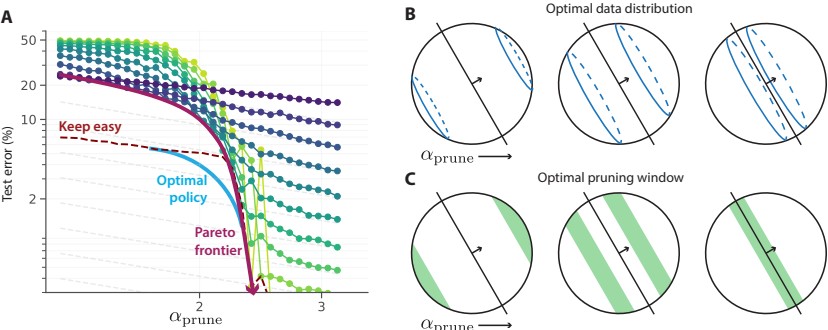

Figure 7: Optimal pruning policy as a function of $\alpha_{\text{prune}}$. **A**, The Pareto frontier in Fig. A**A** can be lowered in the small $\alpha_{\text{prune}}$ regime if one adaptively switches pruning policies from a "keep easy" to a "keep hard" policy. The dashed purple line indicates the "keep easy" frontier (computed using numerical simulations). The optimal pruning window (derived below) interpolates between the two policies, achieving the lowest possible Pareto frontier (cartooned in blue). **B**, The optimal data distribution along the teacher $p(z|\alpha_{\text{prune}}, f)$ is a delta function, which selects easy examples for small $\alpha_{\text{prune}}$, intermediate examples for intermediate $\alpha_{\text{prune}}$, and hard examples for large $\alpha_{\text{prune}}$. **C**, The optimal pruning window similarly selects easy examples for small $\alpha_{\text{prune}}$, intermediate examples for intermediate $\alpha_{\text{prune}}$, and hard examples for large $\alpha_{\text{prune}}$

These observations beg the question: what is the best policy in the intermediate $\alpha_{\text{prune}}$ regime? Is there a globally optimal pruning policy which interpolates between the "keep easy" and "keep hard" strategies and achieves the lowest possible Pareto frontier (blue curve in Fig. 7A)?

In this section we investigate this question. Using the calculus of variations, we first derive the optimal data distribution $p(z|\alpha_{\text{prune}}, f)$ along the teacher for a given $\alpha_{\text{prune}}, f$. We begin by framing the problem using the method of Lagrange multipliers. Seeking to optimize $R$ under the constraints imposed by the saddle point equations Eqs. 66,67, we define the Lagrangian,

$$\mathcal{L} = R + \mu\left(R - 2\alpha\int_0^\infty dz\, p(z)\varphi(z; R, k)\right) + \lambda\left(1 - R^2 - 2\alpha\int_0^\infty dz\, p(z)\psi(z; R, k)\right). \quad (82)$$

Where,

$$\varphi(z; R, k) = \int_{-\infty}^\kappa \frac{dt}{\sqrt{2\pi}\sqrt{1-R^2}} \exp\left(-\frac{(t-Rz)^2}{2(1-R^2)}\right)\left(\frac{z-Rt}{1-R^2}\right)(\kappa - t), \quad (83)$$

and,

$$\psi(z; R, k) = \int_{-\infty}^\kappa \frac{dt}{\sqrt{2\pi}\sqrt{1-R^2}} \exp\left(-\frac{(t-Rz)^2}{2(1-R^2)}\right)(\kappa - t)^2 \quad (84)$$

Taking a variational derivative $\frac{\delta\mathcal{L}}{\delta p}$ with respect to the data distribution $p$, we obtain an equation for $z$, indicating that the optimal distribution is a delta function at $z = z^*$. To find the optimal location of the delta function $z^*$, we take derivatives with respect to the remaining variables $R, k, \mu, \lambda$ and solve the resulting set of equations numerically. The qualitative behavior is shown in Fig. 7A. As

$\alpha_{\text{prune}}$ grows, the location of the delta function shifts from infinity to zero, confirming that the optimal strategy for small $\alpha_{\text{prune}}$ is to keep the "easy" (large-margin) examples, and for large $\alpha_{\text{prune}}$ to keep the "hard" (small-margin) examples.

Interestingly, this calculation also reveals that if the location of the delta function is chosen optimally, the student can perfectly recover the teacher ($R = 1$, zero generalization error) for any $\alpha_{\text{prune}}$. This observation, while interesting, is of no practical consequence because it relies on an infinitely large training set from which examples can be precisely selected to perfectly recover the teacher. Therefore, to derive the optimal pruning policy for a more realistic scenario, we assume a gaussian distribution of data along the teacher direction and model pruning as keeping only those examples which fall inside a window $a < z < b$. The saddle point equations, Eqs. 66,67, then take the form,

$$R = \frac{2\alpha}{f\sqrt{\pi/2}\sqrt{1-R^2}} \int_{-\infty}^{\kappa} Dt \left[ \exp\left(-\frac{(a-Rt)^2}{2(1-R^2)}\right) - \exp\left(-\frac{(b-Rt)^2}{2(1-R^2)}\right) \right] (\kappa - t) \quad (85)$$

$$1 - R^2 = \frac{4\alpha}{f} \int_{-\infty}^{\kappa} Dt \left[ H\left(\frac{a-Rt}{\sqrt{1-R^2}}\right) - H\left(\frac{b-Rt}{\sqrt{1-R^2}}\right) \right] (\kappa - t)^2 \quad (86)$$

Where $a$ must satisfy $a = H^{-1}(f/2 + H(b))$. For each $f, \alpha$, we find the optimal location of this window using the method of Lagrange multipliers. Defining the Lagrangian as before,

$$\mathcal{L} = R + \mu\left(R - 2\alpha \int_0^\infty dz\, p(z)\varphi(z; R, k)\right) + \lambda\left(1 - R^2 - 2\alpha \int_0^\infty dz\, p(z)\psi(z; R, k)\right), \quad (87)$$

Where now,

$$\phi(b; R, k) = \left[ \exp\left(-\frac{(a-Rt)^2}{2(1-R^2)}\right) - \exp\left(-\frac{(b-Rt)^2}{2(1-R^2)}\right) \right] (\kappa - t) \quad (88)$$

$$\psi(b; R, k) = \left[ H\left(\frac{a-Rt}{\sqrt{1-R^2}}\right) - H\left(\frac{b-Rt}{\sqrt{1-R^2}}\right) \right] (\kappa - t)^2 \quad (89)$$

To find the optimal location of the pruning window, we take derivatives with respect to the remaining variables $b, R, k, \mu, \lambda$ and solve the resulting set of equations numerically. Consistent with the results for the optimal distribution, the location of the optimal window shifts from around infinity to around zero as $\alpha_{\text{prune}}$ grows (Fig. 7C).

### A.9 Exact saddle point equations

To obtain exact expressions for the generalization error for all $\theta$, we can extremize Eq. 63 wrt $R, q, \rho$.

**Derivative wrt $R$**

$$\frac{R - \rho\cos\theta}{(1-q)\sin^2\theta} = \int dt \int dz\, p(z) \frac{\sqrt{2}\alpha}{\pi} \left( \frac{t\sqrt{q-\rho^2}}{\sqrt{2}\sqrt{(\rho^2-q)}\Lambda} - \frac{(R-\rho\cos\theta)\Gamma(t,z)}{\sqrt{2}\sqrt{q-\rho^2}\Lambda^3} \right) \quad (90)$$

$$\times \log\left( H\left(\frac{\kappa - t\sqrt{q-\rho^2} - \rho z}{\sqrt{1-q}}\right) \right) \exp\left( -\frac{\Gamma(t,z)^2}{2(q-\rho^2)\Lambda^2} - \frac{t^2}{2} \right) \quad (91)$$

Where we have defined,

$$\Lambda = \sqrt{q\sin^2\theta - R^2 - \rho^2 + 2R\rho\cos\theta}, \quad (92)$$

$$\Gamma(t,z) = (R - \rho\cos\theta)\left(t\sqrt{q-\rho^2} + \rho z\right) + qz\cos\theta - \rho Rz. \quad (93)$$

Integrating the right-hand side by parts,

$$\frac{R-\rho\cos\theta}{(1-q)\sin^2\theta} = -\int dt\int dz\, p(z)\frac{\alpha}{\pi\Lambda}\frac{\sqrt{q-\rho^2}\,e^{-\frac{\left(\kappa-t\sqrt{q-\rho^2}-\rho z\right)^2}{2(1-q)}}}{\sqrt{2\pi}\sqrt{1-q}\mathrm{H}\left(\frac{\kappa-t\sqrt{q-\rho^2}-\rho z}{\sqrt{1-q}}\right)}\exp\left(-\frac{\Delta(t,z)}{2\Lambda^2}\right) \quad (94)$$

where

$$\Delta(t,z) = 2tz\sqrt{q-\rho^2}(R-\rho\cos\theta)\cos\theta + qt^2\sin^2\theta + qz^2\cos^2\theta - \rho^2t^2\sin^2\theta - \rho^2z^2\cos^2\theta \quad (95)$$

Changing variables to $t\to t\sqrt{q-\rho^2}+\rho z$ and taking the limit $q\to 1$,

$$\frac{R-\rho\cos\theta}{\sin^2\theta} = \left\langle\int_{-\infty}^{\kappa}dt\frac{\alpha}{\pi\Lambda}\exp\left(-\frac{\Delta(t,z)}{2\Lambda^2}\right)(\kappa-t)\right\rangle_z \quad (96)$$

Where with this change of variables,

$$\Lambda = \sqrt{q\sin^2\theta - R^2 - \rho^2 + 2\rho R\cos\theta} \quad (97)$$

$$\Delta(t,z) = z^2\left(\rho^2+\cos^2\theta-2\rho R\cos\theta\right) + 2tz(R\cos\theta-\rho) + t^2\sin^2\theta \quad (98)$$

**Derivative wrt $q$,**

$$\frac{q-(\rho^2+R^2-2\rho R\cos\theta)/\sin^2\theta}{2(1-q)^2} = \int dt\int dz\, p(z)\frac{2\alpha}{\pi}\left(\frac{\kappa-t\sqrt{q-\rho^2}-\rho z}{(1-q)^{3/2}}-\frac{t}{\sqrt{1-q}\sqrt{q-\rho^2}}\right) \quad (99)$$

$$\times\exp\left(-\frac{\left(\kappa-t\sqrt{q-\rho^2}-\rho z\right)^2}{2(1-q)}-\frac{t^2}{2}\right) \quad (100)$$

$$\times H\left(-\frac{(R-\rho\cos\theta)\left(t\sqrt{q-\rho^2}+\rho z\right)+qz\cos\theta-\rho Rz}{\sqrt{(\rho^2-q)\left(\rho^2-q\sin^2\theta+R^2-2\rho R\cos\theta\right)}}\right)H\left(\frac{\kappa-t\sqrt{q-\rho^2}-\rho z}{\sqrt{1-q}}\right) \quad (101)$$

$$-\frac{\sqrt{2}\alpha}{\pi}\left(\frac{\frac{t(R-\rho\cos\theta)}{2\sqrt{q-\rho^2}}+z\cos}{\sqrt{2}\sqrt{(q-\rho^2)}\Lambda}-\frac{\left((q-\rho^2)\sin^2\theta+\Lambda^2\right)\Gamma(t,z)}{2\sqrt{2}\left(\rho^2-q\right)^{3/2}\Lambda^3}\right) \quad (102)$$

$$\times\log\left(H\left(\frac{\kappa-t\sqrt{q-\rho^2}-\rho z}{\sqrt{1-q}}\right)\right)\exp\left(-\frac{\left((R-\rho\cos\theta)\left(t\sqrt{q-\rho^2}+\rho z\right)+qz\cos\theta-\rho Rz\right)^2}{2\left(q-\rho^2\right)\Lambda^2}-\frac{t^2}{2}\right) \quad (103)$$

Where $\Gamma(t,z) = (R-\rho\cos\theta)\left(t\sqrt{q-\rho^2}+\rho z\right)+qz\cos\theta-\rho Rz$. After integating by parts,

$$\frac{q-(\rho^2+R^2-2\rho R\cos\theta)/\sin^2\theta}{2(1-q)^2} = \int dt\int dz\, p(z)\frac{\alpha\exp\left(-\frac{2t\sqrt{q-\rho^2}(\rho z-\kappa)-\left(\rho^2-1\right)t^2+(\kappa-\rho z)^2}{2(1-q)}\right)}{4\pi H\left(\frac{\kappa-t\sqrt{q-\rho^2}-\rho z}{\sqrt{1-q}}\right)^2} \quad (104)$$

$$\times \sqrt{\frac{2}{\pi}} e^{-\frac{\left(\kappa - t\sqrt{q-\rho^2} - \rho z\right)^2}{2(1-q)}} \mathrm{H}\left(-\frac{\Gamma(t,z)}{\sqrt{(q-\rho^2)\,\Lambda}}\right) \tag{105}$$

Changing variables to $t \to t\sqrt{q-\rho^2} + \rho z$ and taking the limit $q \to 1$,

$$1 - \frac{\rho^2 + R^2 - 2\rho R \cos\theta}{\sin^2\theta} = 2\alpha\left\langle \int_{-\infty}^{\kappa} \frac{dt\, e^{-\frac{(t-\rho z)^2}{2(1-\rho^2)}}}{\sqrt{2\pi}\sqrt{1-\rho^2}} H\left(\frac{\Gamma(t,z)}{\sqrt{1-\rho^2}\Lambda}\right)(\kappa - t)^2\right\rangle_z \tag{106}$$

Where now $\Gamma(t,z) = z(\rho R - \cos\theta) - t(R - \rho\cos\theta)$.

**Derivative wrt $\rho$,**

$$\frac{\rho - R\cos\theta}{(1-q)\sin^2\theta} = \frac{\alpha}{2\pi} \frac{\left(\frac{\rho t}{\sqrt{q-\rho^2}} - z\right)\exp\left(-\frac{\left(\kappa - t\sqrt{q-\rho^2} - \rho z\right)^2}{2(1-q)} - \frac{t^2}{2}\right) H\left(-\frac{\Gamma(t,z)}{\sqrt{q-\rho^2}\Lambda}\right)}{\sqrt{1-q}\mathrm{H}\left(\frac{\kappa - t\sqrt{q-\rho^2} - \rho z}{\sqrt{1-q}}\right)} \tag{107}$$

$$+ \frac{\sqrt{2}\alpha}{\pi}\left(\frac{(R - \rho\cos\theta)\left(z - \frac{\rho t}{\sqrt{q-\rho^2}}\right) - \left(t\sqrt{q-\rho^2} + \rho z\right)\cos\theta - Rz}{\sqrt{2}\sqrt{q-\rho^2}\Lambda}\right. \tag{108}$$

$$\left. - \frac{\left(-2\rho\Lambda^2 - \left(q-\rho^2\right)(2\rho - 2R\cos\theta)\right)\Gamma(t,z)}{2\sqrt{2}\left(q-\rho^2\right)^{3/2}\Lambda^3}\right) \tag{109}$$

$$\times \log\left(\frac{1}{2}\mathrm{erfc}\left(\frac{\kappa - t\sqrt{q-\rho^2} - \rho z}{\sqrt{2}\sqrt{1-q}}\right)\right)\exp\left(-\frac{\Gamma(t,z)^2}{2\left(q-\rho^2\right)\Lambda^2} - \frac{t^2}{2}\right) \tag{110}$$

Integrating the second term by parts,

$$\frac{\rho - R\cos\theta}{(1-q)\sin^2\theta} = \frac{\alpha}{\pi} \frac{\left(\frac{\rho t}{\sqrt{q-\rho^2}} - z\right)\exp\left(-\frac{\left(\kappa - t\sqrt{q-\rho^2} - \rho z\right)^2}{2(1-q)} - \frac{t^2}{2}\right) H\left(-\frac{\Gamma(t,z)}{\sqrt{q-\rho^2}\Lambda}\right)}{\sqrt{1-q}\mathrm{H}\left(\frac{\kappa - t\sqrt{q-\rho^2} - \rho z}{\sqrt{1-q}}\right)} \tag{111}$$

$$- \frac{\alpha}{\pi\Delta}\exp\left(-\frac{\Delta(t,z)}{2\Lambda^2}\right)\left(\frac{\rho R - q\cos\theta}{q - \rho^2}\right) \tag{112}$$

Changing variables to $t \to t\sqrt{q-\rho^2} + \rho z$ and taking the limit $q \to 1$,

$$\frac{\rho - R\cos\theta}{\sin^2\theta} = 2\alpha\left\langle \int_{-\infty}^{\kappa} dt \frac{e^{-\frac{(t-\rho z)^2}{2(1-\rho^2)}}}{\sqrt{2\pi}\sqrt{1-\rho^2}} H\left(\frac{\Gamma(t,z)}{\sqrt{1-\rho^2}\Lambda}\right)\left(\frac{z - \rho t}{1 - \rho^2}\right)(\kappa - t)\right. \tag{113}$$

$$\left. + \frac{1}{2\pi\Lambda}\exp\left(-\frac{\Delta(t,z)}{2\Lambda^2}\right)\left(\frac{\rho R - \cos\theta}{1 - \rho^2}\right)(\kappa - t)\right\rangle_z \tag{114}$$

So together we have three saddle point equations:

$$\frac{R - \rho\cos\theta}{\sin^2\theta} = \frac{\alpha}{\pi\Lambda}\left\langle \int_{-\infty}^{\kappa} dt\ \exp\left(-\frac{\Delta(t,z)}{2\Lambda^2}\right)(\kappa - t)\right\rangle_z \tag{115}$$

$$1 - \frac{\rho^2 + R^2 - 2\rho R\cos\theta}{\sin^2\theta} = 2\alpha\left\langle \int_{-\infty}^{\kappa} dt\,\frac{e^{-\frac{(t-\rho z)^2}{2(1-\rho^2)}}}{\sqrt{2\pi}\sqrt{1-\rho^2}}H\left(\frac{\Gamma(t,z)}{\sqrt{1-\rho^2}\Lambda}\right)(\kappa - t)^2\right\rangle_z \tag{116}$$

$$\frac{\rho - R\cos\theta}{\sin^2\theta} = 2\alpha\left\langle \int_{-\infty}^{\kappa} dt\,\frac{e^{-\frac{(t-\rho z)^2}{2(1-\rho^2)}}}{\sqrt{2\pi}\sqrt{1-\rho^2}}H\left(\frac{\Gamma(t,z)}{\sqrt{1-\rho^2}\Lambda}\right)\left(\frac{z-\rho t}{1-\rho^2}\right)(\kappa - t)\right. \tag{117}$$

$$\left. + \frac{1}{2\pi\Lambda}\exp\left(-\frac{\Delta(t,z)}{2\Lambda^2}\right)\left(\frac{\rho R - \cos\theta}{1-\rho^2}\right)(\kappa - t)\right\rangle_z \tag{118}$$

Where

$$\Lambda = \sqrt{\sin^2\theta - R^2 - \rho^2 + 2\rho R\cos\theta}, \tag{119}$$

$$\Gamma(t,z) = z(\rho R - \cos\theta) - t(R - \rho\cos\theta), \tag{120}$$

$$\Delta(t,z) = z^2\left(\rho^2 + \cos^2\theta - 2\rho R\cos\theta\right) + 2tz(R\cos\theta - \rho) + t^2\sin^2\theta. \tag{121}$$

Solving these equations numerically yields an excellent fit to numerical simulations on structured data (Fig. 2**BCD**).

## B  Model training method details & dataset information

**Perceptron in the teacher-student setting**    All code to reproduce these simulations can be found at: `https://colab.research.google.com/drive/1in35C6jh7y_ynwuWLBmGOWAgmUgpl8dF?usp=sharing`. Perceptrons were trained on a synthetic dataset of $P$ examples $\{\mathbf{x}^\mu, y^\mu\}_{\mu=1,\dots,P}$, where $\mathbf{x}^\mu \sim \mathcal{N}(0, I_N)$ are i.i.d. zero mean unit variance random Gaussian inputs, and $y^\mu = \text{sign}(\mathbf{T}\cdot x)$ are labels generated by a teacher perceptron $\mathbf{T} \in \mathbb{R}^N$, which was randomly drawn from a uniform distribution on the sphere $\mathbf{T} \sim \text{Unif}(\mathbb{S}^{N-1}(\sqrt{N}))$. For all of our experiments we fixed $N = 200$ and set $P = \alpha N$ where $\alpha$ varied between $10^{0.1}$ and $10^{0.5}$. Each synthetic dataset was pruned to keep a fraction $f$ of the smallest-margin examples, where $f$ varied between 0.1 and 1 in Fig. 1 and between 0.2 and 1 in Figs. 2,3 to match the real-world experiments in Fig. 3. Perceptrons were optimized to find the max-margin separating solution using a standard quadratic programming (QP) algorithm from the CVXPY library (for analysis of the computational complexity of this algorithm see Fig. 8). Results were averaged over 100 independent draws of the teacher and training examples.

**ImageNet.**    ImageNet model training was performed using a standard ResNet-50 through the VISSL library [40] (stable version `v0.1.6`), which provides default configuration files for supervised ResNet-50 training (accessible here; released under the MIT license). Each model was trained on a single node of 8 NVIDIA V100 32GB graphics cards with BATCHSIZE_PER_REPLICA = 256, using the Stochastic Gradient Descent (SGD) optimizer with a base learning rate = 0.1, nesterov momentum = 0.9, and weight decay = 0.001. For our scaling experiments (Fig. 3 and Fig. 9), we trained one model per fraction of data kept (0.1-1.0) for each dataset size. In total, these plot required training 97 models on (potentially a subset of) ImageNet. All the models were trained with matched number of iterations, corresponding to 105 epochs on the full ImageNet dataset. The learning rate was decayed by a factor of 10 after the number of iterations corresponding to 30, 60, 90, and 100 epochs on the full ImageNet dataset.

For our main ImageNet experiments (Fig. 5) we trained one model per fraction of data kept (1.0, 0.9, 0.8, 0.7, 0.6) $\times$ metric (11 metrics in total). In the plot itself, since any variation in the "fraction of data kept = 1.0" setting is due to random variation across runs not due to potential metric differences, we averaged model performances to obtain a single datapoint here (while also keeping track of the variation across models, which is plotted as $\pm 2$ standard deviations). In total, this plot required training 55 models on (potentially a subset of) ImageNet. For Fig. 5C, in order to reduce noise from random variation, we additionally trained five models per datapoint and metric, and plot the averaged performance in addition to error bars showing one standard deviation of the mean. ImageNet [41]

is released under the ImageNet terms of access. It is important to note that ImageNet images are often biased [42, 43]. The SWaV model used to compute our prototypicality metrics was obtained via `torch.hub.load('facebookresearch/swav:main', 'resnet50')`, which is the original model provided by [35]; we then used the `avgpool` layer's activations.

**CIFAR-10 and SVHN.**   CIFAR-10 and SVHN model training was performed using a standard ResNet-18 through the PyTorch library. Each model was trained on a single NVIDIA TITAN Xp 12GB graphics card with batch size = 128, using the Stochastic Gradient Descent (SGD) optimizer with learning rate = 0.1, nesterov momentum = 0.9, and weight decay = 0.0005. Probe models were trained for 20 epochs each for CIFAR-10 and 40 epochs each for SVHN. Pruning scores were then computed using the EL2Ns metric [10], averaged across 10 independent initializations of the probe models. To evaluate data pruning performance, fresh models were trained from scratch on each pruned dataset for 200 epochs, with the learning rate decayed by a factor of 5 after 60, 120 and 160 epochs.

**Data pruning for transfer learning.**   To assess the effect of pruning downstream finetuning data on transfer learning performance, vision transformers (ViTs) pre-trained on ImageNet21k were fine-tuned on different pruned subsets of CIFAR-10. Pre-trained models were obtained from the timm model library [44]. Each model was trained on a single NVIDIA TITAN Xp 12GB graphics card with batch size = 128, using the Adam optimizer with learning rate = 1e-5 and no weight decay. Probe models were trained for 2 epochs each. Pruning scores were then computed using the EL2Ns metric [10], averaged across 10 independent random seeds. To evaluate data pruning performance, pre-trained models were fine-tuned on each pruned dataset for 10 epochs.

To assess the effect of pruning upstream pretraining data on transfer learning performance, each of the ResNet-50s pre-trained on pruned subsets of ImageNet1k in Fig. 3**D** was fine-tuned on all of CIFAR-10. Each model was trained on a single NVIDIA TITAN Xp 12GB graphics card with batch size = 128, using the RMSProp optimizer with learning rate = 1e-4 and no weight decay. Probe models were trained for 2 epochs each. Pruning scores were then computed using the EL2Ns metric [10], averaged across 10 independent random seeds. To evaluate data pruning performance, pre-trained models were fine-tuned on each pruned dataset for 10 epochs.

## C   Breaking compute scaling laws via data pruning

Do the savings in training dataset size we have identified translate to savings in compute, and can data pruning be used to beat widely observed compute scaling laws [2, 5, 7]? Here we show for the perceptron that data pruning can afford exponential savings in compute, and we provide preliminary evidence that the same is true for ResNets trained on CIFAR-10 and ImageNet. We repeat the perceptron learning experiments in Fig. 1A, keeping track of the computational complexity of each experiment, measured by the time to convergence of the quadratic programming algorithm used to find a max-margin solution (see B for details). Across all experiments, the convergence time $T$ was linearly proportional to $\alpha_{\text{prune}}$ with $T = 0.96\alpha_{\text{prune}} + 0.80$, allowing us to replace the x-axis of 1A with compute to produce Fig. 8A, which reveals that data pruning can be used to break compute scaling laws for the perceptron.

Motivated by this, we next investigate whether the convergence time of neural networks trained on pruned datasets depends largely on the number of examples and not their difficulty, potentially allowing for exponential compute savings. We investigate the learning curves of a ResNet18 trained on CIFAR-10 and a ResNet50 on ImageNet for several different pruning fractions (Fig. 8B). While previous works have fixed the number of iterations [10], here we fix the number of *epochs*, so that the model trained on 60% of the full dataset is trained for only 60% the iterations of the model trained on the full dataset, using only 60% the compute. Nevertheless, we find that the learning curves are strikingly similar across pruning fractions, and appear to converge equally quickly. These results suggest that data pruning could lead to large compute savings in practical settings, and in ongoing experiments we are working to make the analogs of Fig. 8A for ResNets on CIFAR-10 and ImageNet to quantify this benefit.

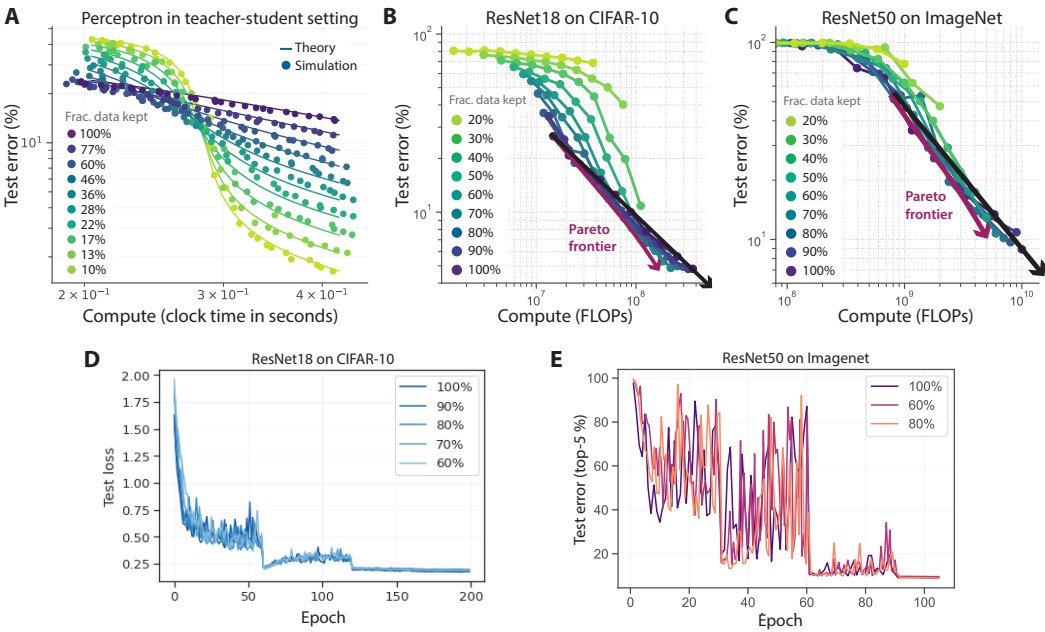

Figure 8: Breaking compute scaling laws via data pruning. **A,B,C**, We repeat the experiments in Figs. 1**A**, 3**C,D**, replacing the x-axis with compute, measured as clock time to convergence for the perceptron (**A**), and FLOPs in a fixed-epoch training setting for the ResNets (**B,C**). Theoretical curves in **A** are overlaid by linearly regressing clock time to convergence $T$ from $\alpha_{\text{prune}}$, with $T = 0.96\alpha_{\text{prune}} + 0.80$. Perceptrons in **A** are trained on a CPU on a google Colab. **E,D**, CIFAR-10 and ImageNet learning curves for fixed epochs.

# D   Additional scaling experiments

In Fig. 9 we perform additional scaling experiments using the EL2Ns and self-supervised prototypes metrics. In Fig. 10 we give a practical example of a cross over from exponential to power-law scaling when the probe student has limited information about the teacher (here a model trained for only a small number of epochs on SVHN) .

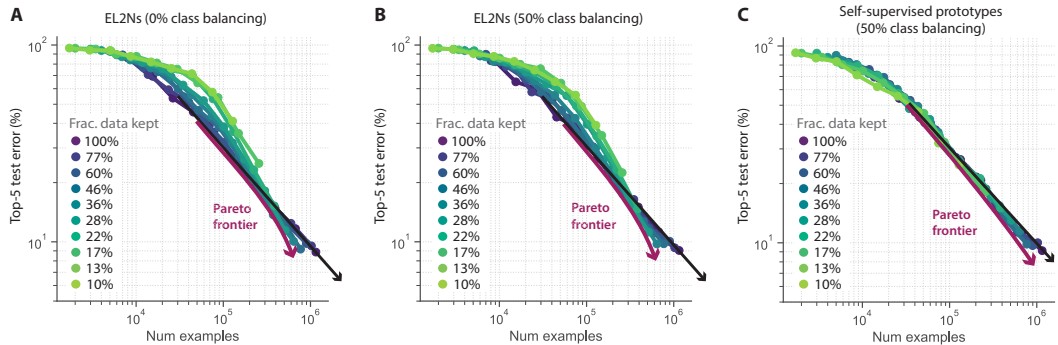

Figure 9: Additional scaling experiments. We reproduce the scaling results on ImageNet in Fig. 3**D** using three additional metrics: two supervised and one self-supervised. Each shows some signatures of breaking power law scaling, although the effect is less dramatic than for the best metric, memorization (Fig. 3**D**). (**A**) EL2Ns with a class balancing fraction of 0% (see App. Section H for details), (**B**) EL2Ns with a class balancing fraction of 50%, and (**C**) Self-supervised prototypes with a class balancing fraction of 50%.

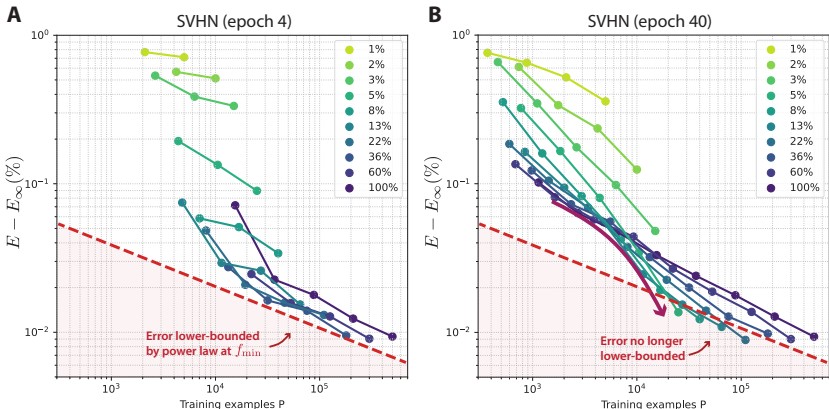

Figure 10: Consistent with a prediction from the perceptron theory (Fig. 2), when SVHN is pruned using a weak metric (a probe trained for only 4 epochs), the learning curve envelope is lower-bounded by a power law at some $f_{min}$ (**A**). However, with a stronger pruning metric (a probe trained for 40 epochs), the learning curve can break through this power law to achieve lower generalization error (**B**).

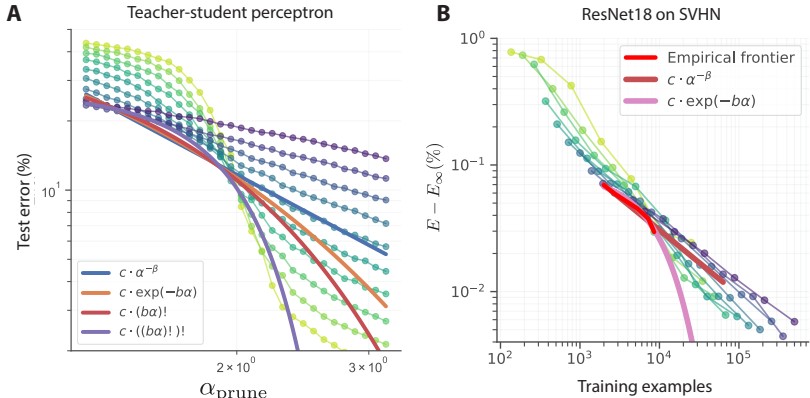

Figure 11: Scaling of the Pareto frontier. (**A**) Different functional forms fitted to the Pareto frontier of the teacher-student perceptron using least squares: power-law (blue), exponential (orange), factorial (red), and iterated factorial (purple). (**B**), We estimate the empirical Pareto frontier (red) for ResNet18 on SVHN at higher resolution by training an additional 300 models in the region around the frontier. We then fit two functional forms to the empirical frontier using least squares: power-law (dark red) and exponential (pink). The exponential shows a much better fit to the empirical Pareto frontier, indicating that the scaling is at least exponential.

## E    Extremal images according to different metrics

In Fig. 6, we showed extremal images for two metrics (self-supervised prototypes, memorization) and a single class. In order to gain a better understanding of how extremal images (i.e. images that are easiest or hardest to learn according to different metrics) look like for all metrics and more classes, we here provide additional figures. In order to avoid cherry-picking classes while at the same time making sure that we are visualizing images for very different classes, we here show extreme images for classes 100, ..., 500 while leaving out classes 0 and 400 (which would have been part of the visualization) since those classes almost exclusively consist of images containing people (0: tench, 400: academic gown). The extremal images are shown in Figures 12,13,14,15,16,17,18,19.

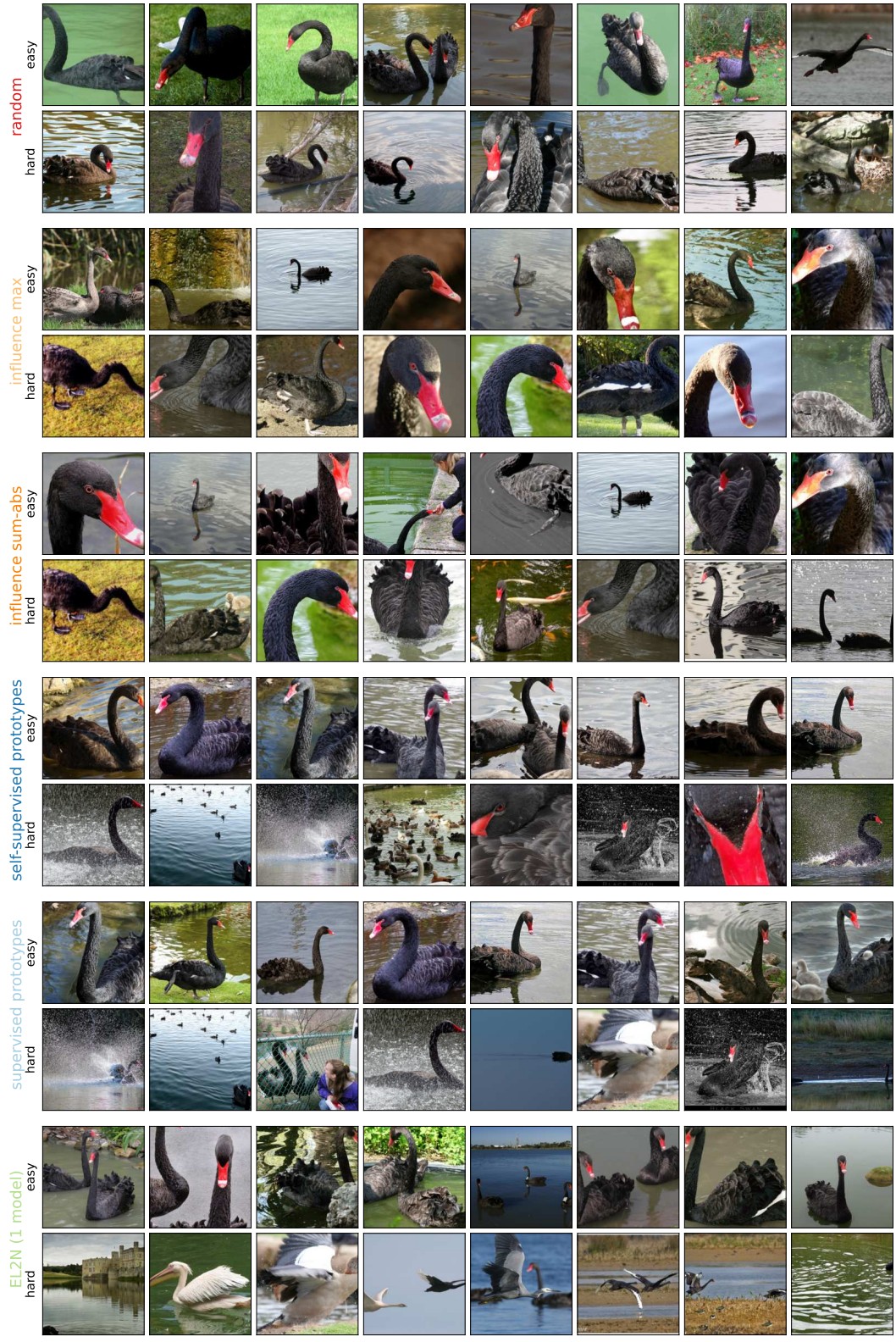

Figure 12: Extreme images according to different metrics for ImageNet class 100 (`black swan`). For each metric, the top row shows images that are ranked as "easy" (most pruneable) according to the metric, and the bottom row shows images that are ranked as "hard" (least pruneable).

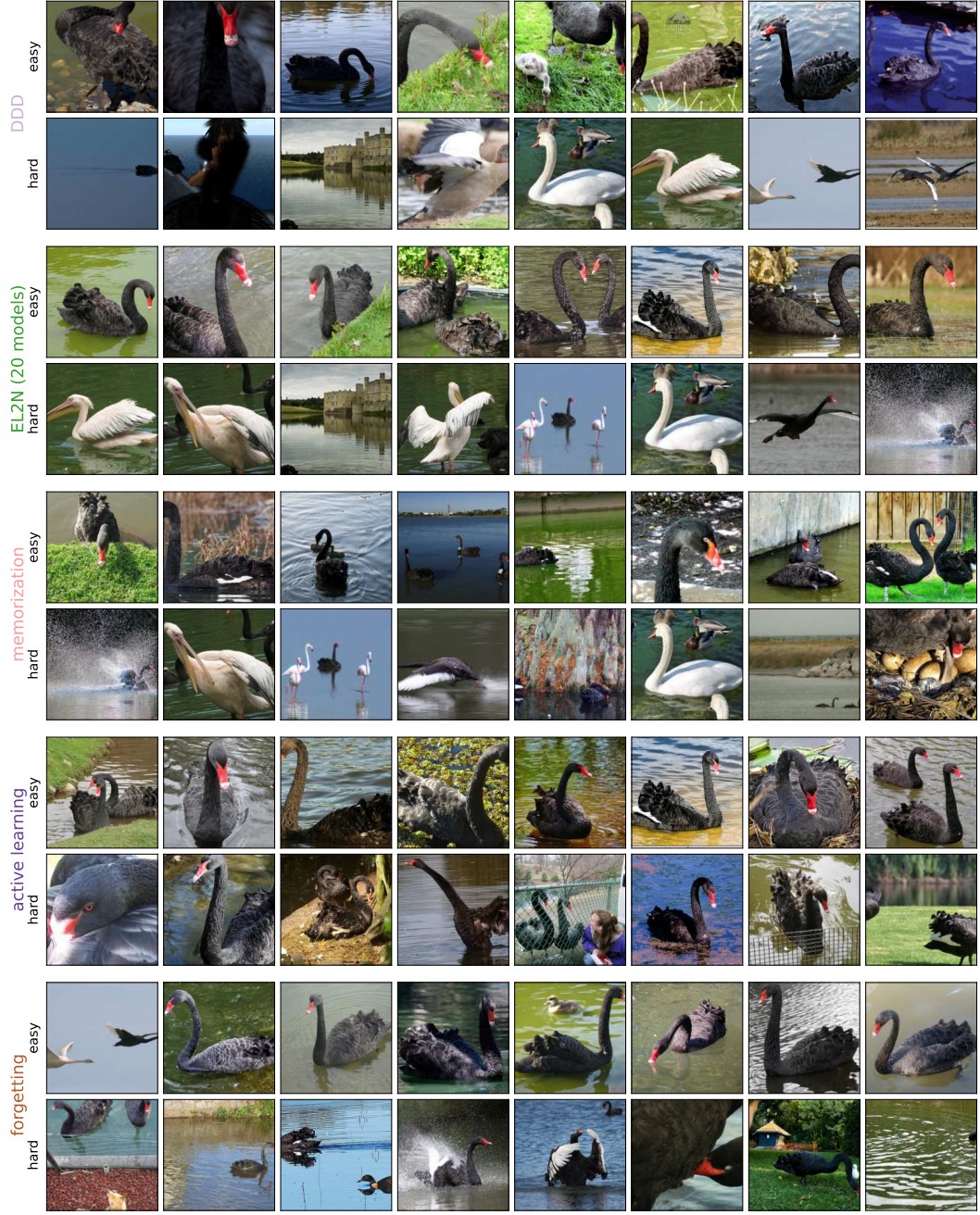

Figure 13: Extreme images according to different metrics for ImageNet class 100 (`black swan`). For each metric, the top row shows images that are ranked as "easy" (most pruneable) according to the metric, and the bottom row shows images that are ranked as "hard" (least pruneable).

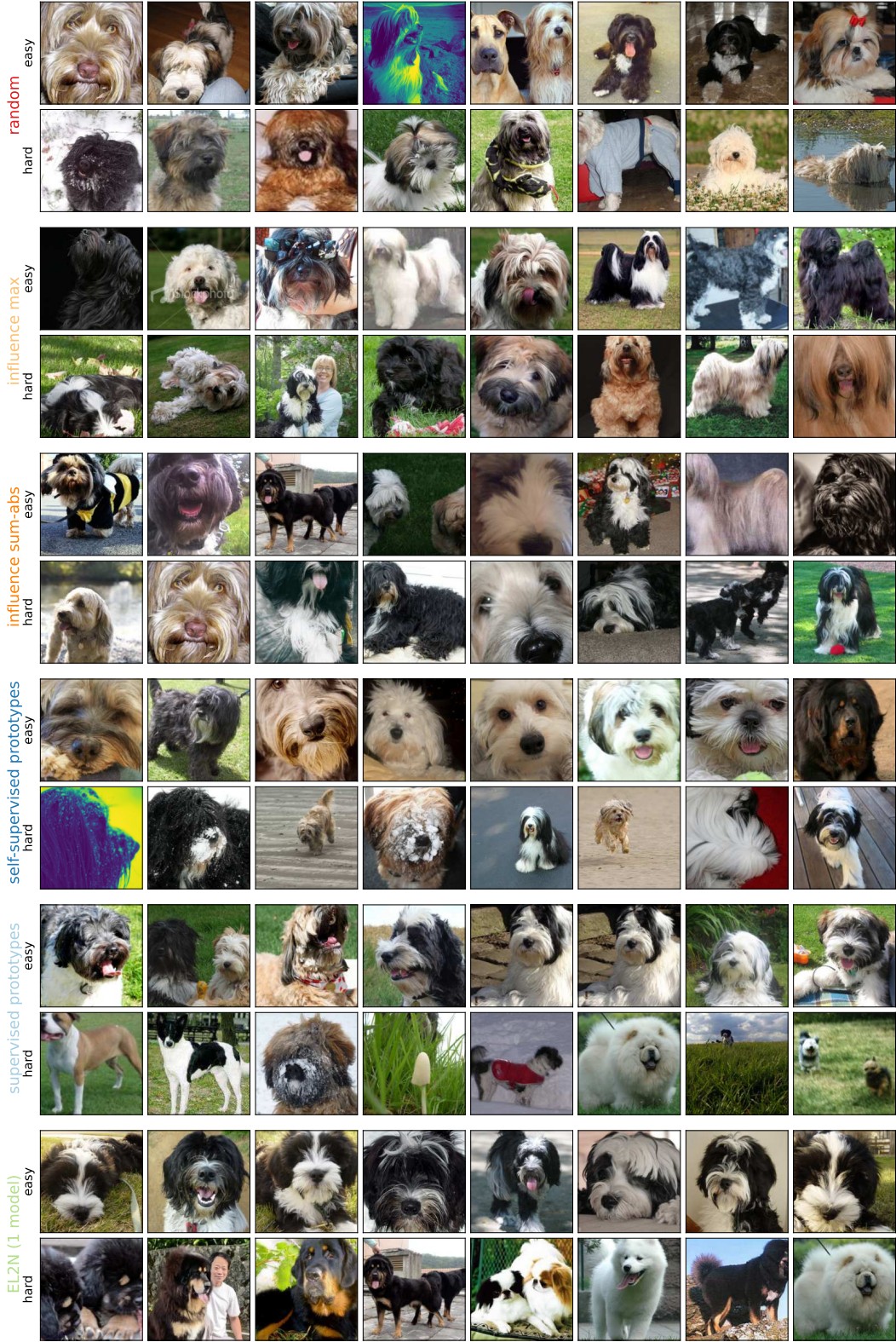

Figure 14: Extreme images according to different metrics for ImageNet class 200 (`Tibetan terrier`). For each metric, the top row shows images that are ranked as "easy" (most pruneable) according to the metric, and the bottom row shows images that are ranked as "hard" (least pruneable).

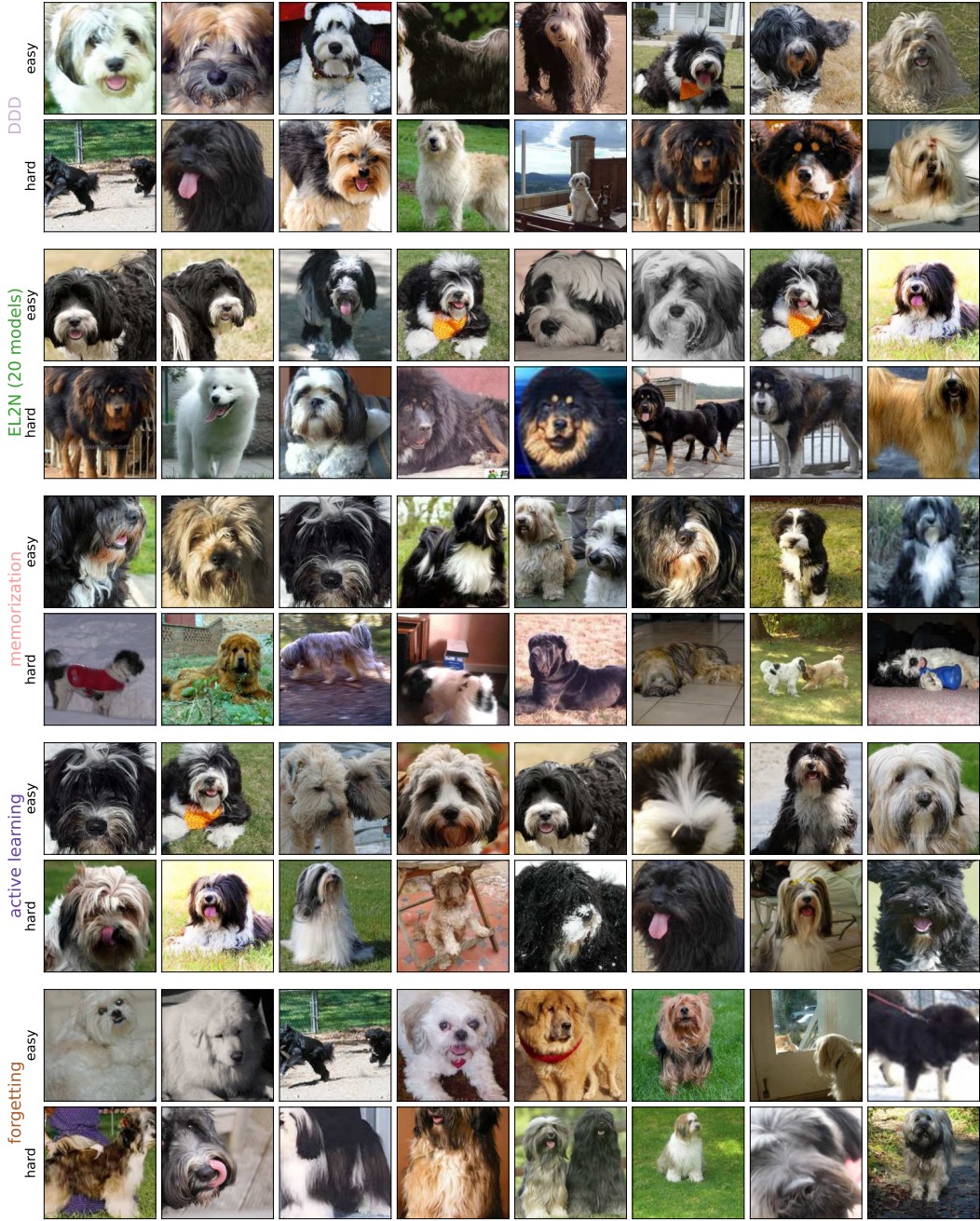

Figure 15: Extreme images according to different metrics for ImageNet class 200 (`Tibetan terrier`). For each metric, the top row shows images that are ranked as "easy" (most pruneable) according to the metric, and the bottom row shows images that are ranked as "hard" (least pruneable).

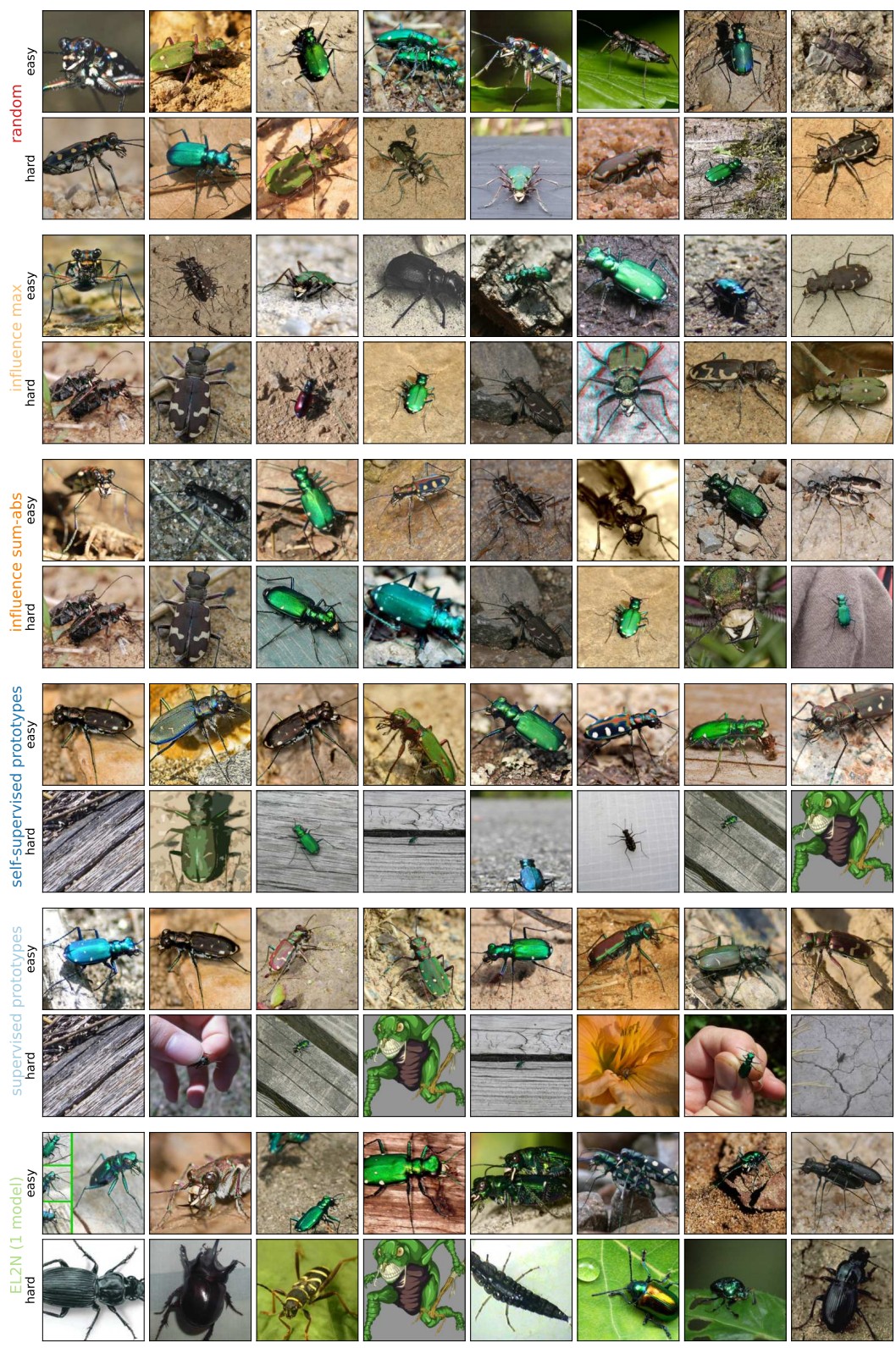

Figure 16: Extreme images according to different metrics for ImageNet class 300 (`tiger beetle`). For each metric, the top row shows images that are ranked as "easy" (most pruneable) according to the metric, and the bottom row shows images that are ranked as "hard" (least pruneable).

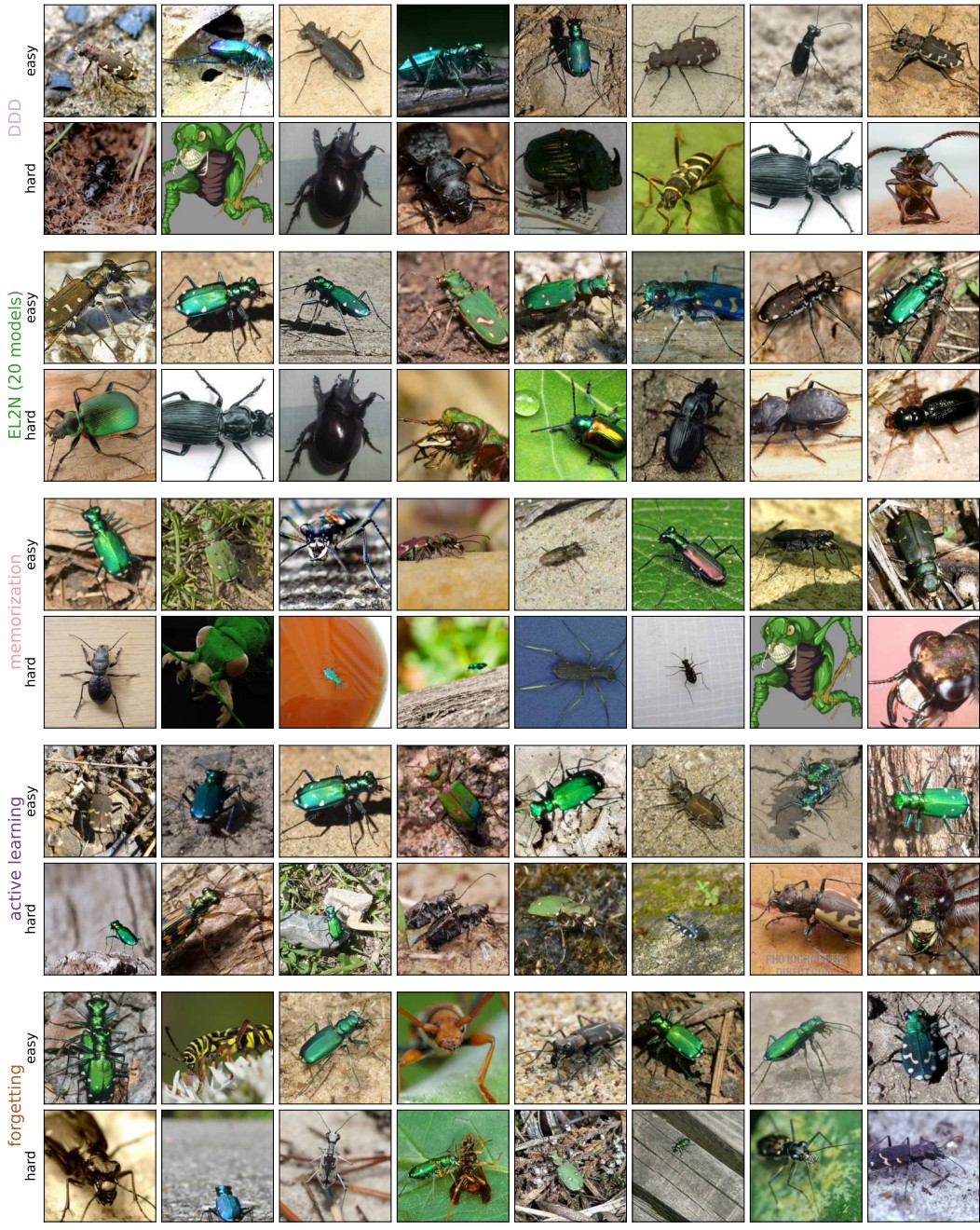

Figure 17: Extreme images according to different metrics for ImageNet class 300 (`tiger beetle`). For each metric, the top row shows images that are ranked as "easy" (most pruneable) according to the metric, and the bottom row shows images that are ranked as "hard" (least pruneable).

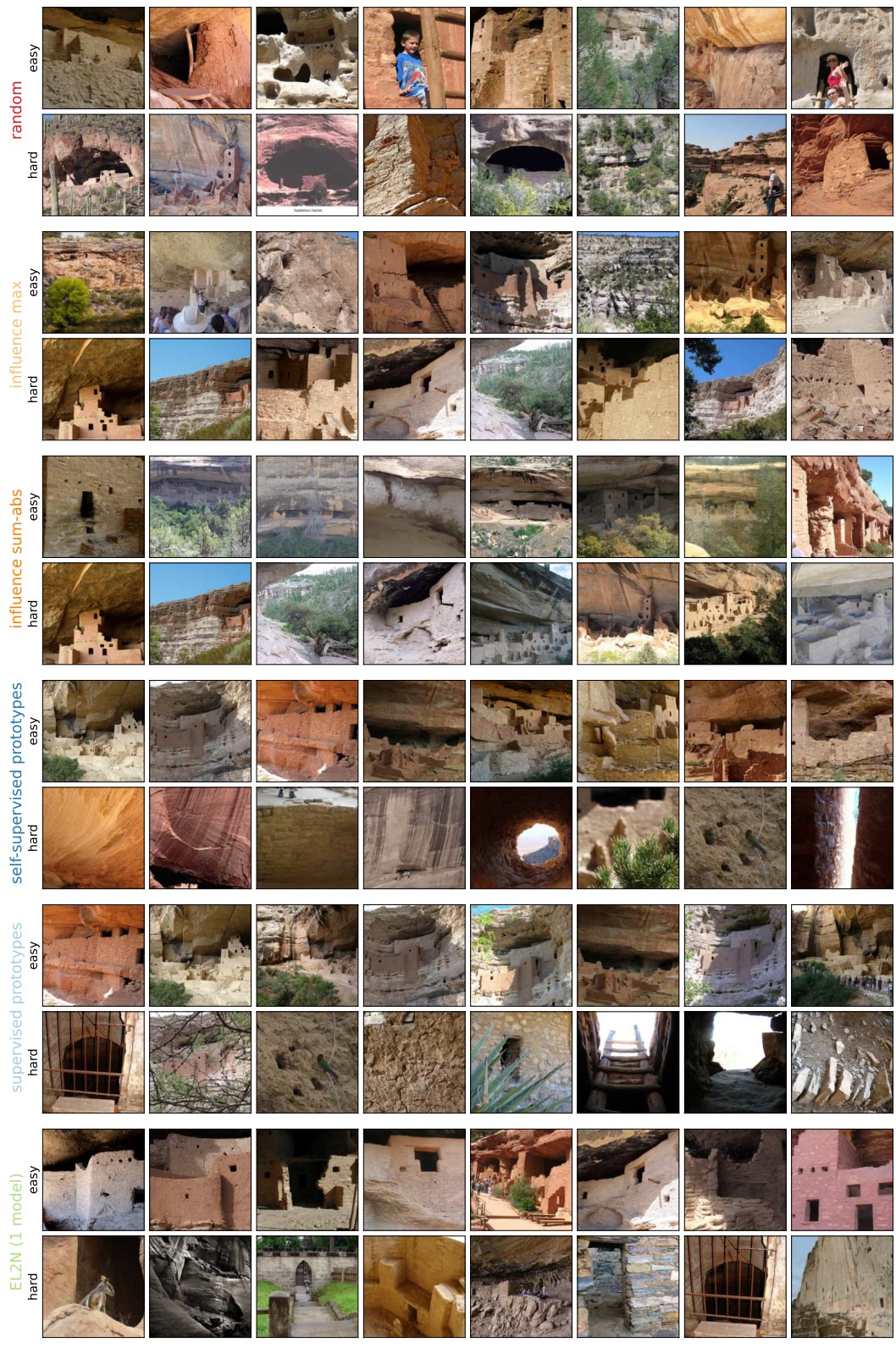

Figure 18: Extreme images according to different metrics for ImageNet class 500 (`cliff dwelling`). For each metric, the top row shows images that are ranked as "easy" (most pruneable) according to the metric, and the bottom row shows images that are ranked as "hard" (least pruneable).

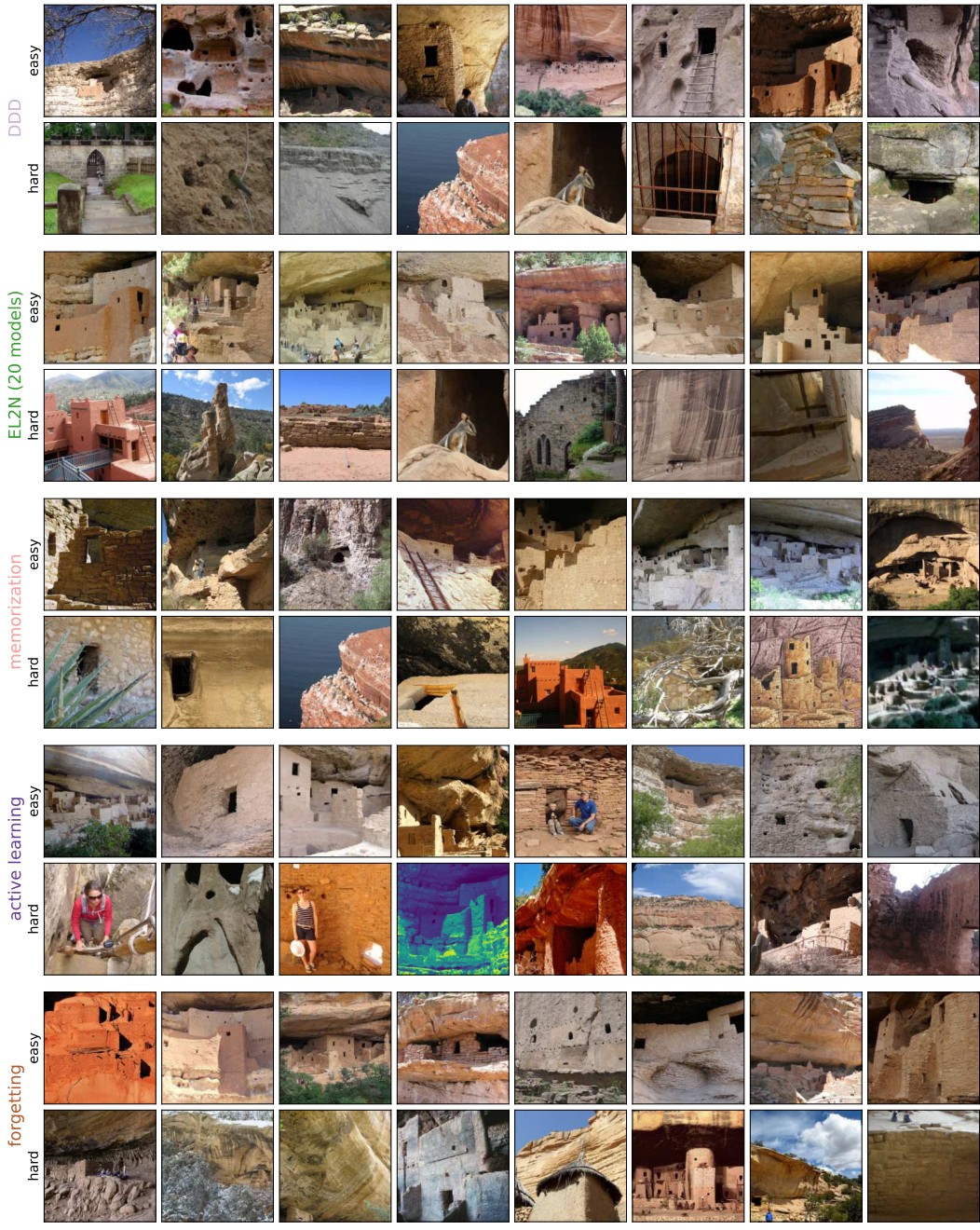

Figure 19: Extreme images according to different metrics for ImageNet class 500 (`cliff dwelling`). For each metric, the top row shows images that are ranked as "easy" (most pruneable) according to the metric, and the bottom row shows images that are ranked as "hard" (least pruneable).

# F  Impact of number of clusters $k$ on self-supervised prototypes

Our self-supervised prototype metric is based on $k$-means clustering, which has a single hyperparameter $k$. By default and throughout the main paper, we set $k = 1000$, corresponding to the number of classes in ImageNet. Here, we investigate other settings of $k$ to understand how this hyperparameter impacts performance. As can be seen in Table 1, $k$ does indeed have an impact on performance, and very small values for $k$ (e.g. $k < 10$) as well as very large values for $k$ (e.g. $k = 50,000$) both lead to performance impairments. At the same time, performance is relatively high across very different in-between settings for $k$. In order to assess these results, it may be important to keep in mind that $\pm 0.54\%$ corresponds to plus/minus 2 standard deviations of performance when simply training the same model multiple times (with different random initialization). Overall, these results suggest that if the number of clusters $k$ deviates at most by one order of magnitude from the number of classes in the dataset (for ImageNet-1K), the exact choice of $k$ does not matter much.

| $k$ | 1 | 5 | 10 | 50 | 100 | 200 | 400 | 600 | 800 | 1K | 5K | 10K | 50K |
|---|---|---|---|---|---|---|---|---|---|---|---|---|---|
| $acc$ | 88.35 | 89.09 | 88.76 | 89.04 | 89.48 | 90.27 | 89.85 | 89.96 | 90.44 | 90.56 | 90.33 | 90.57 | 88.67 |

Table 1: Performance (top-5 accuracy, denoted as $acc$) when pruning away 20% of ImageNet according to our self-supervised prototype metric as a function of the number of prototypes $k$ (hyperparameter used for self-supervised clustering indicating the number of clusters). Results based on training a ResNet-50 architecture with VISSL without class balancing.

# G  Impact of ensemble prototypes

The self-supervised prototypes metric is based on $k$-means clustering in the embedding space of a self-supervised (=SSL) model. Since even otherwise identical models trained with different random seeds can end up with somewhat different embedding spaces, we here investigated how the performance of our self-supervised prototypes metric would change when averaging the scores derived from five models, instead of just using a single model's score. The results, shown in Table 2, indicate that ensembling the self-supervised prototype scores neither improves nor hurts performance. This is both good and bad news: Bad news since naturally any improvement in metric development leads to better data efficiency; on the other hand this is also good news since ensembles increase the computational cost of deriving the metric—and this suggests that ensembling is not necessary to achieve the performance we achieved (unlike in other methods such as ensemble active learning).

| Fraction of data kept | 0.9 | 0.8 | 0.7 |
|---|---|---|---|
| score from single model | 90.03 | 90.10 | 89.38 |
| score from ensemble model | 89.93 | 90.40 | 89.38 |

Table 2: Performance (top-5 acuracy) when pruning away 10%, 20% or 30% of ImageNet based on the self-supervised prototype metric derived from either a single SSL model, or an ensemble of SSL models. Supervised model training on the pruned dataset was performed with a ResNet-50 architecture trained using VISSL without class balancing. Overall, we do not observe any performance difference between the two settings: the differences are still well within a single standard deviation of training multiple models with identical settings. In order to reduce the influence of random variations, each data point in this table shows the average of four independent runs.

# H  Relationship between pruning and class (im-)balance

**Motivation.**  It is well-known that strong class imbalance in a dataset is a challenge that needs to be addressed. In order to understand the effect of pruning on class (im-)balance, we quantified this relationship. For context, if pruning according to a certain metric preferentially leads to discarding most (or even all) images from certain classes, it is likely that the performance on those classes will drop as a result if this imbalance is not addressed.

**Class imbalance metric: pruning amplifies class imbalance.**   Since a standard measure of class (im-)balance—dividing the number of images for the majority class by the number of images for the minority class—is highly sensitive to outliers and discards information about the 998 non-extreme ImageNet classes, we instead calculated a class balance score $b \in [0\%, 100\%]$ as the average class imbalance across any two pairs of classes by computing the expectation over taking two random classes, and then computing how many images the minority class has in proportion to the majority class. For instance, a class balance score of 90% means that on average, when selecting two random classes from the dataset, the smaller of those two classes contains 90% of the number of images of the larger class (higher=better; 100% would be perfectly balanced).

In Fig. 20, we observe that dataset pruning strongly increases class imbalance. This is the case both when pruning away easy images and when pruning away hard images, and the effect occurs for all pruning metrics except, of course, for random pruning. Class imbalance is well-known to be a challenge for deep learning models when not addressed properly [45]. The cause for the amplified class imbalance is revealed when looking at class-conditional differences of metric scores (Figs. 22 and 23): The histograms of the class-conditional score distributions show that for many classes, most (if not all) images have very low scores, while for others most (if not all) images have very high scores. This means that as soon as the lowest / highest scoring images are pruned, certain classes are pruned preferentially and thus class imbalance worsens.

We thus use 50% class balancing for our ImageNet experiments. This ensures that every class has at least 50% of the images that it would have when pruning all classes equally (essentially providing a fixed floor for the minimum number of images per class). This simple fix is an important step to address and counteract class imbalance, although other means could be used as well; and ultimately one would likely want to use a self-supervised version of class (or cluster) balancing when it comes to pruning large-scale unlabeled datasets. For comparison purposes, the results for ImageNet pruning *without* class balancing are shown in supplementary Fig. 21.

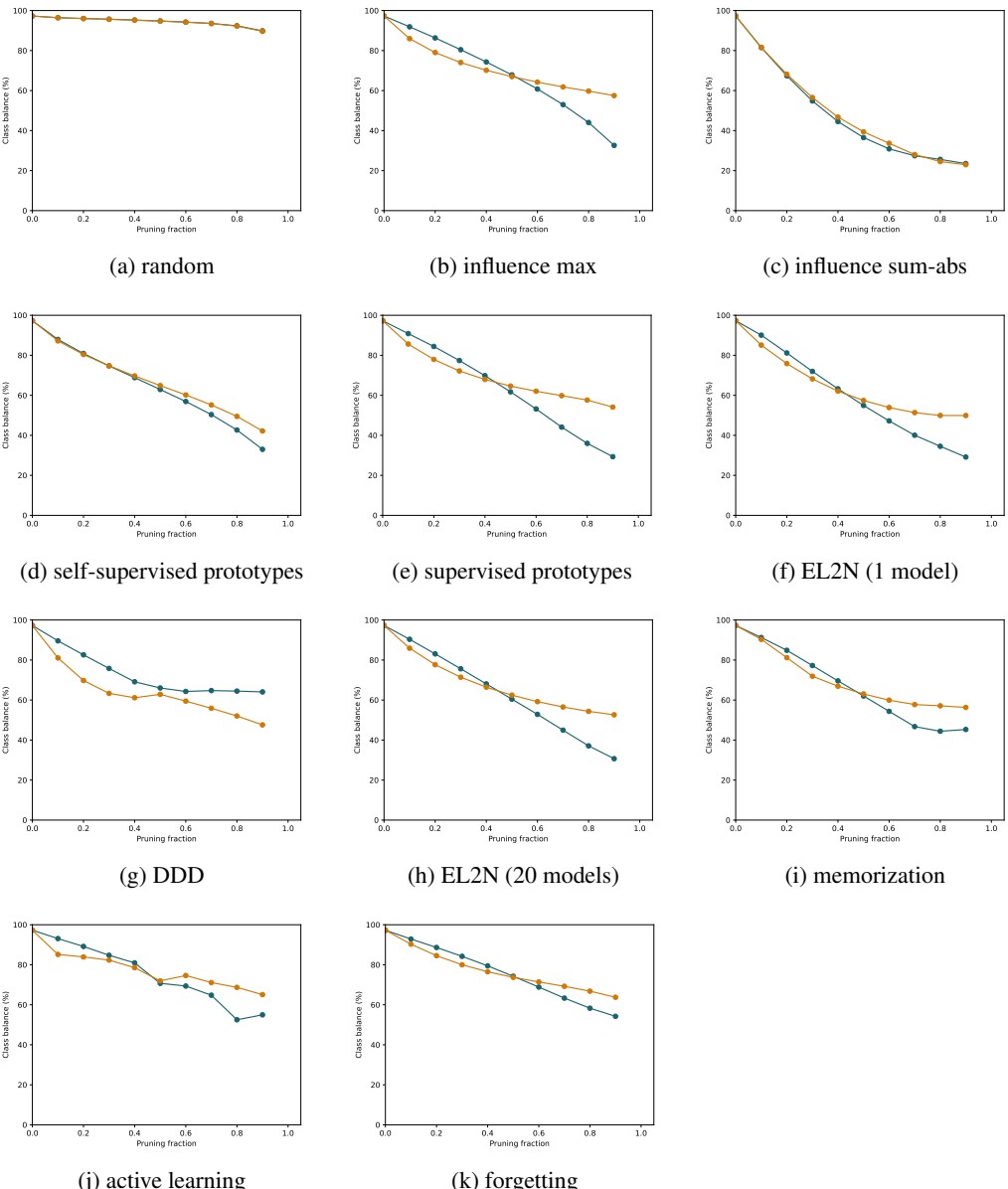

Figure 20: Pruning amplifies class imbalance. With larger pruning fractions, class balance decreases—both when pruning "easy" images (turquoise) and when pruning "hard" images (orange). This effect occurs for all pruning metrics except for random pruning (top left). For details on the class imbalance metric see Appendix H. Pruning fraction refers to the fraction of data pruned, from 0 (no pruning) to 0.9 (keeping only 10% of the data).

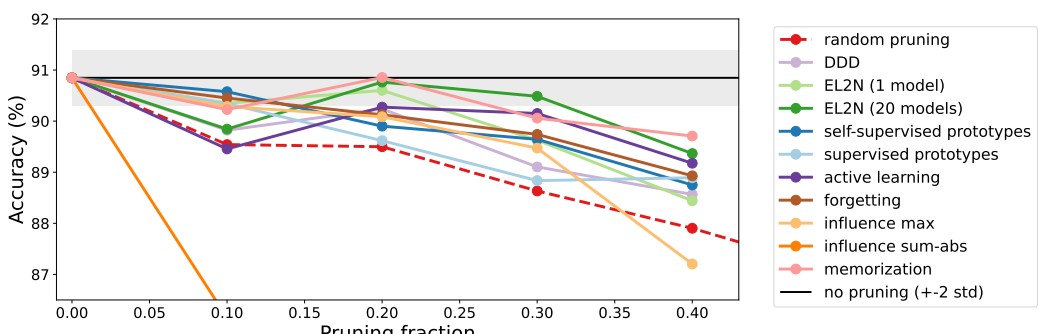

Figure 21: ImageNet-1K pruning results for different metrics, compared against random pruning. 'Pruning fraction' refers to the fraction of the dataset that is pruned away. Results obtained without any class balancing are worse than results with 50% class balancing (Fig. 5BC), confirming the finding that vanilla pruning amplifies class imbalance.

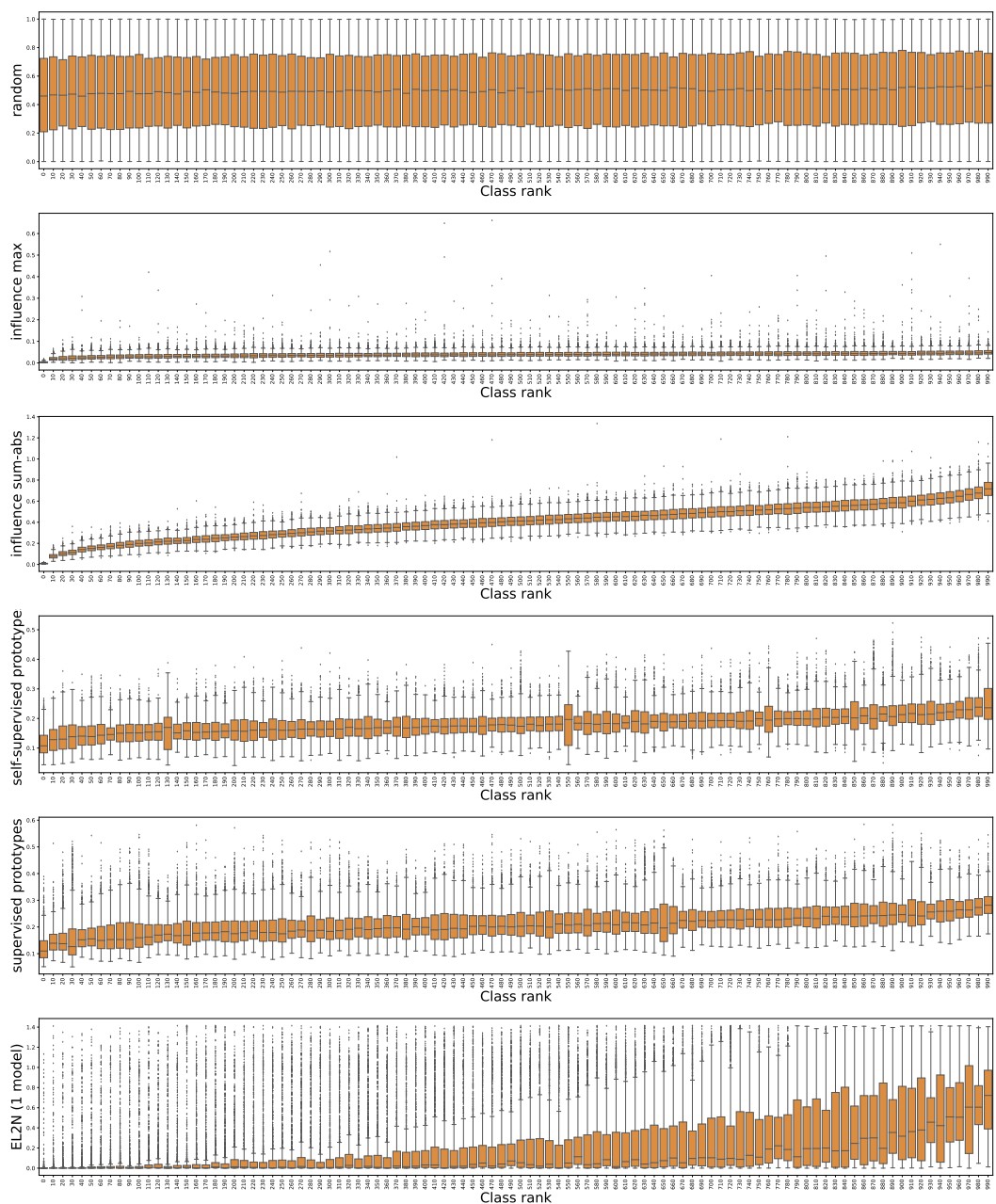

Figure 22: How do ImageNet metric scores differ across classes? Class-conditional score distribution histograms across metrics. For the purpose of visualization, only every $10^{th}$ class is shown.

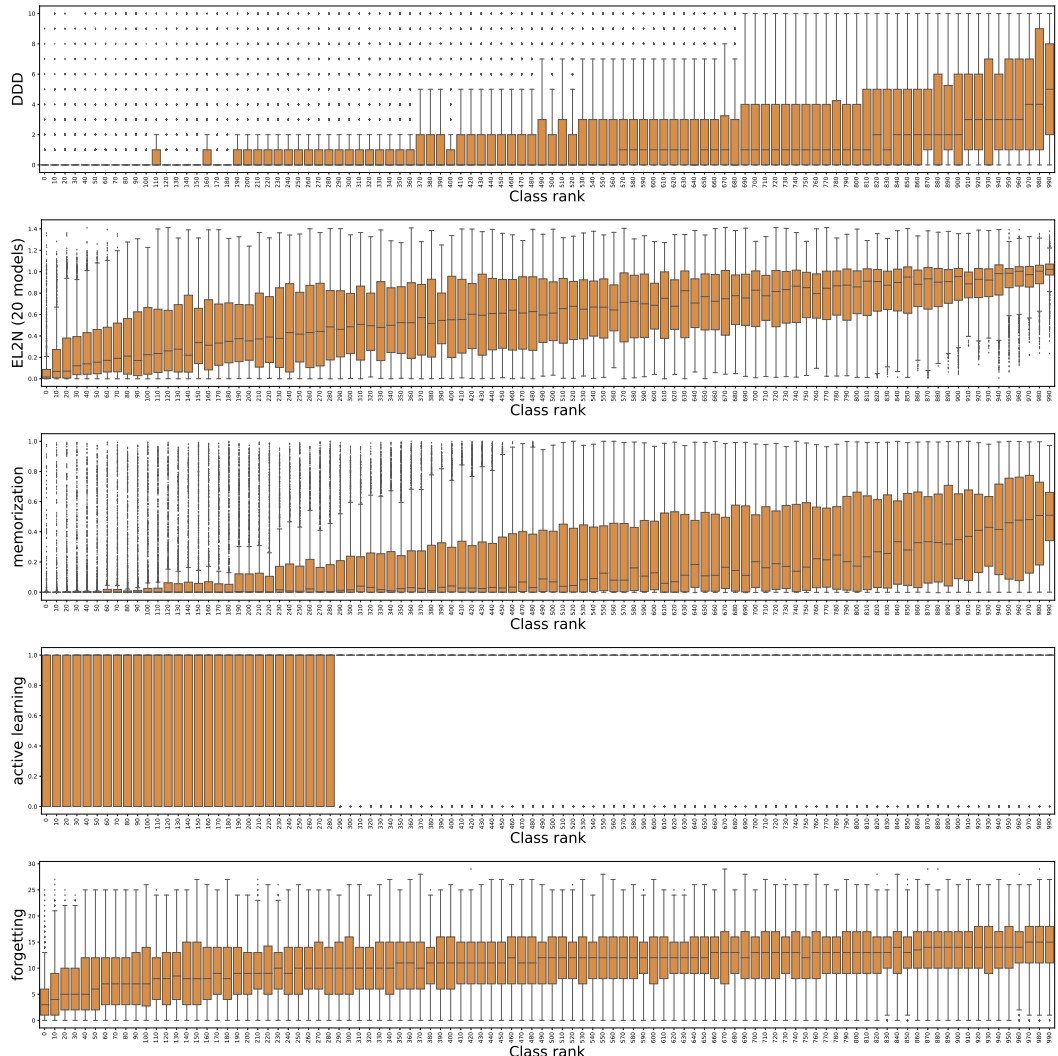

Figure 23: How do ImageNet metric scores differ across classes (continued)? Class-conditional score distribution histograms across metrics. For the purpose of visualization, only every $10^{th}$ class is shown. Please note that the active learning plot is to be taken with a grain of salt since the authors provided a binary score (included/excluded) for ImageNet corresponding to 80% "important" images, thus the scores are either zero or one and a boxplot fit does not apply here.

# I Effect of pruning on class-conditional accuracy and fairness

In order to study the effect of dataset pruning on model fairness, at least with respect to specific ImageNet classes, we compared the class-conditional accuracy of a ResNet-50 model trained on the full ImageNet dataset, versus that of the same model trained on an 80% subset obtained after pruning. We used two supervised pruning metrics (EL2N, memorization) and one self-supervised pruning metric (self-supervised prototypes) for obtaining the pruned dataset. In all three cases, and across all 1000 classes, we found that the class-conditional accuracy of the model trained on a pruned subset of the dataset remains quite similar to that of the model trained on the full dataset (Fig. 24). However, we did notice a very small reduction in class-conditioned accuracy for ImageNet classes that were least accurately predicted by models trained on the entire dataset (blue lines slightly above red unity lines when class-conditioned accuracy is low). This suggests that pruning yields a slight systematic reduction in the accuracy of harder classes, relative to easier classes, though the effect is small.

While we have focused on fairness with respect to individual ImageNet classes, any ultimate test of model fairness should be conducted in scenarios that are specific to the use case of the deployed model. Our examination of the fairness of pruning with respect to individual ImageNet classes constitutes only an initial foray into an exploration of fairness, given the absence of any specific deployment scenario for our current models other than testing them on ImageNet. We leave a full exploration of fairness in other deployment settings for future work.

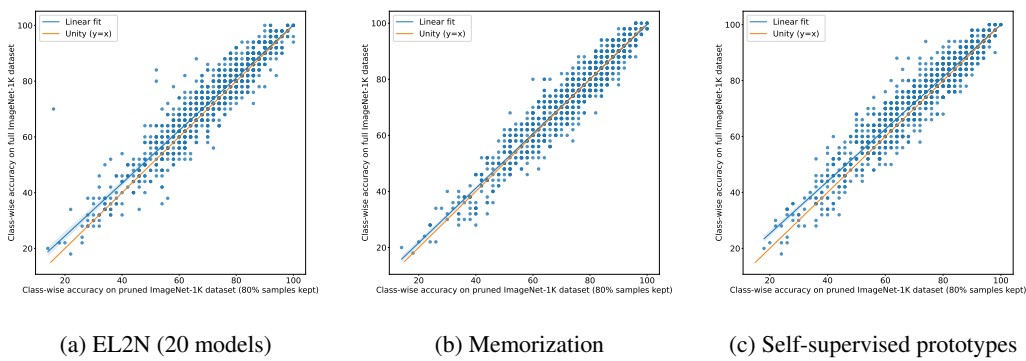

       (a) EL2N (20 models)         (b) Memorization        (c) Self-supervised prototypes

Figure 24: The effect of data pruning on ImageNet test accuracy on individual classes. For all 3 plots, each point corresponds to an ImageNet class, and for all classes, the class specific test accuracy when training on the entire dataset (y-axis) is plotted against the test accuracy when training on the pruned dataset (x-axis).

# J Interaction between data pruning and training duration

Throughout the main paper, our ImageNet experiments are based on a setting where the number of training *epochs* is kept constant (i.e. we train the same number of epochs on the smaller pruned dataset as on the larger original dataset). This means that data pruning directly reduces the number of *iterations* required to train the model specifically by reducing the size of the dataset. However, this simultaneously places two constraints on model performance: not only training on a smaller data set, but also training for fewer iterations.

We therefore investigate how model performance changes if we train longer, as quantified by a *matched iterations factor*. A matched iterations factor of 0 corresponds to the default setting used in the paper of training for the same number of epochs (so that a smaller dataset means proportionally fewer training iterations). In contrast a matched iterations factor of 1 corresponds to training on the smaller pruned dataset for a number of iterations equal to that when training on the larger initial dataset (e.g. when pruning away 50% of the dataset one would train twice as long to match the number of iterations of a model trained on 100% of the dataset). Otherwise the matched iterations factor reflects a linear interpolation in the number of training iterations as the factor varies between 0 and 1.

The results are shown in Table 3 and indicate that training longer does indeed improve performance slightly; however, a matched iterations factor of around 0.4–0.6 may already be sufficient to reap the full benefit. Any matched iterations factor strictly smaller than 1.0 comes with reduced training time compared to training a model on the full dataset.

| matched iterations factor | 0.0 | 0.2 | 0.4 | 0.6 | 0.8 | 1.0 |
|---|---|---|---|---|---|---|
| ImageNet top-5 accuracy | 90.20 | 90.41 | 90.48 | 90.56 | 90.45 | 90.50 |

Table 3: Comparing different settings for training longer when pruning: Performance (top-5 acuracy) when pruning away 20% of ImageNet based on our self-supervised prototype metric. Supervised model training on the pruned dataset performed with a ResNet-50 architecture trained using VISSL without class balancing. Performance tends to increase when training longer (i.e. with a larger matched iterations factor). Interestingly, the benefit of training longer may already be achieved with a matched iterations factor of around 0.4–0.6.

# K  Out-of-distribution (OOD) analysis of dataset pruning

Pruning changes the data regime that a model is exposed to. Therefore, a natural question is how this might affect desirable properties beyond IID performance like fairness (see Appendix I) and out-of-distribution, or OOD, performance which we investigate here. To this end, we use the model-vs-human toolbox [46] based on data and analyses from [47, 48, 49, 50, 51]. This toolbox is comprised of 17 different OOD datasets, including many image distortions and style changes.

In Figure 25a, OOD accuracies averaged across those 17 datasets are shown for a total of 12 models. These models all have a ResNet-50 architecture [52]. Two baseline models are trained on the full ImageNet training dataset, one using torchvision (purple) and the other using VISSL (blue). Human classification data is shown as an additional reference in red. The remaining 10 models are VISSL-trained on pruned versions of ImageNet using pruning fractions in {0.1, 0.2, 0.3, 0.4, 0.5} and our self-supervised prototype metric. A pruning fraction of 0.3 would correspond to "fraction of data kept = 0.7", i.e. to training on 70% of ImageNet while discarding the other 30%. We investigated two different settings: discarding easy examples (the default used throughout the paper), which is denoted as "Best Case" (or BC) in the plots; and the reverse setting, i.e. discarding hard examples denoted as Worst Case, or WC. (These terms should be taken with a grain of salt; examples are only insofar best- or worst case examples as predicted by the metric, which itself is in all likelihood far less than perfect.)

The results are as follows: In terms of OOD accuracy (Figure 25a), best-case pruning in green achieves very similar accuracies to the most relevant baseline, the blue ResNet-50 model trained via VISSL. This is interesting since oftentimes, OOD accuracies closely follow IID accuracies except for a constant offset [53], and we know from Figure 5 that the self-supervised prototype metric has a drastic performance impairment when pruning away 40% of the data, yet the model "BC_pruning-fraction-0.4" still achieves almost the same OOD accuracy as the baseline trained on the full dataset. **The core take-away is: While more analyses would be necessary to investigate whether pruning indeed consistently *helps* on OOD performance, it seems safe to conclude that it does not *hurt* OOD performance on the investigated datasets compared to an accuracy-matched baseline.** (The control setting, pruning away hard examples shown in orange, leads to much lower IID accuracies and consequently also lower OOD accuracies.) For reference, the numerical results from Figure 25a are also shown in Table 4.

Figures 25b, 25c and 25d focus on a related question, the question of whether models show human-like behavior on OOD datasets. Figure 25b shows that two models pruned using our self-supervised prototype metric somewhat more closely match human accuracies compared to the baseline in blue; Figures 25c and 25d specifically focus on image-level consistency with human responses. In terms of overall consistency (c), the baseline scores best; in terms of error consistency pruned models outperform the VISSL-trained baseline. For details on the metrics we kindly refer the interested reader to [46]. Numerical results are again also shown in a Table (Table 5).

Finally, in Figure 26 we observe that best-case pruning leads to slightly higher shape bias as indicated by green vertical lines plotting the average shape bias across categories, which are shifted to the left

of the baseline; while worst-case pruning in orange is shifted to the right. An outlier is the torchvision-trained model in purple with a very strong texture bias; we attribute this to data augmentation differences between VISSL and torchvision training.

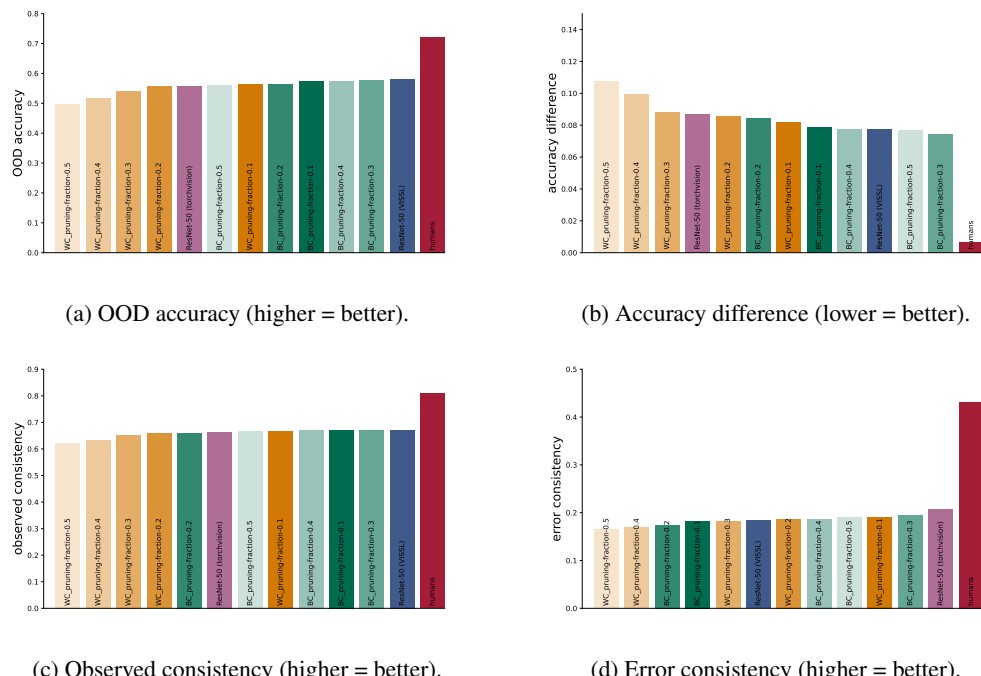

(a) OOD accuracy (higher = better).  (b) Accuracy difference (lower = better).

(c) Observed consistency (higher = better).  (d) Error consistency (higher = better).

Figure 25: OOD benchmark results for different models, aggregated over 17 datasets. All models have a ResNet-50 architecture [52]. Two baseline models are trained on the full ImageNet training dataset, one using torchvision (purple) and the other using VISSL (blue). Human comparison data is shown in red.

Table 4: Benchmark table of model results for highest out-of-distribution robustness.

| model | OOD accuracy ↑ | rank ↓ |
|---|---|---|
| ResNet-50 (VISSL) | **0.582** | **1.000** |
| BC_pruning-fraction-0.3 | 0.578 | 2.000 |
| BC_pruning-fraction-0.4 | 0.574 | 3.000 |
| BC_pruning-fraction-0.1 | 0.574 | 4.000 |
| BC_pruning-fraction-0.2 | 0.565 | 5.000 |
| WC_pruning-fraction-0.1 | 0.565 | 6.000 |
| BC_pruning-fraction-0.5 | 0.560 | 7.000 |
| ResNet-50 (torchvision) | 0.559 | 8.000 |
| WC_pruning-fraction-0.2 | 0.556 | 9.000 |
| WC_pruning-fraction-0.3 | 0.540 | 10.000 |
| WC_pruning-fraction-0.4 | 0.516 | 11.000 |
| WC_pruning-fraction-0.5 | 0.498 | 12.000 |

Table 5: Benchmark table of model results for most human-like behaviour. The three metrics "accuracy difference" "observed consistency" and "error consistency" (plotted in Figure 25) each produce a different model ranking.

| model | accuracy diff. ↓ | obs. consistency ↑ | error consistency ↑ | mean rank ↓ |
|---|---|---|---|---|
| BC_pruning-fraction-0.3 | **0.075** | 0.671 | 0.194 | **1.667** |
| ResNet-50 (VISSL) | 0.077 | **0.671** | 0.184 | 3.667 |
| BC_pruning-fraction-0.5 | 0.077 | 0.666 | 0.190 | 4.000 |
| BC_pruning-fraction-0.4 | 0.078 | 0.670 | 0.187 | 4.333 |
| WC_pruning-fraction-0.1 | 0.082 | 0.666 | 0.191 | 4.667 |
| ResNet-50 (torchvision) | 0.087 | 0.665 | **0.208** | 5.667 |
| BC_pruning-fraction-0.1 | 0.079 | 0.671 | 0.182 | 5.667 |
| WC_pruning-fraction-0.2 | 0.086 | 0.658 | 0.186 | 7.667 |
| BC_pruning-fraction-0.2 | 0.084 | 0.660 | 0.175 | 8.333 |
| WC_pruning-fraction-0.3 | 0.088 | 0.652 | 0.183 | 9.333 |
| WC_pruning-fraction-0.4 | 0.099 | 0.635 | 0.169 | 11.000 |
| WC_pruning-fraction-0.5 | 0.108 | 0.623 | 0.165 | 12.000 |

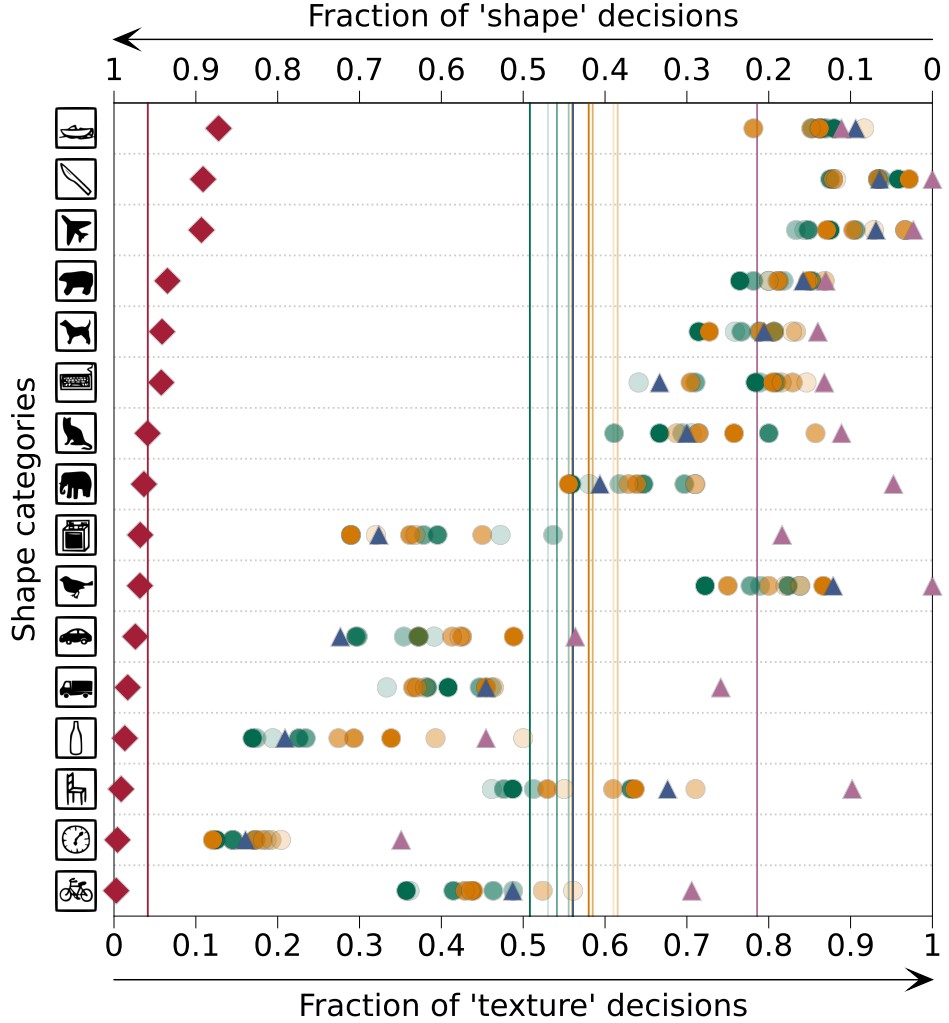

Figure 26: Shape vs. texture bias [49]: category-level plot. Horizontal lines indicate average shape/texture bias; values to the left lean towards a shape bias while values to the right lean towards a texture bias. For details on the plot see [49]; model colors are identical to Figure 25.