# OpenReview forum: "Beyond neural scaling laws: beating power law scaling via data pruning"
_NeurIPS.cc/2022/Conference — NeurIPS 2022 Accept_

### Official Review · Reviewer_D9bn · 2022-07-10

**Rating:** 7
**Confidence:** 4
**Soundness:** 3 good
**Presentation:** 3 good
**Contribution:** 3 good

**Summary:**

The paper considers an interesting question of whether we can "beat" power-law scalings in data with better pruning strategies.
The authors first introduce a theoretical model to study the effect of pruning and its dependence on a few natural parameters (pruning ratio,  initial dataset size), and show that its theoretical analysis matches empirical simulations.
Then, the authors also observe similar behavior on real datasets.
The authors then motivate the importance of a good pruning metric with both theoretical (quantifying the misalignment between student and teacher models) and empirical results (beating power law scaling becomes only possible with a sufficiently good pruning metric).
Then, the authors carry out a thorough evaluation of existing pruning metrics on large scale datasets, and also develop a new metric which does not require any labels
Overall, the work demonstrates the possibility of beyond-power law scaling and evaluates many different pruning metrics in large scale settings.

**Questions:**

Main:
- Why did the authors not reproduce the scaling experiments for Figure 3 using the various metrics (and particularly the newly developed self-supervised metric?) Was it due to computational budget limitations or some other reason?  Given the overall narrative in the paper, I wanted to see if using better pruning metrics lead to better scaling behavior / better pareto optimal frontier, but I couldn't find an experiment along this line in the paper/Appendix besides Figure 7 (which are based on variations of an existing pruning metric rather than the newly introduced metric).
It leaves me wondering if the results were negative, and hence not presented.
In any case, I think demonstrating the impact of the quality of various (existing) pruning metrics on the scaling behavior is much more important than the particular metric introduced in the paper.

Others (some are just curiosity on my part):
- Is it possible to intuitively gleam the the empirical scaling behavior (for the perceptron model) from the equations given in A.2? Or is numerically solving them the only way?
- I liked the experiment on the transfer setting (as it is arguably more relevant in practice), but was a bit disappointed it was only covered briefly. Using your theoretical results/framework, is there something interesting that you can say about this setting?
- From my loose familiarity with the theoretical literature in DL, I understand that people are now capable of giving interesting theoretical results on certain regimes of two layer networks (e.g. NTK). I'm curious, is it possible to extend the perceptron results in the paper to those settings?

**Limitations:**

Yes (none other than my questions/concerns above).

**Strengths And Weaknesses:**

*Strengths*
- Originality: The work is original in its thesis as well its details: theoretical analysis of pruning on different factors and corresponding empirical demonstration of beyond power law scaling. Related work is well cited.
 I also particularly enjoyed the analysis of the misalignment between student and teacher, and how it impacts optimal pruning strategy.
- Soundness: The work is overall quite thorough and sound, with theory matching simulation, and a wealth of empirical experiments. That said, there were a few logical/experimental gaps in and leading into the second part of the paper (semi-supervised metric) that I was not sure if addressed properly (see Weaknesses and Questions).
 (I cannot speak much for the theory parts, as I am not familiar with the seemingly standard tools for theoretical analysis on this topic using statistical physics, but I am confident in my evaluation of empirical results.)
- Clarity: Overall well written and polished. That said, exposition could be improved. In particular, Section 3 is a bit hard to follow due to overuse of mathematical notation and long paragraphs. The logical flow throughout the paper needs some work too (see Weakness 1).
- Impact: Provides an interesting new perspective (are we limited to power law scaling?), and I believe that the theoretical results as well as empirical results will be valuable to future work in the area.

*Weaknesses*
- While the paper overall is insightful, it suffers from a suboptimal narrative/focus as well as missing a few important experiments. The paper seems to be sort of two papers stapled into one, unless read very carefully:
In the first half, the focus is on scaling behavior, and its theoretical and empirical analysis; the second half focus on a better unsupervised pruning metric for larger scale dataset (ImageNet).

There is an important logical transition from the first part to the second part, namely in the last paragraph of Section 3 (reference to App. Fig 7 on showing that beyond power law scaling becomes possible for ResNet18 only with a better pruning metric.)
But this was easy to miss, and I think this logical transition should be emphasized (perhaps by bringing the Figure into the main paper).
So I think this "weakness" of the paper is partially addressable with better prose and organization / focusing of results.

That said, I am left unconvinced by the rigor in which "beyond power law scaling" is claimed in the the results in Fig 7b. First, there are only two data points on the scatterplot below the the power law line, and quite close, so it's unclear to me if it's just noise.
Also, what prevents me from drawing a slightly steeper power law line that also includes these two data points? (it's possible I missed some justification.)
It would be interesting to see if there is some kind of statistical test that can be done to reject the "null hypothesis" (power law scaling) with some confidence.

- The benefit of the new semi-supervised metric was not clear to me due to limited experiments on its scaling (see first question). In particular, I would have liked to see a demonstration that the "pareto optimal" frontier can be expanded by using this new metric.
Also, it is a bit strange to use a semi-supervised model pretrained on ImageNet to prune datasets for ImageNet.
More appropriate would be to take a model pretrained on some other dataset (e.g., CLIP) and using that instead.
Given these, I felt that the self-supervised metric distracted from the strengths of the rest of the paper rather than adding to them.

- A meta-comment related to above two points. I think the paper would be more impactful if it focused on a subset of its results with a tighter narrative.

I hope to adjust to scores appropriately if my concerns/questions are well addressed.

---

> ### Author Response · Authors · 2022-08-02
> **Initial response to reviewer D9bN (Part 1/2)**
>
> Dear reviewer D9bn,
>
> Thank you for taking the time to review our paper. We were glad to hear that you found our work **“original in its thesis as well its details”** and **“insightful”**. Please find a detailed response to your suggestions below.
>
> >_Writing, logical transition:_
>
> Thanks for pointing this out. Consistent with the reviewer’s idea, we indeed wanted to motivate the transition from the first part to the second by making explicit the idea that better pruning metrics should lead to better performance, including the possibility of beating power law scaling.  In fact, Fig. 2 in the main paper was devoted to this idea, exploring both in theory and experiment how imperfections in the pruning metric affect the ability to achieve power law scaling.  Nevertheless, in practice the pruning metrics we currently have can beat power law scaling (Fig. 3BCD), with the potential for improvement on ImageNet.  This latter potential for improvement (as well the search for a cheap metric, unlike the expensive memorization in Fig 3D)  motivated the search for a new cheap metric (unsupervised prototypes in Fig. 5C) that can to some extent beat power law scaling (Fig. 9C).  Thus our logic is in agreement with the reviewer. In the extra space given in the camera ready we will make this logic even more explicit.
>
> >_“That said, I am left unconvinced by the rigor in which "beyond power law scaling" is claimed in the results in Fig 7b”_
>
> Apologies for the confusion - Figure 7 in the original submission (Figure 10 in the revised submission) is intended to highlight a setting where data pruning fails to break power law scaling, when we use an intentionally weak metric. We’ll make this more clear in the camera ready. Our empirical results for beating power law scaling are shown for:
>
> - ResNet18 on SVHN pruned using EL2N (Fig. 3B)
> - ResNet18 on CIFAR10 pruned using EL2N (Fig. 3C)
> - ResNet50 on ImageNet pruned using Memorization (Fig. 3D)
> - Vision Transformer fine-tuned on CIFAR 10 and pruned using EL2N (Fig. 4A)
> - ResNet50 on ImageNet pruned using EL2N (Fig. 9AB)
> - ResNet50 on ImageNet pruned using Self-supervised prototypes (Fig. 9C)
>
> We hope these 7 figure panels provide ample evidence that beating power law scaling with current metrics is feasible. Moreover, we hope that the combination of this empirical observation along with the theory in Fig. 2 indicating that better metrics can lead to better beating of power law scaling, can achieve significant impact on the community by motivating many new works searching for better pruning metrics.
>
> >_“Why did the authors not reproduce the scaling experiments for Figure 3 using the various metrics (and particularly the newly developed self-supervised metric?). The benefit of the new semi-supervised metric was not clear to me due to limited experiments on its scaling (see first question).”_
>
> We performed a scaling analysis of our new self-supervised pruning metric on ImageNet in the original submission. It was presented in Fig. 9C (Fig. 8C in the original submission)\ . There we can see a signature that it beats power law scaling and is somewhat comparable to how memorization beats power law scaling (Fig. 3D).  This is an important result as our self-supervised pruning metric is much less computationally expensive to compute than the memorization metric. Using the extra space in the camera ready we will call out this figure more explicitly from the main text (perhaps even moving it into the main text). We’ll also update the figure by adding a second row zooming in on the right hand tail of the power law, to show the significant number of points that beat the power law.

---

> > ### Comment · Reviewer_D9bn · 2022-08-08
> > **Response to both part 1 and 2**
> >
> > Thank you to the authors for the detailed response.
> > Thanks in particular to the detailed response in part 2 regarding my more (open ended) questions.
> >
> > Some comments:
> > 1. "Figure 7 in the original submission (Figure 10 in the revised submission) is intended to highlight a setting where data pruning fails to break power law scaling"
> > - The purpose of Figure 7B (8B in the new version) as stated is to show that a stronger metric can beat power law scaling in comparison to 7A. I'm still left unconvinced by the rigor at which the conclusion is drawn for Figure 7B (and more broadly in other similar figures), as it seems to be based on just eye balling, without some sort of statistical significance test.
> > To be clear, I don't doubt that better pruning can yield to better scaling based on the wealth of results.
> > However, it's not clear to me whether *"exponential" scaling* can be claimed in some of the results (Figure 8 in new version). If I understand correctly, the trend should be non-linear for it to be considered non-power law scaling, but it seems that some of them trends are just lines with a slightly larger slope/slightly smaller intercept (Figure 8C, for instance; potentially 8AB, but harder to tell without some kind of statistical significance).
> >
> > 2. I agree that Figure 8C (original) is important and would be worth emphasizing more. Thanks for pointing out.

---

> > > ### Author Response · Authors · 2022-08-09
> > > **Second response to reviewer D9bn (Part 1/2)**
> > >
> > > Dear reviewer D9bn,
> > > Thanks for taking the time to assess our initial response! We’re glad we were able to clarify a number of points, and we also realize that we haven’t been able to sufficiently address your concern regarding statistical significance testing to distinguish between exponential and power law scaling. We have therefore, in response to your comment, carefully repeated the experiments in Fig. 7 (now Fig. 10), and have performed a new quantitative analysis which does indeed provide statistical evidence for beating power law scaling. Detailed comments follow,
> > >
> > >
> > > **1.** We thank the reviewer for calling to attention that we should be absolutely precise about two different claims: (1) beating power law scaling; (2) achieving exponential scaling.  We emphasize that we only wish to claim we beat power law scaling in our experiments in practice (hence the title of our paper).  We do not wish to claim necessarily that in our current experiments we achieve exponential scaling at large scale on all benchmark tasks, though on the smaller ones like SVHN we may come close.  We noticed our abstract was not as careful in delineating these two claims. Therefore we toned down our abstract.  In our first sentence in reference to theory we modified it to the following:
> > >
> > > *"Here we focus on the scaling of error with dataset size and show how in theory we can break beyond power law scaling and potentially even reduce it to exponential scaling instead."*
> > >
> > > i.e. in this sentence we have removed any reference to what we do in practice, though we do show in practice that we can achieve exponential scaling for perceptron learning.
> > > Then in the first sentence about practice we remove all reference to exponential scaling. The new sentence in the abstract reads:
> > >
> > > *"We then test this improved scaling prediction with pruned dataset size empirically, and indeed observe better than power law scaling in practice on ResNets trained on CIFAR-10, SVHN, and ImageNet."*
> > >
> > > We hope this more precisely separates the two claims in the abstract.
> > >
> > >
> > > **2.** It would be very compelling to identify a high-powered statistical test which verifies exponential scaling in the test error with dataset size. The most straightforward approach would be to compare power law and exponential curve fits to the Pareto frontier, and show that the frontier is better fit by an exponential. However, this is not easy to do in practical settings (where each data point involves training a deep neural network to convergence) because out of 100 data points only a few actually fall on the Pareto frontier (3-5), and hence it is not easy to compare the goodness-of-fit of power-law and exponential curve fits to the frontier, which each involve fitting three parameters (constant offset, multiplicative constant, and exponent) to 3-5 data points. We are currently running experiments to much more finely sample the parameter space of $\alpha_\text{prune}$, $f$, to trace out the frontier at a higher resolution, so that we will be able to perform this analysis. These experiments will not be ready by the end of the rebuttal period but we will include them in the revised manuscript.
> > >
> > > Fortunately, even without running additional experiments we can perform a (different) test providing evidence for exponential scaling with greater statistical significance by taking advantage of the fact that the Pareto frontier is an envelope traced out by a *family* of curves, each of which follows the three predictions from our quantitative theory: (1) each curve transiently scales faster than the random-pruning power law, (2) the slopes of these curves *become steeper* as $f$ decreases, and (3) each curve eventually settles back onto the random-pruning power law.  Importantly, exponential scaling is achieved in our theory because the slopes of successive curves *become steeper* as f decreases; hence the frontier must scale faster than any power law.
> > >
> > > We verify these predictions quantitatively for the SVHN experiments in Fig. 7B (original), and find that the slopes of successive learning curves **do** indeed become steeper as $f$ decreases when the pruning metric is good (slopes $28.2,32.7,35.7,43.9,47.8,63.3,75.0$ for $f=1.  , 0.6 , 0.36, 0.22, 0.13, 0.08$), but **do not** become steeper when the pruning metric is poor (slopes $25.2,19.4,18.0,16.4,37.5,23.7,34.6$) - indicating that a good pruning metric can allow for better-than-power law scaling. We will repeat these analyses for the other scaling experiments and include them in the revised manuscript.

---

> > > > ### Comment · Reviewer_D9bn · 2022-08-09
> > > > **response**
> > > >
> > > > I appreciate the authors' thorough response.
> > > >
> > > > "These experiments will not be ready by the end of the rebuttal period but we will include them in the revised manuscript."
> > > >
> > > > I look forward to seeing these results.
> > > >
> > > > In light of authors' engaged response and improvements to the paper, I have updated my score.
> > > > Overall, I think the paper would still benefit from making the writing / content in the main paper a little less dense and making it easier for the reader to follow.

---

> > > > > ### Author Response · Authors · 2022-08-09
> > > > > **Second response to reviewer D9bn**
> > > > >
> > > > > Thanks again for all the valuable feedback - we will continue updating the manuscript as new results come in!

---

> > > ### Author Response · Authors · 2022-08-09
> > > **Second response to reviewer D9bn (Part 2/2)**
> > >
> > > 3. The original Fig. 7a was an experiment intended to demonstrate that when the pruning metric is poor, the test error is lower-bounded by a region of unattainable test error with an envelope taking the shape of a power law.  7b was simply meant to show that a good pruning metric could achieve test error lower than this lower bound - not that the shape of the Pareto frontier beats the power law, which would require carefully fitting a power law to the random pruning learning curve and identify and subtracting off the asymptotic test error $E_\infty$ as in Fig. 3B. However, we admit that this was unclear on our part and the reviewer’s comments are completely reasonable: a good pruning metric *should* break power-law scaling. So we have carefully repeated the experiments in Fig. 7, fitted a power law to the random-pruning learning curve and subtracted off $E_\infty$. The resulting figure (new Fig. 10) we believe more clearly makes both points: a poor metric leads to a power law lower-bounded test error, and a good metric can break power-law scaling. (Experiments with a 4-epoch probe have not finished running, but we will continue to update the manuscript as they come in). We thank the reviewer for this insightful comment as we think it has made the manuscript much clearer.
> > >
> > > We would appreciate it if you could let us know whether we have been able to address your remaining concerns!

---

> ### Author Response · Authors · 2022-08-02
> **Initial response to reviewer D9bn (Part 2/2)**
>
>
> > *Is it possible to intuitively gleam the empirical scaling behavior (for the perceptron model) from the equations given in A.2? Or is numerically solving them the only way?*
>
> We do not have an intuitive explanation for the scaling behavior for the perceptron model beyond the idea that as you go to larger dataset sizes and prune more aggressively, the examples that survive pruning are increasingly concentrated near the correct decision boundary (i.e. the equator in Fig. 1B) and thereby yield finite information gain per each additional pruned example (Fig. 1F).  Thus each additional example multiplicatively reduces the student perceptron’s uncertainty about the ground truth teacher perceptron’s true weights.  This successive multiplicative reduction in uncertainty leads to at least exponential scaling of error with the number of examples in the pruned dataset.
>
> We also believe there is an intuitive explanation for the phase transition as data size grows where for small datasets, it is better to keep easy examples, but better to keep hard examples for large datasets. Intuitively, in the limited data regime, it is challenging to model outliers since the basics are not accurately captured; hence, it is more important to keep easy examples so that the model can get to moderate error. However, with a larger dataset, the easy examples can be learned without difficulty, making modeling outliers the fundamental challenge. In the extra space given in the camera ready, we will add this intuition.
>
> > *I liked the experiment on the transfer setting (as it is arguably more relevant in practice), but was a bit disappointed it was only covered briefly. Using your theoretical results/framework, is there something interesting that you can say about this setting?*
>
> This is a very interesting question!  In order to say something about this, we would need a model for the difference between the pre-training task and the subsequent downstream task to which the transfer will occur.  One interesting and simple model in the perceptron setting would be that the pre-training task is defined by a dataset generated by a first teacher perceptron, while the downstream task is defined by a second teacher perceptron.  The difference between the two is quantified by the angle between the two perceptrons.  Pruning is done via margin with respect to the first teacher.  Interestingly this is simply a reinterpretation of Fig. 2 where theta is now thought of as the angle between the two teacher perceptrons rather than the angle between the teacher and the student.  This reinterpretation of Fig. 2 indicates that pruning via margin with respect to a pre-training task will yield a small dataset suitable for training on a downstream transfer task  as long as the downstream transfer task and the pre-training task are similar.  We will add a discussion of this in the supplementary material.
>
> > *From my loose familiarity with the theoretical literature in DL, I understand that people are now capable of giving interesting theoretical results on certain regimes of two layer networks (e.g. NTK). I'm curious, is it possible to extend the perceptron results in the paper to those settings?*
>
> This is a very interesting idea and a promising direction for future research. Though it may be outside the scope of this work, we believe the analysis we have performed could be extended to the NTK setting and may yield important new insights and potentially better pruning strategies. In particular, a key consideration in the kernel setting is anisotropy in the data, induced by the geometry of the kernel, and characterized by the eigenvalue spectrum of the kernel. One could imagine that in this setting the “pruning score” of a training example would be a function not only of its overlap with the target function (margin), but also its overlap with the eigenfunctions of the kernel. For example, a hard example in the kernel setting might not only have small overlap with the teacher, but also small overlap with the large-eigenvalue eigenfunctions of the kernel. This new pruning could be thought of as an “effective margin” not in Euclidean space, but in a geometry induced by the kernel, and may differ importantly from the naive margin prediction. This theory could be tested in realistic settings by computing pruning scores using a metric induced by the empirical NTK of a trained network, and could conceivably lead to even better pruning performance. We thank the reviewer for this interesting idea and are excited to pursue it further.
>
> We would appreciate it if you could let us know whether we have been able to address your concerns and suggestions!

---

### Official Review · Reviewer_QtTy · 2022-07-11

**Rating:** 8
**Confidence:** 2
**Soundness:** 2 fair
**Presentation:** 3 good
**Contribution:** 3 good

**Summary:**

This paper explores neural scaling laws and looks at the power law scaling behavior of error with dataset size. While power law scaling has shown that larger dataset sizes can improve accuracy, it comes at a significant cost as power scaling is extremeley weak and unsustainable. This paper demonstrates that exponential scaling is in-fact possible both in thoery and practice, provided we chose the right dataset to train on. Power law scaling indicates that many training examples are hihgly redundant. Thus, one should be able to train on a much smaller sized dataset. This requires data selection or pruning strategies. Prior work has shown this. However, prior work lacks theoretical and empirical rigor. It leaves many questions unanswered like when is data pruning possible, how to do it at scale, can it beat power law scaling and can pruning be done using unlabeled dataset.

This paper answer these questions by by first unify existing literature on dataset pruning, statistical mechanics of learning and observations of neural scaling law to provide an analytical theory of data pruning. This theory and its associated empirical results help answer many of the questions asked above including showing how to achieve exponential scaling with dataset. They show that their theory is applicable in real world use cases also, like CIFAR10 and SVHN. To scale to ImageNet sized dataset, it is important for the data pruning metric to be cheap to calculate and also not require labeled dataset. In order to achieve this objective, they propose a clustering based unsupervised data pruning metric that performs equally well or better than existing data pruning techniques while still providing exponential scaling capabilities .


**Questions:**

Questions -

1. In Figure 1A, if one draws a vertical line for a fixed alpha_prune, does that represent accuracy/test error at iso-dataset size for various experiments? If yes, how is that possible given that alpha_tot is fixed. If no,

2. Figure 1E’s description suggests a blue curve, 1E does not have any blue curve.

3. Its unclear reading the figure 1 and the description, as to how to achieve exponential scaling. Given a fixed alpha_total and ability to prune by a fraction f, is the idea to decrease f such that for large f (less training data), use easy examples but for small f (more training data) use hard examples?

4. In Figure 4A, a vertical like at x = 10^3 interesects different lines representing different fraction of original dataset. Will all those points be iso-size? If yes, how, given that they all start from the same dataset size.

5. Overall, given that this work assumes a large dataset is always available. Given a large dataset, one can train on a small fraction and achive a better accuracy using intelligent data pruning techniques. This makes it an approach towards efficient training (mentioned in line 336). Is this understanding correct? If yes, this is approach becomes a bit orthogonal to neural scaling law work whose aim is to understand how much resources (data, compute etc) will be needed to improve the accuracy of the model. Neural scaling laws guide investments into various accuracy improvement strategies.

**Ethics Review Area:**

["I don’t know"]

**Limitations:**

Have the authors adequately addressed the limitations and potential negative societal impact of their work? - Yes

**Strengths And Weaknesses:**

Strengths -

1. The paper targets an important problem of data quality and its impact on accuracy

2. The paper evaluates its claims on a wide range of datasets and backs them with strong theoretical foundation

3. Insights around the kind of data to prefer at what scale were interesting and novel. Preferring easy samples in low resource regime and hard samples in large dataset regime was something that was not quantified before in prior work

4. They show a simple, easy to use and unsupervised data pruning technique to measure the hardness of a sample.  The technique further scales to a larger dataset. The simplicity of the technique makes it a really strong contribution

5. This is the first paper that shows exponential scaling is possible, given a high quality dataset. Where quality refers to how much does a model learn from each sample in the dataset. These results can further fuel more research in dataset creation.



Weakness -

The results in the paper are really difficult to parse, making it very difficult to match the figures with the claims in the paper. I do believe that the paper requires some work in its presentation to make it readable to a general audience. Section 3’s Figures which forms the foundation of the claims in this paper are not well explained. Below are some instances

1. Section 3 introduces alpha_tot as a ratio of total training examples (P) and number of parameters in the model (N). alpha_pruning is basically f*alpha_tot where f represents the fraction of the  examples selected. Using this knowledge (as established in lines 159-165) seems incomplete to parse Figure 1. For example, in Figure 1A x axis represents alpha_prune as an increasing variable. That means, f is decreasing. However, the figure also has datapoints associated with different f for the same alpha_prune. Given that alpha_tot is constant, how is that possible?

There are other similar issues in readability (See questions 1,2,3) that make it really difficult to understand the claims. This makes parsing the next section that shows these results on CIFAR-10 and SVHN equally hard.

2. The technique described in the paper is post-hoc - Given a dataset, how do you prune it to a few high quality samples to reduce the number of samples to train on. This can potentially reduce the training time of the model (line 336). But this aspect was not discussed in detail in the paper. Given that this paper does not discuss how to build a high quality dataset, its impact can be limited.

---

> ### Author Response · Authors · 2022-08-02
> **Initial Response to Reviewer QtTy (Part 1/3)**
>
> Dear reviewer QtTy,
>
> Thank you for taking the time to review our paper. We’re glad you found the paper to **“target an important problem”**, with **“strong theoretical foundation”** and **“interesting and novel insights”**.
>
> We apologize for issues regarding readability and figure accessibility, and have taken your feedback as an opportunity for improvement. Please find detailed responses to your suggestions below.
>
> > *Parsing Fig. 1:*
>
> We apologize that Fig. 1 was difficult to parse. We have updated the Figure 1 caption and legends to make it more readable. We also provide an explanation here and we will use the extra space in the camera ready to incorporate aspects of this explanation.
>
> We begin by noting that many of the reviewer’s questions seem to stem from a single assumption that alpha_total is held constant, which it is not. The reviewer’s questions are completely understandable given this assumption. We hope the following explanation clarifies things.  With this clarification we hope the reviewer will consider raising his/her score especially as his/her score is an outlier low score of 4 (the other reviewers were 6,7,9).
>
> The reviewer is correct that there are 3 relevant quantities obeying a single relation:
>
> alpha_total = total initial dataset size (per number of parameters)
> f = fraction of examples kept during pruning
> alpha_pruned = size of pruned dataset (per number of parameters)
>
> These 3 quantities obey the relation: alpha_pruned = f * alpha_total
>
> Now in Fig. 1 we jointly varied *both* alpha_total and f  (i.e. we did not hold alpha_total fixed) and we computed the test error E as a joint function of alpha_total and f.   However we plotted E as a function of alpha_pruned, which is the size of the pruned dataset used to train subsequent models and therefore the right quantity to plot final test error against.  So training set size alpha_pruned is the horizontal axis and test error E is the vertical axis.  Now for any fixed alpha_pruned there are many ways to achieve it by jointly varying both alpha_total and f subject to  alpha_pruned = f * alpha_total. This is why at any fixed point on the x-axis there are many curves of different colors corresponding to different f.  For example, for an alpha_pruned of 2, one could achieve this by starting with a small original dataset of alpha_total=2 and keeping all examples (f=100% corresponding to the black curve in Fig. 1A) or starting with a large dataset of alpha_total=20 and keeping only f=10% of the examples (corresponding to the lightest green curve in Fig. 1A).  We explicitly jointly vary both alpha_total and f because we wish to understand how pruning behaves as a joint function of the size of the original dataset (alpha_total) and the aggressiveness of pruning (f).  This same method of varying both alpha_total and f, but plotting results as a function of alpha_pruned that was carried out in theory in Fig. 1 was also carried out in Fig 2,3,4,9.  In benchmark datasets like SVHN, CIFAR10, and ImageNet, we varied alpha_total by randomly sub-selecting the original data to create an initial dataset, and then pruning this initial dataset to achieve a training dataset of size alpha_prune.
>
> > *“The technique described in the paper is post-hoc - Given a dataset, how do you prune it to a few high quality samples to reduce the number of samples to train on. This can potentially reduce the training time of the model (line 336). But this aspect was not discussed in detail in the paper. Given that this paper does not discuss how to build a high quality dataset, its impact can be limited.”*
>
> Data pruning can indeed reduce the time it takes to train subsequent models simply by reducing the amount of training data. We regret that we didn’t discuss this practically relevant aspect in much detail in our initial submission.  In response to feedback from you and other reviewers, we have now added a detailed investigation of the potential compute savings resulting from different pruning settings to the Appendix C.  The basic idea is that when we train on a pruned dataset, we train for the same number of epochs on the smaller pruned dataset as we do on the larger unpruned dataset.  Thus training time, in terms of number of iterations for forward and backward passes, is directly proportional to dataset size.  Smaller pruned datasets therefore lead to faster training.  We have a discussion of this in the supplementary material section I, titled: “Interaction between data pruning and training duration.”
>
> (response continued in Part 2)

---

> ### Author Response · Authors · 2022-08-02
> **Initial Response to Reviewer QtTy (Part 2/3)**
>
> Regarding the question of how to build a high quality dataset, we have outlined our vision in the discussion section (“Outlook: towards foundation datasets”): Currently, there is a clear trend towards giant unlabeled datasets scraped from various web sources, without much quality or redundancy control. The immense scale of those datasets strongly increases compute requirements. This is where our method comes in: being able to select which dataset samples to use (and pruning away redundant ones) using a good self-supervised pruning metric can go a long way in increasing dataset quality. This approach is indeed post-hoc, i.e. it starts with a (potentially large) dataset of mixed-quality samples, but weeding out non-useful examples via dataset pruning will nonetheless increase the average quality or “usefulness” of dataset samples, and might thus be an important contribution towards building high-quality “foundation datasets”. Our work already shows a proof of principle: we can find a data-subset of ImageNet that is only about 80% of ImageNet, and training on this subset achieves the same test accuracy as training on the entire dataset. We are unaware of any other results that show this using a computationally cheap pruning metric like our self-supervised prototypes. Furthermore, our theory shows that data-set pruning will be even more successful for larger initial datasets and more aggressive pruning. We hope this answers your question - if not, kindly let us know!
>
> Answers to specific questions:
>
> > 1. *In Figure 1A, if one draws a vertical line for a fixed alpha_prune, does that represent accuracy/test error at iso-dataset size for various experiments? If yes, how is that possible given that alpha_tot is fixed. If no,...*
>
> Correct, as we explained above in our answer about parsing Fig. 1, a vertical line represents a fixed pruned dataset size. This is possible because when using synthetic data, we have the luxury of generating as much data as we want (varying \alpha_{tot}) to investigate how data pruning scales to ever-larger dataset sizes. For fixed \alpha_{tot} see Fig. 3A, where the solid lines represent a fixed dataset size (in analogy to a fixed dataset like CIFAR-10 or ImageNet) while the dotted lines extrapolate the behavior to larger dataset sizes (indicating how we would expect scaling to behave if we could draw more samples from CIFAR-10 or ImageNet).
>
> > 2. *Figure 1E’s description suggests a blue curve, 1E does not have any blue curve.*
>
> Thank you for pointing this out. The curve is purple and we have corrected this in the manuscript.
>
> > 3. *Its unclear reading the figure 1 and the description, as to how to achieve exponential scaling. Given a fixed alpha_total and ability to prune by a fraction f, is the idea to decrease f such that for large f (less training data), use easy examples but for small f (more training data) use hard examples?*
>
> The key idea again is that alpha_total is not fixed in Figure 1, as explained in our “Parsing Fig. 1” explanation.   Also f is the fraction of examples kept so all else held equal, reducing f would reduce training set size.   Fig. 1 shows the possibility of finding a sequence of datasets of increasing size alpha_prune obtained by increasing alpha_total (i.e. collecting more data) but decreasing f (keeping less of it) such that test error E as a function of pruned dataset size alpha_prune falls of exponentially with alpha_prune.  The practical setting in which this is relevant are scenarios in which new data is collected daily (i.e. new text generated on the web each day, i.e. new interactions generated on a social network each day).  In such a setting data grows without bound over time and there is no sense in which alpha_total is a fixed quantity.  In such settings we can become increasingly selective about which data we train on (i.e. reduce the fraction of data f that is kept as a function of all data alpha_total generated so far).  Then alpha_prune = f * alpha_total is the actual training set size.
>
> (response continued in Part 3)

---

> ### Author Response · Authors · 2022-08-02
> **Initial Response to Reviewer QtTy (Part 3/3)**
>
> > 4. *In Figure 4A, a vertical like at x = 10^3 intersects different lines representing different fraction of original dataset. Will all those points be iso-size? If yes, how, given that they all start from the same dataset size.*
>
> Yes, all points intersected by a vertical line correspond to iso-size pruned training sets.  The key idea is that each of the 10 points on a curve of a constant color in Fig 4a (and other points on a curve of constant color in Figs 1,2,3) represents the test error obtained by starting from 10 *different* initial dataset sizes. The 10 differently sized initial datasets are generated by randomly subsampling CIFAR-10. Each dataset is then further *pruned* by keeping only a fraction f of the hardest (smallest-margin) examples. The fraction f retained is indicated by the color of the curves. For example, along the black curve corresponding to f=100% no examples are pruned, so the point at x=10^3 corresponds to a training set which is generated by randomly selecting 1,000 examples from CIFAR-10. For the blue curve corresponding to f=20%, the point at x=10^3 corresponds to a training set which is constructed by first randomly selecting 5,000 examples from CIFAR-10, and then *pruning* to keep only the 1,000 hardest examples among these.
>
> > 5. *“Overall, given that this work assumes a large dataset is always available. Given a large dataset, one can train on a small fraction and achive a better accuracy using intelligent data pruning techniques. This makes it an approach towards efficient training (mentioned in line 336). Is this understanding correct? If yes, this is approach becomes a bit orthogonal to neural scaling law work whose aim is to understand how much resources (data, compute etc) will be needed to improve the accuracy of the model. Neural scaling laws guide investments into various accuracy improvement strategies.”*
>
> We do not view our approach as orthogonal to neural scaling law work. Indeed, the central objective of our work is to understand how data pruning strategies can improve (or even beat!) the observed power law scaling of accuracy with data and compute (please see our new section on compute, Appendix C). Our results point to the tantalizing possibility that the same accuracy may be obtainable with exponentially fewer resources (data + compute), if data is carefully curated. The reviewer is absolutely correct that neural scaling laws guide investments into different accuracy improvement strategies. As an example, DeepMind’s Chinchilla model was able to outperform the previous state of the art Gopher, without using any more compute, by noticing an improved neural scaling law with data relative to what had previously been shown. If we can show that neural scaling laws can be not only improved, but beaten, we believe it may radically reshape how we guide investments and allocate resources when training large models.  Indeed our work suggests that much gains might be achieved by carefully curating data.
>
> Again, thanks for your comments & your time! We would appreciate it if you could let us know whether we have been able to address your suggestions and concerns.

---

> > ### Comment · Reviewer_QtTy · 2022-08-08
> > **Thank you for your responses, I have updated my score based on the rebuttal**
> >
> > Thank you for your detailed responses and opting to update descriptions of the figure to better guide a reader. With a better understanding of the figures, I am able to correlate the text with the empirical evidence. Very excited to see the impact of this work.

---

> > > ### Author Response · Authors · 2022-08-09
> > > **Second response to reviewer QtTy**
> > >
> > > Thank you again for your comments - they have greatly improved the clarity of the manuscript - and for your enthusiasm regarding our work!

---

### Official Review · Reviewer_ycTR · 2022-07-11

**Rating:** 8
**Confidence:** 4
**Soundness:** 3 good
**Presentation:** 3 good
**Contribution:** 4 excellent

**Summary:**


Motivated by the unscalable computational costs inherent to the power-law nature governing neural scaling laws in data, parameters and compute, and the insight that this powerlaw implies high redundancy (diminishing informational gain) among increasing amounts of samples the authors systematically evaluate the potential of data pruning theoretical and experimentally through the prism of data scaling properties.

The authors claim that reducing the generalization functional dependency in data to an $\textit{exponential}$ dependency is expected both as predicted by an analytic theory they derive and observed in practice through experimentation --- when the data is pruned with a 'high-quality' data pruning metric.

$\textbf{Theory}$

The paper establishes a analytic theory of data pruning in the perceptron teacher-student setting --- at the limit of data $P \rightarrow \infty$ and perceptron dimension $N \rightarrow \infty$ jointly large, with a finite number of samples per dimension $\alpha_{tot} = \frac{P}{N}$). The pruning metric is the per-sample margin taken from an 'student-probe' (oracle) preceptron such that a fraction $f$ of the data is retained ranked by the metric, resulting in a smaller (pruned) number of retained samples per dimension $\alpha_{prune}=f\alpha_{tot}$. The theory predicts the generalization error of the student (a newly trained perceptron) as a function of number of samples per dimension $\alpha_{prune}$, and the quality of the oracle, in this case the angle $\theta$ of deviation of the student-probe perceptron from the teacher (the theory is derived for a general distribution along the probe-student decision boundary normal direction, and analytically solved for the case of ranking by margin, in the case of an originally isotropic Gaussian data distribution, prior to pruning).

The theory has several insightful and important predictions to the behavior of the generalization error of the student depending on the metric strength:

$\textit{Optimal pruning: }$
* When data is 'abundant', restricting the samples to 'hard' (small margin) samples results in superior performance relative to no pruning --- i.e. in the sense that for the same number of samples per dimension, lower generalization error is attainable. When data is 'scarce', such restriction is detrimental in terms of generalization error as a function of training samples.

$\textit{Scaling functional dependency: }$
- the error scales as a function of number of samples, for any given (pruned) dataset initially as a better than power-law (claimed exponential) phase which then reverts to a (universal) powerlaw.
- examining the Pareto of optimal pruning ( as a function of initial dataset size and pruning degree ) better than powerlaw scaling is achievable limited by the quality of the pruning metric. Specifically, there is a lower bound in which the $\textit{pareto}$ will transition to powerlaw which is associated with the maximal pruning (minimal data fraction $f_{min}$) of the dataset set by the student-probe quality.
 decrease in error

$\textit{Information theoretic, per sample contribution:}$

The theory further predicts the information gain per sample added per dimension $I(\alpha_{prune})$

$\textbf{Experiments}$

The theoretical (asymptotic) predictions are shown to hold with excellent agreement in the $\textit{finite}$ dimension ($N=200$) perceptron learning case trained on data $P_{prune} = N\alpha_{prune}$ derived as different difficulty subsets of total data sizes of $P_{tot} = P_{prune}/f$ with various oracle qualities (ideal, where the oracle is the teacher $\theta =0$ and non-ideal $\theta >0$).

For real data on CIFAR10, SVHN and Imagenet:

* Hard sample preference in the abundance of data (and vice versa) is shown to hold also in practice (CIFAR10)
* The 'signatures' of initial steeper than powerlaw pareto frontiers in real world are shown for all datasets, pruned according to previous art and further developed (self supervised) metrics
* the (lower limit transition to powerlaw and associated maximal pruning) limitation imposed by weak oracle / 'student probe' and how that limitation is pushed with the improvement (further training of the student probe) is shown (with SVHN results in the appendix).

* Transfer learning (CIFAR10):
Analysis of pruning the up and downstream datasets is performed with superior performance than full data attained.

$\textbf{pruning metrics benchmarking and self-supervised metric introduction}$

Recognizing the criticality of the pruning metric (as it sets a lower limit on the transition to powerlaw and max beneficial pruning factor $f$), the authors systematically analyze previous methods at imagent scale. Evaluating both their rank-agreement and relative performance down to $f=0.6$ .

All above methods require labels and the best performing are computationally intensive. The authors thus construct a self supervised (and supervised version) method and demonstrate its comparable (to best method) performance down to $f=0.8$ (and to memorization-metric-methid down to 70%). The method relies on a (embedding clustering based) self-typicality estimate  from strong pre-trained self-supervised model on Imagenet (SWaV).





**Questions:**

Questions/Requests/Recommendations embedded above, denoted (R#$\alpha$)

**Limitations:**

Yes, the authors engage deliberately and adequately with this subject! Specifically uncovering and investigating pruning class imbalance effects --- for both previous methods benchmarked and developed methods --- and addressing it.

There are additional areas of investigation which are active open questions associated more broadly with the line of research dealing with data pruning and are not specific to this paper --- and should thus be, in this reviewers view, dealt with in other, future, work.

Such topics include vulnerability implications of training on pruned data. Specifically, it is an open question (to this reviewer's knowledge) how training on pruned data affects robustness to both naturally occurring phenomena (e.g. OOD performance) and bad actor actions (e.g. adversarial attacks, model stealing).


**Strengths And Weaknesses:**

ֿ$\textbf{Pre-face:}$
It is this reviewer's view that this is a paper of significant insight, methodological implications and multiple important contributions to the field. An exciting paper for its bridge of phenomena (generalization scaling laws, data pruning) and theory drawing and expanding on classical (statistical mechanics perceptron learning) theory, with inroads to practical understanding of implications, role and potential of data pruning, as well as practical contribution at large (Imagenet) scale of pruning (self supervised based) metrics critical to (limiting of) said potential realization.

With that said, the main improvements recommended are softening/solidifications of some claims - primarily by corresponding with previous art and adding further analysis where merited.

Addressing the below recommendations/requests would improve the paper materially and would likely elevate this reviewer's recommendation further.

Details below:

$\textbf{Strengths}$

* the expansion of classical — statistical mechanics perceptron learning — theory, to the data pruning setting is elegant and insightful.
Notable is the excellent agreement of the $\textit{asymptotic}$ theory in the $\textit{finite}$ dimension (N=200) perceptron regime used throughout the paper.
It sets the stage for a better understanding, through its predictions which are consistent with observations in real data over both prior art and further introduced pruning methods.

The predictions are, at least, consistent with observations for practical (residual) networks at large (Imagenet) scale:
* Easy / hard training advantage — figure 1C (preceptron), 1D (CIFAR10).

* Better than powerlaw / exponential phase of individual constant hardness $f$ as well as resulting pareto — up to the metric limit (upon reverting to powerlaw). Predicted and shown for perceptron 1A, 1F, 3A.
This is consistent with observations in figure 3BCD. It is however hard to quantifiably assess the agreement of an actual exponent functional dependence in figure 3BCD or if it is already hitting a powerlaw limit.

* prediction of metric quality on transition to limiting powelaw, and minimal data hardness ($f_{min}$). At real data scale --- consistent with observed $f_{min}$ powerlaw limit and (a little, fortunate) evidence that improved metric via student-probe further training allows to breach said limit Appendix C figure 7.

Beyond theory:

* While the theory does not deal with transfer learning, it is in practice demonstrated that a steep benefit manifests when downstream pruning (notably, adding visual transformer network into the fray).
It is worth to dwell on this, as it is different from the results portrayed in the regular pruning scenario in figure 3BCD. In these experiments, as the total dataset size is fixed, and one is only choosing a subset according to hardness ($f$) then no significant performance gain is evident.
In contrast, in the downstream Transfer Learning experiment 4A, a noticeable advantage in performance manifests when training on *less*   data (a subset).

Implication strengths:

* Data curation exponential gain potential:
While the authors lead with computation implications, this requires further evidence (see below requests for clarification and concern that pruned data may actually be hard to train on). However, it should be noted, that even if computational gain at training is not available, there is significant value in reducing the burden of data collection and especially reducing data labeling quantities in supervised settings where these are expensive. These benefits are orthogonal to the potential computational savings.

* Practical data curation $\textit{without labels}$:
Thus, the introduction of an at-Imagenet-scale self-supervised metric which is able to match SOTA supervised data-pruning metrics, is a significant and novel (to the best of this reviewers knowledge) contribution.

The limitation here is that this assumes a strong (and is limited by its strength) metric (oracle / student-probe). See below elaboration on this in the weakness section.

$\textbf{Weaknesses}$

$\textit{Major:}$

1. Computation --- are pruned datasets harder to learn?:
add an analysis (theory and experiments) of the computational effect of training on pruned data. numbered (R1a-c) below:

In order to make claims regarding computational savings which are at the premise of the paper's motivations, these need to be evaluated. There may be reason for concern here, that a pruned dataset does not translate to computational savings. To elucidate this point, hard examples are expected to be iterated upon numerous times (e.g. giving rise to the very notion of the memorization metric []), thus potentially negating the perceived computational benefit of having fewer samples.
By way of example and request --- even for the simple perceptron learning cases evaluated --- what were the actual times until convergence. We know the classical upper bound is inversely proportional to the margin squared, i.e. to the hardness (governed by $f$). What does this imply on the expected time to convergence in the statistical-mechanics limit?
In other words: $\textbf{(R1a)}$ what is the theoretical convergence (e.g. relaxation to ground state from high energy initial state or other appropriate expected value) time, and $\textbf{(R1b)}$ convergence in terms of number of iterations in practice (empirically) of the trained preceptors ?

For assessing this question in the real data scenarios, there is a need to adopt a well calibrated learning schedule, such that convergence times may be (properly) compared, such that the computational benefit may be assessed.
While the paper mentions the comparison in terms of number of epochs and marginal benefit of further iterations in the real data experiments, consistent with a potential computational advantage, it seems that the training schedules are constant epochs [lines 727, 333]. Thus this does not necessarily demonstrate computational savings --- e.g. a different (shorter) schedule may have marginal (comparable) effect on the full data.

Previous work has dealt with this optimality question by using adaptive learning schedules and early stopping.

$\textbf{(R1c)}$ These CIFAR10, Imagenet computation scaling (at early stopping with adaptive learning) should be added to the paper also in order to significantly strengthen the assessment of the computational implications of data pruning (which as noted is one of the core impetuses the authors lead with, and rightfully so, as it is of significant practical implications).

A potential easing of this computationally intensive evaluation (involving the re-training and assessment of multiple models, demanding primarily at imagent scale) may be given by examining the learning curves:

Previous work [1] showed close relationship between the online and finite data settings under adaptive learning schemes. Since the setting of a constant $f$ falls into the setting investigated by these works, there is potential to verify this finding here on a single $f$ learning curve vs models on the curve (different amounts of samples) and then (assuming it holds) examine just the training curves of the other models on the full data.

2. The practical metric limitation.

The authors, rightfully, note the limitation imposed by the data pruning metric — effectively requiring a powerful oracle (or method) which sets a lower limit on the maximal pruning possible.

Not that this problem is not one of mere theoretical inconvenience: The exact common scenario where one encounters new data, on which employing a powerful metric for reduction, would be potentially pertinent, is the case where out-of-distribution characteristics limit any practical oracle (regardless of how well in-distribution it was trained).

For this reviewer, there is no question that amortization (e.g. capitalizing on mutual information between ID, OOD) is a powerful concept.
However a more quantitative analysis of the actual benefit and limitation would be beneficial.

To make this discussion more quantifiable (and interesting), let us consider the two scenarios of amortization effectively discussed in the paper:
Use foundation model to imbue a metric, data prune according to the metric, train new model on the pruned model
Use foundation model in transfer learning (TL) setting — e.g. fine tune on the new data.

$\textbf(R2)$ Is one superior to the other, in terms of computation? Performance? Quantifiably (theoretically?).

$\textit{Medium:}$

3. Related work:

$\textbf{(R3a)}$ Scaling laws --- while the authors mention at the preface notable scaling law works, they miss early work [3] on neural scaling laws, theoretical work [4, 5] as well as pertinent work corresponding with the beyond-powerlaw potential of data-scaling [6]. Also relevant is [7] which explores a resulting pruning pareto — in the complementary setting of parameter pruning, also finding a pareto frontier with varying optimal pruning fraction as a function of original model size (total parameters).

$\textbf{(R3b)}$ Exponential scaling with data is a known (classical) result when 'high-quality' side information is available --- a correspondence and comparison with query based learning classical expectations of the Pareto frontier is merited (Req):

The Pareto presented, while methodologically and theoretically insightful, would do well to correspond with the prior art which classically presents expectations for better-than-powerlaw (e.g. exponential) scaling with samples, \textit{when these are allowed to be chosen}. To put this in the most classical setting (pertinent to the paper) --- in the perceptron learning setting (finding a separating hyperplane), with access to a perfect oracle (i.e. $\theta=0$), binary search will result in an error scaling exponential with number of samples (queries).

So with access to additional data (the metric) it is a classical expectation to have, at least, exponential scaling with queries.
The explicit addition of this known classical expectation (e.g. binary search), in this reviewer's view, does not erode the novelty of the paper which shows early signs of this prediction in practice, but rather further solidifies its theoretical groundings.

$\textbf{(R3c)}$ Relatedly, corresponding with curriculum learning in the related work section is advisable --- e.g. as related to the notion of realizing the pareto frontier.

4. Implications of the hard/easy predictions on pareto:

The authors empirically evaluate a easy (large-margin p(z) specialization — blue in figure 1B) vs hard (small-margin, green in 1B) and show a predicted cross-over.
However, in the remainder of the paper, derivations and well as experiments, it is the small-margin case only which is considered (to the best of this reviewer's understanding), and in particular as giving rise to the pareto frontier and $f_{opt}(\alpha_{prune})$.
$\textbf{(R4a)}$ - This reviewer could not find the derivation of f_{opt} (figure 1E), please add it or if not derived mention that.

Further, in light of the crossover, it is this reader’s understanding that the true optimal pareto is upper bounded by a pareto $\textit{lower} $ than the one used throughout the paper. I.e. the crossover implies that taking the minimum of the large-margin (not explicitly derived) and small-margin pruning policies for any given point ($f, \alpha_{prune}$) has a section lower than the pareto in the paper, up to that switching point. $\textbf{(R4b)}$ — it would be good to at least make this point clear in the paper, and to put on a graph like 1A the pareto drawn by this minimum (the derivation, though tedious, would entail the complementary large margin truncated gaussian specialization of $p(z)$, alternatively… just empirically for the trained perceptrons).


The reason that even this, is viewed by this reviewer as an $\textit{upper bound}$ on the pareto frontier is that there is an expected optimal choice of pruning policy / metric usage $p(z|\alpha_{prune},f)$. This policy has a margin support as a function of $(\alpha_{prune},f)$ amenable to the paper derivation framework $\textbf{(R4c)}$ — is this is correct? If so, can you derive / or at least simulate it? Or, alternatively, show that it is expected to have negligible / non-important effect?).

5. experimental discriminative power of Imagenet pruning methods — need for repetition and clarification:

This is actually a very important methodological point, it is only making the medium cut, because even if all these methods are actually comparable, the concept of having a self supervised method in the ballpark of supervised, expensive, metrics is in-itself a significant contribution.
However, it would be much preferable if the authors modify the experiment such that more statistical confidence in the claims is readily observable — as it currently stands, the level of noise in the experiment (figure 5C) is high. $\textbf{(R5)}$ The self supervised experiments as well as the random baseline should have (at least) 3 repetitions, or alternatively, at least a X3 finer pruning grid (‘fractions of data kept’).
Especially worrisome is the drop of the proposed self-supervised method below random pruning.

Minor:

6. Clarity of content — general:

$\textbf{(R6)}$ Please add experimental descriptions where missing experiments and figures, reference to appendix OK. This reader spent significant time divining the experimental setups. Examples: how were the perceptron trained? Is this the classic perceptron learning algorithm? What were the grids of evaluation of all experiments (they vary across figures) etc.


7. Typos and minor clarifications:

Figure 1A — garbled $f$ values in legend

Caption of figure 1:

* “The read curve indicates … at fixed $\alpha_{prune}$ would be better understood if read “The read curve indicates … as function of $\alpha_{prune}$

* "E : Test accuracy … f_{opt} (blue curve)” — is it blue ?? looks purple, maybe change to black…

Line 197: “ … exponential scaling law (Fig 1C…” $\rightarrow$should be “ … exponential scaling law (Fig 1E…”


$\textbf{References:}$

[1] Nakkiran, Preetum, Behnam Neyshabur, and Hanie Sedghi. "The deep bootstrap framework: Good online learners are good offline generalizers." arXiv preprint arXiv:2010.08127 (2020).

[2] Hernandez, Danny, et al. "Scaling laws for transfer." arXiv preprint arXiv:2102.01293 (2021).

[3] A Constructive Prediction of the Generalization Error Across Scales. Jonathan S. Rosenfeld, Amir Rosenfeld, Yonatan Belinkov, Nir Shavit. 2020 International Conference on Learning Representations. https://openreview.net/forum?id=ryenvpEKDr

[4] Scaling Laws from the Data Manifold Dimension. Utkarsh Sharm, Jared Kaplan. 2022 Journal of Machine Learning Research. http://jmlr.org/papers/v23/20-1111.html

[5] Explaining Scaling Laws of Neural Network Generalization.Yasaman Bahri, Ethan Dyer, Jared Kaplan, Jaehoon Lee, Utkarsh Sharma 2021 https://arxiv.org/abs/2102.06701

[6] Scaling Laws for Deep Learning. Jonathan S. Rosenfeld. 2021. https://arxiv.org/abs/2108.07686

[7] Rosenfeld, Jonathan S., et al. "On the predictability of pruning across scales." International Conference on Machine Learning. PMLR, 2021.

---

> ### Author Response · Authors · 2022-08-02
> **Initial response to reviewer ycTR (Part 1/3)**
>
> Dear reviewer ycTR,
>
> We would like to thank you for your deeply thorough review - this is probably one of the most detailed reviews we have ever received, and it contains numerous helpful suggestions that we were excited to pursue and which have led to results we feel have meaningfully improved the paper! We’re very happy to hear that you found ours to be a **“paper of significant insight, methodological implications and multiple important contributions to the field”**. Please find a point-by-point response below.  Please find a point-by-point response below.
>
> > *$(\textbf{R1})$. Computation --- are pruned datasets harder to learn?: add an analysis (theory and experiments) of the computational effect of training on pruned data. numbered (R1a-c) below:*
>
> This is a very important point, and motivated by the reviewer’s comments we have set out to conduct as thorough an investigation as time permits.$(\textbf{R1a,b})$ First, although the reviewer is correct that the classical perceptron learning algorithm convergence time is upper bounded by one over the margin squared, our theory (at the price of a more involved calculation) studies the more powerful max-margin perceptron, which achieves good generalization performance even for small \alpha, and has a unique solution on separable data which can be efficiently found via quadratic programming (QP) with much faster convergence times than the classical perceptron. Guided by the reviewer’s comment we have investigated the convergence time of the QP algorithm and found that it does not depend on the margin (ie pruned datasets are not harder to learn), and instead is linearly proportional to the number of training examples, \alpha_prune. This allows us to replace the x-axis of Fig. 1A with compute (new Fig. 8A in the new appendix section C on “Breaking compute scaling laws via data pruning”), revealing that, at least for the perceptron, data pruning allows for exponential saving in compute, and showing excellent agreement with the theory. These experiments can be reproduced in the anonymous Colab notebook at: https://colab.research.google.com/drive/1in35C6jh7y_ynwuWLBmGOWAgmUgpl8dF?usp=sharing.  We thank the reviewer for leading us to this important point.
>
>
> > *$(\textbf{R1c})$  CIFAR10, Imagenet computation scaling (at early stopping with adaptive learning)*
>
> Unfortunately, finding optimal learning rate schedules for both pruned and unpruned ImageNet datasets is not something we can accomplish during the rebuttal period. However, fortunately, we believe this is not required. First, we note that we are starting with standard hyperparameters for training on ImageNet on the full dataset, which have already been reasonably well optimized over time.  Second, on the pruned datasets, importantly we did not change any learning hyperparameters, and even so we find excellent performance with no accuracy loss down to pruned datasets of size 80% of the original data.  We view this as a feature.  Namely, we don’t need to perform extensive hyperparameter re-optimization in conjunction with data-pruning in order to achieve good test error when training on much smaller pruned dataset sizes.  Indeed further optimization of learning hyperparameters on pruned datasets may lead to even better performance.  But the fact that any such expensive hyperparameter optimization is unnecessary to achieve the performance we did is, we feel, a strength of our work. (Future work may well be able to improve upon this, possibly via optimized schedules.)
>
> For CIFAR-10 we experimented with adaptive learning and early stopping but were not able to find a schedule which achieved better performance on the full dataset than the standard  learning rate schedule in our original experiments (which had already been optimized for performance on the full CIFAR-10 dataset, see experimental details section B).
>
> Also for both CIFAR-10 and ImageNet, we included a new section and Figs 8 B and C (new appendix section on “Breaking compute scaling laws via data pruning”) showing that learning curves are very similar across pruning fractions, indicating that pruned datasets are not harder to learn in practical settings.

---

> > ### Author Response · Authors · 2022-08-09
> > **Update to reviewer ycTR**
> >
> > Dear reviewer ycTR,
> >
> > Thank you again for your thorough feedback and excellent suggestions. We have gone through them in detail and incorporated them via several substantial additions to our manuscript (see our update above), which we feel have greatly improved the paper. As the discussion period is ending, we kindly ask for your reply to our response and to consider raising your score if we have adequately answered your concerns.

---

> ### Author Response · Authors · 2022-08-02
> **Initial response to reviewer ycTR (Part 2/3)**
>
> > *$(\textbf{R2})$. The practical metric limitation.*
>
> The reviewer is right to ask about the practical ability of a pretrained model to prune an out-of-distribution dataset – we highlight this as a fundamental constraint on data pruning (Fig. 2). Exponential scaling quickly breaks down if the pretrained model has inadequate overlap with the target function. However, we believe that Fig. 4a provides compelling evidence that pretrained models, finetuned for a short amount of time on a transfer task, can serve as a powerful pruning metric for transfer learning. A pretrained ViT can prune CIFAR-10 down to just 10% with no sacrifice in performance after finetuning for just 2 epochs (note that this effect is not simply due to pretrained models being good few-shot learners, because a random subset of 10% of CIFAR-10 does sacrifice performance). This striking observation suggests that data pruning may be even more powerful for transfer learning than for pretraining.
>
> Saying anything theoretically would require a model of transfer learning. While a detailed model is outside the scope of this work, a toy model which fits within our framework is a slightly modified version of Fig 2a, where the “probe student” is reinterpreted as the target function of the pretraining task, and the teacher “T” is reinterpreted as the target function of the downstream transfer task. In this model the ability of a pretrained model to prune the downstream task is governed by the overlap between the pretraining target function and the transfer target function. For example, since a ViT pretrained on ImageNet can prune CIFAR-10 to 10% with no sacrifice in performance, f_min (Fig. 2C) is at least as low as 10%, and hence the effective  “angle” between ImageNet and CIFAR-10 is at least as small as 3°. Measuring the “angle” between tasks based on the ability of pretrained models to prune them may be an interesting direction for future research.
>
> To directly answer the reviewer’s question, we believe it is overwhelmingly likely that finetuning a pretrained model on the transfer task will be superior to training a new model from scratch on a pruned version of the transfer task, both in terms of performance and compute. This intuition is based on the striking transfer abilities of foundation models, which acquire through pretraining a rich set of features which can be rapidly and flexibly adapted to new tasks (in the “modern” limit of very large pretraining datasets, a model trained from scratch on any finite transfer dataset will simply not be able to acquire these features). However, our work suggests a promising third option: use a foundation model to prune a transfer dataset, and then finetune the foundation model on the pruned dataset. Our experiments indicate that the transfer dataset may be pruned down to a very small number of maximally informative examples which allow the foundation model to quickly adapt to solve the new task.
>
> > *$(\textbf{R3a})$ Related work on scaling laws*
>
> Thanks for pointing these out.  In the additional space afforded by the camera-ready version we will add these important citations and discuss them.
>
> > *$(\textbf{R3b})$ Exponential scaling with data is a known (classical) result when 'high-quality' side information is available.*
>
> The reviewer’s intuition is spot-on, and classical works have shown that training examples can be carefully chosen so that each new example halves the version space (i.e. the space of weights consistent with the training data) of perfect linear classifiers, resulting in exponential scaling akin to binary search (see e.g. the Query by Committee paper which we cite in our related works section). Intriguingly, however, we find that our approach actually achieves super-exponential scaling, owing to the fact that it takes advantage of partial knowledge of the target function. In the extra space afforded by the camera ready version, we will expand our discussion and comparison to classical results. Please let us know if you have other references in mind we should include!
>
>
> > *$(\textbf{R3c})$ Relatedly, corresponding with curriculum learning in the related work section is advisable - e.g. as related to the notion of realizing the pareto frontier.*
>
> Excellent point. In the extra space afforded by the camera ready version, we will add some references to curriculum learning in our related works section.

---

> ### Author Response · Authors · 2022-08-02
> **Initial response to reviewer ycTR (Part 3/3)**
>
> > *$(\textbf{R4a})$. This reviewer could not find the derivation of f_{opt} (figure 1E), please add it or if not derived mention that.*
>
> $f_\text{opt}$ is obtained by maximizing Eq. 76 wrt f. The solution does not have a simple closed form, but we have laid out in detail the procedure for obtaining $f_{opt}$ in Appendix section A.6.
>
> > *$(\textbf{R4b,c})$: Does keeping easy examples when \alpha is small get you a lower Pareto frontier in the small \alpha regime? It would be good to at least make this point clear in the paper, and to put on a graph like 1A the pareto drawn by this minimum.*
>
> In response to the reviewer’s insightful question, we have performed a novel derivation, using the calculus of variations, to identify the optimal pruning policy $p(z|\alpha_\text{prune},f)$ among all possible pruning policies. This derivation reveals that the optimal data distribution along the teacher is a delta function $p(z)=\delta(z-z^*)$ whose location $z^*$ changes as a function of $ \alpha_\text{prune}$. When $\alpha_\text{prune}$ is small, $z^*$ approaches infinity and when $\alpha_\text{prune}$ is large $z^*$ approaches zero, confirming that the optimal pruning strategy when data is scarce is to keep the easy (large-margin) examples, while the optimal strategy when data is plentiful is to keep the hard (small-margin) examples. Intriguingly, this calculation also reveals that if the delta function is placed optimally as a function of $\alpha_\text{prune}$, the student can perfectly recover the teacher ($R=1$, zero generalization error) for any $\alpha_\text{prune}$.
>
> This observation, while interesting, is of no practical consequence because it relies on an infinitely large training set from which examples can be precisely selected to perfectly recover the teacher. Therefore, to derive the optimal pruning policy for a more realistic scenario, we assume a gaussian distribution of data along the teacher direction $p(z) \sim \mathcal N(0,1)$ and model pruning as keeping only those examples which fall inside a window a<z<b. For each f,alpha, we find the optimal location of this window using the method of Lagrange multipliers. Consistent with the results for the optimal distribution, the location of this window shifts from around infinity to around zero as $\alpha_\text{prune}$ grows. We further show that this single-window policy uniformly outperforms a two-window policy which keeps both easy and hard examples. We speculate that it will also dominate k-window policies, and hence may be the optimal policy when the data is gaussian-distributed along the teacher.
>
> As the reviewer anticipates, this optimal pruning policy traces out a Pareto frontier lower than the one drawn in Fig. 1A (see the new Fig 7 in our revised submission). But importantly it retains the same qualitative shape, including an exponential scaling of test error with $\alpha_\text{prune}$.
>
> Unfortunately we have not had time to finish writing up the full derivation in the Appendix, but we will continue to update the manuscript during the author-reviewer rebuttal period.
>
> > *$(\textbf{R5})$. experimental discriminative power of Imagenet pruning methods — need for repetition and clarification: The self supervised experiments as well as the random baseline should have (at least) 3 repetitions, or alternatively, at least a X3 finer pruning grid (‘fractions of data kept’). Especially worrisome is the drop of the proposed self-supervised method below random pruning.*
>
> Good point! To address your suggestion, we have trained five repetitions for Figure 5C to reduce the noise from random variation across different model training runs, and we have updated the figure. Regarding the cross-over (drop below random pruning), at some point this is actually a qualitative finding predicted from theory (Figure 1BC): when data is abundant, the best strategy is to keep hard examples; when data is scarce (as it becomes for small “fractions of data kept” in Figure 5BC), however, this strategy becomes suboptimal and one should instead keep easy examples. That being said, the data point below random pruning for frac. data kept = 0.6 appears to have been an outlier: in the updated plot (5 models/data point), this data point is above random pruning.
>
> > *$(\textbf{R6})$. Please add experimental descriptions where missing experiments and figures, reference to appendix OK. This reader spent significant time divining the experimental setups. Examples: how were the perceptron trained? Is this the classic perceptron learning algorithm? What were the grids of evaluation of all experiments (they vary across figures) etc.*
>
> We have added a detailed description of all perceptron learning experiments in Appendix B. Furthermore, the updated manuscript contains a link to a google Colab with code to reproduce all of the theory figures and associated perceptron learning experiments throughout the paper (https://colab.research.google.com/drive/1in35C6jh7y_ynwuWLBmGOWAgmUgpl8dF?usp=sharing).

---

> ### Author Response · Authors · 2022-08-02
> **Initial response to reviewer ycTR (Additional comments)**
>
> > *7. Typos and minor clarifications:*
>
> Thanks, fixed now.
>
> > *It is an open question how training on pruned data affects robustness to naturally occurring phenomena (e.g. OOD performance).*
>
> Interesting point! We have added a detailed investigation of the OOD performance of pruned models in the Appendix (J - Out-of-distribution (OOD) analysis of dataset pruning). Our core take-away is: While more analyses would be necessary to investigate whether pruning (with a decent metric) consistently helps on OOD performance, it seems safe to conclude that it does not hurt OOD performance compared to an accuracy-matched baseline.
> If IID accuracies are strongly affected by pruning, one can expect OOD accuracies to fall of as well, as predicted from “Accuracy on the line: on the strong correlation between out-of-distribution and in-distribution generalization” paper (Miller et al., ICML ‘21).

---

### Official Review · Reviewer_NsH6 · 2022-07-11

**Rating:** 9
**Confidence:** 4
**Soundness:** 4 excellent
**Presentation:** 4 excellent
**Contribution:** 4 excellent

**Summary:**

This paper demonstrates (both theoretically and empirically) that power-law scaling of the error with dataset size can be improved to exponential scaling using high-quality data pruning metrics. First, significant theory work is developed to back this prediction, which is then validated at small scale with ResNet models trained on CIFAR10. The authors notably study the effect of an imperfect "oracle" to inform the data pruning metric, showcasing the importance of the oracle quality. Finally, they demonstrate their approach at larger scale with ResNet/ViT models, and study various oracle metrics, introducing their own self-supervised  as well, which performs competitively against costlier metrics.

**Questions:**

_(Questions/suggestions below follow the numbering from the section above.)_

* **S1**. The paper could benefit from clarifying the fixed multi-epoch setting earlier to streamline reading.

For the rest of the points mentioned in the section above, I do not expect any direct answer for a rebuttal round, unless the authors have available experiments/justifications to these points.

**Limitations:**

The authors discuss the limitations of their approach thoroughly throughout the paper.

**Strengths And Weaknesses:**

## Strengths

This is an outstandingly good paper, which benefits from: (1) strong and thorough theoretical backing; (2) great empirical design to validate the theoretical findings and demonstrate them in "real-world" conditions; (3) good clarity throughout with high-quality figures and an easy-to-follow line of reasoning.

The proposed idea is obviously very interesting, as improving over scaling laws could yield to significant gains across the board. Many studies have looked at this under the angle of architectural change, but the data angle taken here is under-explored.

## Weaknesses
*(All the points below are indexed and linked to questions in the next section, which the authors can answer to clarify/improve the paper. They are ordered by level of importance.)*

* **W1. _(Minor)_ The fixed multi-epoch setting should be clarified and mentioned earlier.** From Appendix B and the discussion l332, the method operates in a "fixed epoch" setting. Data is pruned, and then comparisons are made for the same number of training epochs as when training on the full dataset. This is not immediately evident from reading the paper, and clarifying it from the start would help streamline the reading of this work. Similarly, it would be interesting to consider such approches for pretraining (single-epoch) -- but this is beyond the scope of the paper.

* **W2. _(Minor)_ Statement around 10x more training data only reduces loss from 3.4 nets to 2.8 nats feel a bit overstated.** On l27 and l128, the authors indicate that an order of magnitude in data only changes less by 0.6 nats. This is strictly-speaking true, but a bit misleading: cross-entropy loss is an a log-scale, so small changes are actually larger than they appear. Moreover, in large language models, small changes in upstream loss can have dramatic effect on downstream tasks.

---

> ### Author Response · Authors · 2022-08-02
> **Initial response to reviewer NsH6**
>
> Dear reviewer NsH6,
>
> Thank you for your time to review our paper and your highly positive assessment of our work. We were humbled to read that you found the paper **“outstandingly good”** and happy that you seem to share our excitement! Please find detailed responses to your insightful questions/suggestions below:
>
> > _W1/S1. The paper could benefit from clarifying the fixed multi-epoch setting earlier to streamline reading._
>
> Thank you for pointing this out. We will emphasize in the main paper that we train on the pruned dataset for the same number of epochs as on the full dataset (e.g. for the results in Figure 5), thereby achieving more rapid training due to dataset size reduction.
>
> > _W2. Statement around 10x more training data only reduces loss from 3.4 nats to 2.8 nats feel a
> bit overstated._
>
> Thanks for letting us know! In the extra space afforded by the camera ready, we will remind the reader that nats is on a logarithmic scale and small improvements in nats can lead to large improvements in downstream tasks. This is also why we included the discussion of vision transformers and their slow scaling with respect to ImageNet accuracy.
>
> Thanks again for taking the time to review our paper.

---

> > ### Comment · Reviewer_NsH6 · 2022-08-07
> > **Response to rebuttal**
> >
> >  All my (minor) concerns/recommandations are addressed by the rebuttal above.
> >
> > Thank you for taking the time to answer thoroughly to each reviewer.

---

> > > ### Author Response · Authors · 2022-08-09
> > > **Second response to reviewer NsH6**
> > >
> > > Thank you again for your comments and your recommendations!

---

### Author Response · Authors · 2022-08-02
**Author’s overview of response to all reviewers**

We would like to thank all four reviewers for their highly constructive feedback!

We very much appreciate their assessment of our work as an **“outstandingly good paper”** with **“thorough theoretical backing”** and **“great empirical design”** (NsH6); an **“exciting paper for its bridge of phenomena and theory”** with **“significant insight and multiple important contributions to the field”** (ycTR), **“interesting and novel insights”**  targeting an **“important problem”** which **“evaluates its claims on a wide range of datasets…with strong theoretical foundation”** (QtTy) as well as **“thorough evaluation”** with a **“wealth of empirical experiments”** which will be **“valuable to future work in the area”** (D9bn).

We received excellent suggestions for improvement which we feel will substantially strengthen the paper; here is a summary of main suggestions/concerns and how we addressed them:

- We added a detailed 2.5-page OOD performance analysis of pruned models in App. K (ycTR)
- We performed a novel derivation, using the calculus of variations, to determine the theoretically optimal data pruning strategy for all f and $\alpha$ (ycTR).
- We added an additional figure and App. section C analyzing the compute scaling of data pruning in the perceptron, as well as an investigation of learning curves on pruned versus unpruned datasets, opening the door to substantial compute savings via data pruning (ycTR, D9bn)
- We are currently training more models per datapoint in Figure 5C to reduce the noise (ycTR)
- We identified points where our writing can be improved for better clarity and presentation (QtTy, NsH6, D9bn, ycTR)
- We replaced image samples from a class showing people with a different class to address potential privacy concerns
- Since we are already at the page limit, we will add further clarifications, references and discussion in the extra space (+1 page) afforded by the camera ready version (ycTR, D9bn, QtTy, NsH6)

Thanks again to all reviewers for their time and feedback!

---

### Author Response · Authors · 2022-08-09
**Author's update: revised manuscript**

Now that the rebuttal period has ended, we would like to thank all four reviewers again for their valuable feedback. We have uploaded a revised manuscript with several major additions motivated by the reviewers’ suggestions. At a high level, these additions include,

* **Appendix C (& new fig 8)**: An investigation of the compute savings afforded by data pruning for the perceptron and ResNets trained on CIFAR-10 and ImageNet.
* **Appendix J (& new figs 24,25)**: A detailed 2.5-page OOD performance analysis of pruned models (ycTR)
* **Appendix A8 (& new fig 7)**: A derivation of the optimal data pruning strategy for all $f$ and $\alpha$, which generalizes the “keep easy” and “keep hard” strategies, and lower-bounds the Pareto frontier (ycTR).
* **Updated Fig 5C** with 5 trained models per datapoint to reduce noise, added error bars (ycTR)
* **Updated Fig 7** (now Fig 10) with additional experiments and more careful quantification of scaling behavior (D9bn)
* Improved clarity and presentation throughout the text (QtTy, NsH6, D9bn, ycTR)

We kindly ask the reviewers to look through the updated manuscript and welcome their feedback. We believe these additions, inspired by the reviewers’ comments, have meaningfully improved the clarity and impact of our paper. Since we are already at the page limit, we will add further clarifications, references and discussion in the extra space (+1 page) afforded by the camera ready version.

---

### Meta-Review · Area_Chair_JpQc · 2022-08-26

**Recommendation:** Accept
**Confidence:** Certain

**Metareview:**

The paper provides theory and experiments that power law scaling of error with respect to dataset size can be improved to "exponential" scaling by high-quality intelligent data pruning. The paper first provides a theoretical model to study the effect of pruning and show that it matches well in experiments. Beyond this, the paper also observes similar behavior on real datasets. The analysis demonstrates the possibility of beyond-power law scaling by utilizing pruning metrics in large scale settings.

Reviewers saw many strength on the work: as an “outstandingly good paper” with “thorough theoretical backing” and “great empirical design” `NsH6`; an “exciting paper for its bridge of phenomena and theory” with “significant insight and multiple important contributions to the field” `ycTR`, “interesting and novel insights'' targeting an “important problem” which “evaluates its claims on a wide range of datasets…with strong theoretical foundation” `QtTy` as well as “thorough evaluation” with a “wealth of empirical experiments'' which will be “valuable to future work in the area” `D9bn`. Authors addressed questions and weaknesses raised by the reviewers, during the very active author-reviewer discussion period. The AC encourages the authors to incorporate all the agreed suggested changes in camera ready version.

AC agrees with the reviewers sentiment that the study and insight will be important to the broad NeurIPS audience where correctly characterizing scaling behavior of neural networks is becoming even more important.


**Award:**

Yes

---

### Decision · Program_Chairs · 2022-09-14

Accept